

# On biotic and abiotic drivers of the microphytobenthos seasonal cycle in a temperate intertidal mudflat: a modelling study

Raphaël Savelli[1], Christine Dupuy[1], Laurent Barillé[2], Astrid Lerouxel[2], Katell Guizien[3], Anne Philippe[1], Pierrick Bocher[1], Pierre Polsenaere[4], and Vincent Le Fouest[1]

[1]Littoral, Environnement et SociétéS (LIENSs), Université de La Rochelle, UMR 7266, CNRS-ULR, 2 rue Olympe de Gouges, 17000 La Rochelle, France
[2]Mer Molécules Santé (MMS) - EA 21 60, Université de Nantes, Laboratoire Mer Molécules Santé, 2 rue de la Houssinière, 44322 Nantes Cedex, France
[3]CNRS-Université Pierre et Marie Curie, UMR 8222 Laboratoire d'Ecogéochimie des Environnements Benthiques, Observatoire Océanologique de Banyuls-sur-Mer, UMR8222, rue du Fontaulé, 66650 Banyuls-sur-Mer, France
[4]IFREMER, Laboratoire Environnement Ressources des Pertuis Charentais (LER/PC), BP7, 17137 L'Houmeau, France

*Correspondence to:* Raphaël Savelli (raphael.savelli1@univ-lr.fr)

**Abstract.**

Microphytobenthos (MPB) from intertidal mudflats are key primary producers at the land-ocean interface. MPB can be more productive than phytoplankton and sustain both benthic and pelagic higher trophic levels. The objective of this study is to assess the contribution of light, mud temperature, and gastropod *Peringia ulvae* grazing pressure in shaping the seasonal

5  MPB dynamics on the Brouage mudflat (NW France). We use a physical-biological coupled model applied to the sediment first centimeter for the year 2008. The simulated data compare to observations including time-coincident remotely sensed and *in situ* data. The model suggests a MPB annual cycle characterized by a main spring bloom, a biomass depression in summer, and a moderate fall bloom. In early spring, high simulated photosynthetic rates due to mud surface temperature (MST) values close to the MPB temperature optimum for photosynthesis and to increasing solar irradiance trigger the onset of the MPB

10  spring bloom. After the bloom, high MST values lead to synoptic events when MPB thermo-inhibition (39.5 % of summer) and limitation by *P. ulvae* grazing (8.7 % of summer) superimpose. During these synoptic events, 14 % of the simulated annual MPB primary production is channeled towards the *P. ulvae* secondary production through ingestion. The model suggests that such a combined effect is highly linked to the MPB biomass depression in summer. The model ability to infer on biotic and abiotic mechanisms driving the seasonal MPB dynamics could open the door to a new assessment of the export flux of biogenic

15  matter at the land-ocean interface and, more generally, of the contribution of productive intertidal biofilms to the coastal carbon cycle.




# 1 Introduction

Coastal and nearshore waters receive large amounts of organic matter and inorganic nutrients from land that support a high biological productivity (Mann, 1982; Admiraal, 1984; Hopkinson and Smith, 2005). However, the high turbidity of estuarine influenced coastal waters limits the penetration of downward solar irradiance in the water column and, as such, phytoplankton

production (Cloern, 1987; Struski and Bacher, 2006). In subtidal and intertidal zones, primary production (PP) sustained by benthic microalgae, or microphytobenthos (MPB), can equal or exceed that of phytoplankton (Underwood and Kromkamp, 1999; Struski and Bacher, 2006). MPB are mostly composed of free motile epipelic diatoms and of epipsammic diatoms that live in close association (attached or free-living) with sediment grains (Round, 1971). Epipelic MPB are associated with fine cohesive intertidal sediments and developp within the top few millimiters (Underwood, 2001). During daytime emersion, they

migrate at the sediment surface constituting a dense biofilm of a few hundred micrometers (Herlory et al., 2004). They are fully exposed to solar irradiance at low tide promoting PP that can reach values as high as $1.9\,\mathrm{g\,C\,m^{-2}\,d^{-1}}$ (Underwood and Kromkamp, 1999). During the flood, epipelic MPB move downward within the sediment but can be resuspended into the water column (Demers et al., 1987; de Jonge and van Beusekom, 1992, 1995; Lucas et al., 2001; Orvain et al., 2004; Ubertini et al., 2012). Both epipelic and epipsammic MPB are a key resource for higher trophic levels from benthic fauna to birds on bare

mudflats (Herman et al., 2000; Kang et al., 2006; Jardine et al., 2015), but also for pelagic organisms such as zooplankton and planktivorous fishes (Perissinotto et al., 2003; Krumme et al., 2008).

On intertidal mudflats, MPB PP rates are mainly constrained by solar irradiance and temperature (Barranguet et al., 1998). The MPB biofilm faces strong daily and seasonal variations of mud surface temperature (MST) caused by heating through solar irradiance during low tide exposure periods (Harrison and Phizacklea, 1985; Harrison, 1985; Guarini et al., 1997) and

develops phenological adaptations. Blanchard and Cariou-Le Gall (1994), Barranguet et al. (1998) and Pniewski et al. (2015) showed a light-related seasonal adjustment of photosynthetic parameters (the photosynthetic capacity $P^b_{max}$ and the light saturation parameter $E_k$) from Production-Irradiance (P-E) curves fitted on the model of Platt and Jassby (1976). Photo-inhibition was rarely observed in the field since epipelic diatoms can achieve "micro-migrations", i.e. a negative phototaxic short-term change of position in the sediment (Kromkamp et al., 1998; Perkins et al., 2001; Cartaxana et al., 2011). With

respect to mud temperature, Blanchard et al. (1996) related mathematically $P^b_{max}$ to temperature. Using this relationship, Blanchard et al. (1997) showed that $P^b_{max}$ varies according to seasons suggesting a thermo-inhibition process in response to high mud temperature (> 25 °C). de Jonge (1980) also showed seasonal variations of the carbon (C) to chlorophyll $a$ (Chl $a$) ratio, which is a proxy of the physiological state of autotrophic cells, as a function of air temperature (de Jonge et al., 2012). Regarding nutrients, their limiting role on the MPB growth and photosynthetic rate is not evidenced in fine cohesive

sediments naturally enriched both from within the sediment and the water column (Underwood, 2001; Cadée and Hegeman, 1974; Admiraal, 1984). Vieira et al. (2016) suggested a likely *in vitro* limitation by dissolved inorganic carbon within biofilms. Benthic diatoms were shown to store ammonium and phosphate within the intracellular matrix (García-Robledo et al., 2010; Yamaguchi et al., 2015) potentially usable for assimilation and growth (Garcia-Robledo et al., 2016). The nutrient limitation of MPB is still in debate.





At temperate latitudes, the seasonal cycle of MPB is shaped by the prevailing environmental conditions. Seasonal blooms are reported to occur throughout the year, i.e. in spring (De Jong and de Jonge, 1995; Sahan et al., 2007; Brito et al., 2013), summer (Cadée and Hegeman, 1977) and fall (Hubas et al., 2006; Garcia-Robledo et al., 2016). Along the French Atlantic coast, the spring bloom and summer depression observed in the Brouage mudflat in the Marennes-Oléron Bay are explained

by optimal temperature conditions and thermo-inhibition, respectively (Blanchard et al., 1997). Reported differences in the observed MPB seasonal cycles are also attributed to the diatom assemblage (Underwood, 1994). In terms of biomass, epipelic diatoms associated with muddy sediments show a higher seasonality caused by a marked exposure to stressful environmental conditions (e.g. cycle of deposition/erosion, dessication, grazing) than less motile epipsammic species buried in coarser sandy sediments (Underwood, 1994). In summer, thermo-inhibition and a high grazing pressure by deposit feeders are suggested to

dampen the MPB biomass (Cadée and Hegeman, 1974; Cariou-Le Gall and Blanchard, 1995; Sahan et al., 2007). On intertidal mudflats, the prosobranch gastropod *Peringia ulvae* can reach densities up to $30\,000\ \mathrm{snails\,m^{-2}}$ (Sauriau et al., 1989) with a reported maximal ingestion rate of $385\ \mathrm{ng\,chl}\,a\,\mathrm{snail^{-1}\,h^{-1}}$ (Coelho et al., 2011). Such grazing activity may translate into a theoretical uptake of $12\ \mathrm{g\,C\,m^{-2}\,d^{-1}}$ for a C:Chl $a$ ratio of $45\ \mathrm{g\,C\,g\,chl}\,a^{-1}$ (Guarini, 1998), which is 6-fold more than the daily maximum MPB PP rate reported for MPB (Underwood and Kromkamp, 1999).

In light of the current knowledge, the role of the abiotic and biotic factors involved in the MPB seasonal dynamics is still unclear. This impedes any future assessment on how global change might impact the MPB dynamics and carbon cycle in the land-ocean continuum. The goal of this study is to quantify the relative contribution of light, temperature and grazing on the MPB seasonal cycle and production on an intertidal mudflat (Marennes-Oléron Bay) of the French Atlantic coast. For this purpose, we use a two-layer physical-biological model representing the MPB and *P. ulvae* compartments to assess the

contribution of the three drivers over an annual cycle. In the paper, we describe first the physical-biological coupled model and the *in situ* and remotely sensed data used to investigate the MPB seasonal cycle. Second, we assess the relative contribution of light, MST and *P. ulvae* grazing on the MPB dynamics and PP, and we analyze the model sensitivity to key biological constants. Finally, we discuss the role of light, temperature and grazing in the MPB seasonal cycle and the future challenges of modelling the MPB contribution to the carbon cycle at the land-ocean continuum.

## 2 Material and methods

The study area is the Pertuis Charentais sea on the French Atlantic coast. It is a shallow semi-enclosed sea characterized by semi-diurnal tides and a macrotidal regime. The tidal range is $\sim 6$ m during spring tides. The intertidal zone has two main mudflats composed of fine cohesive sediments, i.e. the Brouage mudflat ($42\ \mathrm{km^2}$) and the Aiguillon mudflat ($28.7\ \mathrm{km^2}$) (Fig. 1). The study site (45°54'50"N, 01°05'25"W) is located on the Brouage mudflat (Fig. 1). It is composed of fine cohesive

sediments (median grain size of 17 μm and 85 % of grain with a diameter lower than 63 μm; Bocher et al., 2007) and sheltered from Atlantic swells by the Ile d'Oléron (Pascal et al., 2009).



## 2.1 Observations

A large multiparametric dataset of physical and biological measurements collected in the Pertuis Charentais was used to constrain the model run and to compare with the model outputs. We provide here a summary of the data used along with their respective references, where a detailed methodology of each measurements can be found.

### 2.1.1 *In situ* data

Atmospheric and hydrologic forcings were required to set the temperature and light environment that constrained the physical/biological model. Atmospheric forcings (Fig. 3a-e) consisted in meteorological observations (shortwave radiation, air temperature in the shade, atmospheric pressure above the sea, wind speed and relative humidity) acquired at the Meteo France weather station located near the airport of La Rochelle (46°10'36"N, 1°11'3"W; data available online: https://publitheque.meteo.fr;

Fig. 1). Hydrology was represented by the absence or presence of seawater at the study site of the Brouage mudflat. Emersion/immersion periods were determined by the observed water height at the tide gauge of La Rochelle-La Pallice (46°9'30"N, 1°13'14"W, data Service Hydrographique et Océanographique de la Marine (SHOM) / Grand Port Maritime La Rochelle-La Pallice; data available online: http://data.shom.fr/) corrected by the bathymetry at the study site. The bathymetry (3.204 m above chart datum) was extracted from a digital elevation model (Litto3D® 2010 Charente Maritime by the Institut National

de l'Information Géographique et Forestière (IGN) and the SHOM) at pixels corresponding to the study site (Fig. 1). The weather and tide gauge stations were located ∼ 30 km away from the study site. Atmospheric and hydrologic forcings were one hour frequency from January 1, 2008 (00:00 AM) to December 31, 2008 (11:00 PM). They were linearly interpolated at the time step of the model (6 min).

In order to validate the model, we used daily measurements of MST (1$^{st}$ cm), Chl *a* concentration (1$^{st}$ cm) and *Peringia*

*ulvae* biomass and density from a multiparametric dataset collected in February 16 - 24 and July 13 - 26, 2008 at the study site where the model was run (45°54'50"N, 01°05'25"W, Fig. 1). The sampling protocol is fully detailed in Orvain et al. (2014). Monthly data of *P. ulvae* abundance and biomass sampled monthly from April, 2014 to July, 2015 over the Aiguillon mudflat were used to estimate a monthly mean individual weight. The monthly mean individual weight was used to convert the simulated biomass per unit of surface into density per unit of surface. The sampling protocol is given in Bocher et al. (2007).

We averaged spatially the *P. ulvae* abundance and biomass data to obtain a monthly mean value for the entire mudflat. Ash-free dry mass (AFDM) was converted to carbon using the relationship derived from Jansson and Wulff (1977) and Remmert (2013) and used by Asmus (1994) for benthic deposit feeders (1 g AFDM = 0.58 g C). When the individual weight was not available, the individual height was used to estimate the AFDM (mg) using the formulation of Santos et al. (2005):

$$AFDM = 0.0154H^{2.61}, \tag{1}$$

where $H$ is the total individual height (mm).



### 2.1.2 Remote sensing data

Moderate Resolution Imaging Spectroradiometer (MODIS) images from the Terra satellite were downloaded from the USGS Earth Resources Observation and Science Center (http://earthexplorer.usgs.gov/). The Terra MODIS Surface Reflectance Daily L2G Global 250m SIN Grid product (MOD09GQ) contains 250-m surface reflectance in a red band (620-670 nm, band center

at 645 nm) and a near-infrared band (841-876 nm, band center at 859 nm). Terra data were used because the morning-pass (10-11h Universal Time) is better adapted than Aqua MODIS data to observe spring low tides at our study site. The data were corrected of atmospheric effects (aerosol, water vapor) and each image was checked for clouds/cirrus and cloud shadows. Cloud-free low-tide scenes were selected to apply a vegetation index. Images were reprojected to UTM/WGS84 coordinate system. The Normalized Difference Vegetation Index (NDVI; Tucker, 1979) was calculated with the reflectance ($\rho$) in the red

(R) and near-infrared (NIR) bands :

$$\mathrm{NDVI} = \frac{\rho(\mathrm{NIR}) - \rho(\mathrm{R})}{\rho(\mathrm{NIR}) + \rho(\mathrm{R})} \qquad (2)$$

The NDVI thresholds proposed by Méléder et al. (2003) to identify MPB with SPOT images was adapted for MODIS data and a range of 0 to 0.35 was used in this study. Negative NDVI values were associated with water and null values to bare sediment, while values higher than 0.35 corresponded to macrophytes (macroalgae and seagrass). For the present study, a NDVI time-

series was extracted for 2008 (47 scene images) at pixels corresponding to the study site (Fig. 1). Scene images were processed with the ENVI® software.

### 2.2 The coupled physical-biological model

The coupled model consisted in a mud temperature model coupled to a 3-compartment biological model. The mud temperature model was a thermodynamic model developed by Guarini et al. (1997) resolving heat fluxes at the surface in a 1-cm thick

sediment layer. Equations are given in Appendix A and Table A1. It was calibrated and validated on the Brouage mudflat by Guarini et al. (1997). During emersion periods, the simulated MST resulted from heat exchanges between the sun, the atmosphere, the sediment surface, from the conduction between mud and air and from evaporation (Fig. 2). Thermal conduction between mud and seawater was the only heat exchange during immersion periods (Fig. 2). The seawater temperature was simulated according to heat fluxes resulting from thermal conduction between air and seawater, from upward seawater radiation,

and from downward solar and atmospheric radiation. The simulated mud temperature was considered homogeneous at the horizontal scale. The heat fluxes were determined according equations given in Table A1 (Appendix A). The MST differential equation (Eq. A1 in Appendix A) was solved with an Euler Cauchy algorithm at a 30-sec time step.

The mud temperature model constrained a 3-compartment biological model, which was modified from Guarini (1998) and Guarini et al. (2000). It is fully detailed in Appendix B. MPB was represented by two compartments including the Chl $a$ con-

centration in the $1^{st}$ cm sediment ($F$, mg chl $a$ m$^{-2}$) and the Chl $a$ concentration within the surface biofilm ($S$, mg chl $a$ m$^{-2}$). The variable $S*$ represents the $S$ compartment that incorporated the $S$ instantaneous production of biomass (mg chl $a$ m$^{-2}$). The model assumed no sediment erosion nor deposition and no horizontal movement of MPB within the sediment. It included





a scheme of MPB vertical migration between the $S$ and $F$ compartments (Guarini, 1998; Guarini et al., 2000). The migration scheme is summarized in Table 1. The MPB growth rate was constrained by the photosynthetically active radiation (PAR) intensity, the simulated MST, and the grazing pressure. The grazing pressure was represented through a new scalar, $Z$, representing the *P. ulvae* biomass ($\mathrm{mg\,C\,m^{-2}}$). *P. ulvae* is a very abundant MPB grazer on the Pertuis Charentais intertidal mudflats

(Sauriau et al., 1989). The *P. ulvae* growth rate was constrained by the simulated MPB biomass and the MST. The fourth-order Runge-Kutta method was used to solve the biological differential equations with a 6-min time step.

The coupled physical-biological model was run at the study site (Fig. 1) from 1 January to 30 December, 2008. Initial conditions were $100\ \mathrm{mg\,chl}\,a\,\mathrm{m^{-2}}$ for $F$ and $1000\ \mathrm{mg\,C\,m^{-2}}$ for $Z$. No biomass was set for $S$ at the beginning of the simulation as it started at midnight (i.e. no light). The initial MST was initialised at the seawater temperature (see Eq. A5-8 in

Appendix A) at the first period of immersion. A 2008 10-year spin-up was performed before the analysis of the model outputs. The spin-ups and initial biomass conditions allowed for the convergence towards similar values of biomass at the end of each run.

We performed a sensitivity analysis to quantify how all-at-a-time variations in key biological parameters would impact the simulated MPB PP. A Monte-Carlo fixed sampling method (Hammersley and Handscomb, 1964) was used to randomly select

values of $T_{opt}$, $T_{max}$, $T_{opt_Z}$, $\alpha_Z$, $E_k$ and $K_E$ within observed ranges (Table 3). A total of 10,000 model runs was performed with the same previous initial conditions. Statistical metrics on simulated annual PP according to parameters values and variations (Pearson's correlation coefficient and parameters average, normalized standard deviation, minimum and maximum) were computed.

## 3 Results

### 3.1 Mud surface temperature

The simulated temperature of surface mud followed the seasonal cycle of air temperature (Fig. 3d and Fig. 4). During winter-spring (November to April), the simulated mud temperature was $8.6 \pm 3.6\ ^\circ\mathrm{C}$ in average. The simulated mean temperature was twice in summer-fall (May to October) reaching $16.9 \pm 3.9\ ^\circ\mathrm{C}$. The seasonal amplitude of the simulated mud temperature was higher in summer-fall ($27\ ^\circ\mathrm{C}$) than in winter-spring ($18.8\ ^\circ\mathrm{C}$). At the synoptic scale, the model reasonably simulated the high

frequency (1 min) variations of MST measured at the study site in February and July 2008 (Fig. 4).

### 3.2 MPB dynamics

In 2008, $S*$ reached a maximum value of $31\ \mathrm{mg\,chl}\,a\,\mathrm{m^{-2}}$ on February 24. The daily maximum values of $S*$ biomass simulated by the model for 2008 were subsampled to match the 2008 NDVI time series data (Fig. 5). Three distinct seasonal phases were identified in both time series using the amplitude of sign change of the $S*$ and NDVI second order time derivatives (Fig. 5). The phase 1 corresponded to the spring bloom during which the biomass in the biofilm and the NDVI data reached their

seasonal maximum value (day 1 to 144 and day 1 to 165 in the NDVI and model data, respectively). The phase 2 coincided



with a summer depression in the simulated MPB biomass and NDVI data (day 145 to 270 and day 166 to 256 in the NDVI and model data, respectively). Finally, the phase 3 showed an increase of both the simulated biomass and NDVI values suggesting a fall bloom (day 271 to 366 and day 257 to 366 in the NDVI and model data, respectively). With respect to the NDVI data, the model showed a 21 days and 14 days longer spring and fall bloom, respectively, and a 35 days shorter summer depression (Fig.

5). Overall, the seasonal cycle of the simulated MPB biofilm compared to that depicted by the remotely sensed NDVI data.

At the seasonal scale, the total MPB biomass ($S + F$) simulated by the model within the 1$^{\text{st}}$ cm sediment was the lowest in January and August ($\sim 30 \, \mathrm{mg \, chl} \, a \, \mathrm{m}^{-2}$) and reached a seasonal maximum in April ($\sim 238 \, \mathrm{mg \, chl} \, a \, \mathrm{m}^{-2}$, Fig. 6a). The seasonal cycle of the MPB biomass was related to the simulated mean time spent by a MPB cell at the sediment surface ($\tau_s$; Fig. 6ab). During daytime emersion time, a MPB cell spent approximatively 54 minutes in January and August and 12

minutes in April at the sediment surface because of the inverse relationship between total MPB biomass and $\tau_s$ (Eq. B5 in Appendix B1). The model reproduced the fortnightly tidal cycle with maximum values of MPB biomass simulated in spring tides (Fig. 6a). The simulated values of biomass of MPB were compared to 2008 time coincident observations (Fig. 6a). In February 2008, the simulated biomass was about $137 \pm 16.3 \, \mathrm{mg \, chl} \, a \, \mathrm{m}^{-2}$, which was close but significantly higher compared to the measured total MPB biomass ($106.5 \pm 11.3 \, \mathrm{mg \, chl} \, a \, \mathrm{m}^{-2}$; Mann Whitney test: p-value < 0.05). In July 2008, the model

also overestimated ($98 \pm 26.4 \, \mathrm{mg \, chl} \, a \, \mathrm{m}^{-2}$) the observed ($58.6 \pm 10.3 \, \mathrm{mg \, chl} \, a \, \mathrm{m}^{-2}$) MPB biomass (Mann Whitney test: p-value < 0.05), more than in February. Overall, the model simulated values that reasonably compared, in average, to matchup measurements gathered.

As for Chl $a$, the simulated daily mass-specific photosynthetic rates were higher in late winter-spring ($0.54 \pm 0.1 \, \mathrm{mg \, C} \, (\mathrm{mg \, chl} \, a)^{-1} \, \mathrm{h}^{-1}$) than in summer ($0.41 \pm 0.06 \, \mathrm{mg \, C} \, (\mathrm{mg \, chl} \, a)^{-1} \, \mathrm{h}^{-1}$) and fall-early winter ($0.25 \pm 0.13 \, \mathrm{mg \, C} \, (\mathrm{mg \, chl} \, a)^{-1} \, \mathrm{h}^{-1}$) (Fig. 6c).

The annual mean of the simulated MPB growth rate in the biofilm was $0.24 \pm 0.07 \, \mathrm{d}^{-1}$ with a range of values between 0.04 $\mathrm{d}^{-1}$ and 0.43 $\mathrm{d}^{-1}$. In the model, the MPB growth rate was related to the C:Chl $a$ ratio (see Eq. B9 in Appendix B2). The simulated mean C:Chl $a$ ratio varied in a range of values between 16 and $75.5 \, \mathrm{g \, C \, g \, chl} \, a^{-1}$ (with a mean value of $34.1 \pm 17.2$ $\mathrm{g \, C \, g \, chl} \, a^{-1}$ in winter-spring-fall and $42.7 \pm 20 \, \mathrm{g \, C \, g \, chl} \, a^{-1}$ in summer). The simulated annual and daily MPB PP rates were $118 \, \mathrm{g \, C \, m}^{-2} \, \mathrm{y}^{-1}$ and $320 \pm 219 \, \mathrm{mg \, C \, m}^{-2} \, \mathrm{d}^{-1}$, respectively.

### 3.3  *P. ulvae* dynamics

The MPB biomass simulated by the model was also constrained by the grazing pressure from the gastropod *P. ulvae*. The simulated density and biomass of *P. ulvae* increased in late spring with a first seasonal peak of ingestion in May 2 (Fig. 7c). The *P. ulvae* density reached a seasonal maximum in August 28 ($39\,950 \, \mathrm{ind \, m}^{-2}$) one months later than the seasonal maximum of *P. ulvae* biomass (July 22, $4.38 \, \mathrm{g \, C \, m}^{-2}$, Fig. 7ab). The simulated density and biomass of *P. ulvae* were compared to 2008

time coincident observations (Fig. 7ab). In February, 2008 the simulated density ($3308 \pm 8.4 \, \mathrm{ind \, m}^{-2}$) was significantly lower than the measured density ($5766 \pm 2985 \, \mathrm{ind \, m}^{-2}$; Mann Whitney test: p-value < 0.05). In July, 2008 an average density of $11851 \pm 3143 \, \mathrm{ind \, m}^{-2}$ was simulated by the model while a significantly higher average density of $17191 \pm 7084 \, \mathrm{ind \, m}^{-2}$ was measured (Mann Whitney test: p-value < 0.05). In February, 2008 the simulated biomass of *P. ulvae* was $381.28 \pm 4.5$ $\mathrm{mg \, C \, m}^{-2}$, which was significantly lower (Mann Whitney test: p-value < 0.05) than the observed biomass ($749.5 \pm 388$





$\mathrm{mg\,C\,m^{-2}}$). In July, 2008 the model underestimated biomass ($2809.3 \pm 700.4 \mathrm{\,mg\,C\,m^{-2}}$) whereas the measured biomass was $4469.8 \pm 1841.9 \mathrm{\,mg\,C\,m^{-2}}$ (Mann Whitney test: p-value < 0.05). The *P. ulvae* gross secondary production simulated by the model was $13.63 \mathrm{\,g\,C\,m^{-2}\,y^{-1}}$. Overall, the model reasonably captured the seasonal features depicted by the matchup observations.

### 3.4 Contribution of light, temperature and grazing to the MPB seasonal cycle

In the model, bottom-up (MST and downward irradiance) and top-down (grazing by *P. ulvae*) processes constrained the simulated MPB growth rate. Light and temperature limitation terms (see Eq. B7 and B8 in Appendix B2) varied between 0 and 1. At each time step, the lowest value was set as the most limiting term constraining the computation of the MPB photosynthetic rate. Over each daytime emersion period, the most limiting bottom-up factor was defined as the factor whose limitation was
the longest.

In phase 1, MST and light limited MPB growth 33.3 % and 66.7 % of the time, respectively, because PAR and simulated MST values were lower than the light saturation parameter ($E_k$, $100 \mathrm{\,W\,m^{-2}}$) and the temperature optimum for photosynthesis ($T_{opt}$, 18 °C), respectively (Table 2).

In phase 2, light was the most limiting factor (60.5%, Table 2), because the daytime duration was the longest over the year
(Fig. 8). The increasing daytime duration allowed MPB to grow on two daytime emersion periods at the beginning and at the end of the daytime period. As a result, the simulated MPB growth rate was limited by low light levels over a longer period than during phase 1 and phase 3 (Fig. 8). With respect to temperature, the MPB growth was more limited by MST in phase 2 (39.5 %) than in phase 1 (33.3 % Table 2). The high summer air temperature and solar irradiance heated the mud surface (Fig. 3 and 4), especially when daytime emersion periods occured in the middle of the day (10 AM - 16 PM) in spring tides (Fig. 8) with,
as a consequence, simulated MST higher in average than the MPB $T_{opt}$ value (Fig. 9).

In phase 3, the MPB growth rate was limited only by downward irradiance (100 %, Table 2). In fall, solar irradiance on daytime emersion periods decreased faster (slope = - 19.2 $\mathrm{W\,m^{-2}\,d^{-1}}$, p-value < 0.05, corresponding to a deviation from $E_k$ of -19 % $\mathrm{d^{-1}}$) than the MST (slope = -1.34 $\mathrm{°C\,d^{-1}}$, p-value < 0.05, corresponding to a deviation from $T_{opt}$ of - 7 % $\mathrm{d^{-1}}$).

Figure 9a shows the daily occurence of MPB limitation by the simulated MST over 2008. In phase 1, the simulated MST
increased towards $T_{opt}$ and, combined with increasing irradiance, led to a seasonal maximum of the mass-specific photosynthetic rate (Fig. 6c). In early May (phase 1), the mass-specific photosynthetic rate started to decrease due to thermo-inhibition as soon as the MST exceeded $T_{opt}$ (Fig. 6c and 9a), and the MPB biomass remained constant ($\sim 200 \mathrm{\,mg\,chl\,a\,m^{-2}}$) until the beginning of phase 2 (Fig. 9). In phase 2, the simulated MST was always higher than $T_{opt}$ when temperature limitation occured (Fig. 9a).

With respect to grazing, it was considered limiting for the MPB growth only when the simulated amount of biomass grazed by *P. ulvae* was higher than the simulated MPB biomass produced over the daytime emersion period (Fig. 9b). The top-down control by *P. ulvae* was mostly limiting in phase 2, when the ingested MPB biomass exceeded the MPB PP during 8.7% of the time (Fig. 9b). In phase 2, three high ingestion events were simulated and resulted into peaks of *P. ulvae* density and biomass in June 10, July 23 and August 29 (Fig. 7 and 9b). The simulated MST was higher in average than the temperature




optimum for grazing ($T_{opt_Z}$) all days when MPB were grazing-limited and triggered these three high ingestion events (Fig. 9b). The simulated peaks of ingestion rate varied between $\sim$ 180 and 334 $\mathrm{ng\,chl}\,a\,\mathrm{ind}^{-1}\,\mathrm{h}^{-1}$ over 2008 (Fig. 7c). 90 % of the grazing-limited days occured when the MST was already limited the MPB growth rate over 2008 (Fig. 9).

One month after the onset of MPB thermo-inhibition (May 2), a 6-day event of high *P. ulvae* ingestion ($> 10 \; \mathrm{mg\,C\,d}^{-1}$)
starting on June 8 (Fig. 7c) resulted into a sharp decrease of the simulated MPB biomass (Fig. 9b) and coincided with the beginning of phase 2. The MPB biomass grazed by *P. ulvae* during this synoptic event represented 5.35 $\mathrm{g\,C\,m}^{-2}$, which corresponded to a MPB biomass removal of 81 $\mathrm{mg\,chl}\,a\,\mathrm{m}^{-2}$. The combined effect of grazing and thermo-inhibition on the simulated MPB growth lasted 10 days in 2008 (2.7% of the year). It corresponded to a MPB biomass removal representing 14 % of the annual MPB PP (15.8 $\mathrm{g\,C\,m}^{-2}$; Fig. 9).

In the model, the occurrence of temperature or light limitation resulted from the coupling of the fortnightly tidal cycle with the seasonal solar irradiance and air temperature cycles. Over 2008, light was the most limiting factor because of low light levels in fall-winter and the occurrence of early and late daytime emersion periods during neap tides in spring-summer. During summer spring tides, the emersion periods in the middle of the day led to high simulated MST value ($> 22\ °C$), hence limiting the MPB growth rate and promoting the *P. ulvae* grazing activity. The combined effect of thermo-inhibition and grazing in
summer resulted into a marked depression of the simulated MPB biomass.

### 3.5 Annual MPB production sensitivity

A total of 10000 model runs (N) was performed where key biological parameters ($T_{opt}$, $T_{max}$, $T_{opt_Z}$, $\alpha_Z$, $E_k$ and $K_E$) were randomly selected within observed ranges (Table 3). The sensitivity analysis resulted in two kinds of model runs according to the sustainability of the MPB PP over the year. Model runs in which PP was sustained (SPP runs, PP > 40 $\mathrm{g\,C\,m}^{-2}\,\mathrm{y}^{-1}$, N =
1817) were distinguished from runs characterized by vanishing PP (VPP runs, PP $\leq$ 40 $\mathrm{g\,C\,m}^{-2}\,\mathrm{y}^{-1}$, N = 8183) according to a graphical representation of the annual PP as a function of the number of runs (Fig. 10).

In SPP runs, the annual production was negatively correlated with the MPB temperature optimum for photosynthesis ($T_{opt}$, Pearson's R = -0.24, p-value < 0.05) and positively correlated with the MPB temperature maximum ($T_{max}$, Pearson's R = 0.22, p-value < 0.05; Table 4). The correlation between PP and the MPB temperature amplitude ($T_{amp}$, the difference between $T_{opt}$
and $T_{max}$) was even higher than the correlation between PP and $T_{opt}$ and $T_{max}$ (Pearson's R = 0.39, p-value < 0.05; Table 4), suggesting that an increase of $T_{amp}$ (i.e. a decrease of $T_{opt}$ concomittant with an increase of $T_{max}$) led to an increase of the PP. The mean values $T_{amp}$, $T_{opt}$ and $T_{max}$ were 15 °C, 19 °C and 34 °C, respectively with relatively low variations of $T_{opt}$ and $T_{max}$ ($\sigma_{norm} \approx 0.14$) in SPP runs (Table 4). In terms of temperature, the use of this set of average values allows a sustainable PP in the model. In VPP runs, the mean value of $T_{amp}$ was 10.1 °C lower than in SPP runs, because the mean $T_{opt}$
value (29 °C) was higher than in SPP runs (19 °C). The maximum value of $T_{opt}$ was 10 °C higher in VPP runs than in SPP runs. The resulting wider range of $T_{opt}$ values led to higher variations in $T_{amp}$ in VPP runs ($\sigma_{norm} = 0.73$). However, SPP runs also showed a $T_{amp}$ minimum of 4.5 °C, which is $\sim$ 3-fold lower that the mean amplitude ($T_{amp}$) value (15 °C). Therefore, in addition to the temperature-related parameters, the light saturation parameter ($E_k$) played a role in the sustainability of PP over the year.





While a small positive correlation was evidenced between $E_k$ and PP in VPP runs (Pearson's R = 0.02, p-value < 0.05), $E_k$ was negatively correlated with PP in SPP runs to a larger extent (Pearson's R = -0.56, p-value < 0.05). In SPP runs, the mean value of $E_k$ (77 W m$^{-2}$) was lower than in VPP runs (95 W m$^{-2}$). However, $E_k$ variations were comparable (0.54 < $\sigma_{norm}$ < 0.64) and the minimum (2.5 W m$^{-2}$) and maximum values (180 W m$^{-2}$) were same in both the SPP and VPP runs.

Consequently, annual PP is less sensitive to variations of $E_k$ than to variations of $T_{opt}$ and $T_{max}$. Nevertheless, in SPP runs, a low value of $E_k$ could sustain PP if $T_{amp}$ was lower than 15 °C. The half-saturation constant for light use ($K_E$) was not correlated with annual PP in SPP runs (Pearson's R = 0.01, p-value > 0.05). $K_E$ was slightly but negatively correlated with annual PP in VPP runs (Pearson's R = -0.04, p-value < 0.05). In VPP runs, low values of $K_E$ slightly increased the annual PP.

In SPP runs, the temperature optimum for grazing by *P. ulvae* ($T_{opt_Z}$) showed a low but significant correlation with PP

(Pearson's R = 0.16, p-value < 0.05) suggesting that high $T_{opt_Z}$ values resulted in high levels of annual PP. The shape parameter of the temperature grazing function ($\alpha_Z$) was negatively correlated with PP in SPP runs (Pearson's R = -0.11, p-value < 0.05). However, $T_{opt_Z}$ and $\alpha_Z$ variations were high and of the same extent ($\sigma_{norm}$ = 0.19 and $\sigma_{norm} \approx 0.6$, respectively) and their mean, maximum and minimum values were very similar in both SPP and VPP runs (Table 4). Overall, the model was therefore sensitive to MPB temperature parameters and to the light saturation parameter. A specific set of these temperature and light

related parameters allowed for a sustainable level of MPB production and biomass, which resulted into a significant effect of grazing on the MPB annual production.

## 4 Discussion

### 4.1 The MPB seasonal cycle

Our study suggests a MPB seasonal cycle on the Brouage mudflat characterized by three phases in 2008, i.e. a bloom in

winter/spring, low biomass levels in summer, and a peak of moderate intensity in fall. Cariou-Le Gall and Blanchard (1995) sampled monthly from March 1992 to February 1993 the MPB Chl *a* concentration within the top 0.5 cm sediment on the Brouage mudflat. Their measurements suggest a bloom in winter/spring and low Chl *a* concentrations in summer, which is consistent with the 2008 NDVI data and MPB biomass simulated by the model. Cariou-Le Gall and Blanchard (1995) did not report any peak of MPB biomass in fall, which may be modulated by the interannual variability driven by the meteorological

conditions. In Northern (De Jong and de Jonge, 1995; Sahan et al., 2007) and Southern (Brito et al., 2013) european mudflats, MPB spring blooms are also observed. However, the contribution of underlying abiotic (e.g. air temperature, irradiance, rain, wind) and biotic (e.g. autotrophic species community, predators) factors are likely to be different in shaping the seasonal MPB cycle at such contrasted latitudes.

In the Brouage mudflat, the seasonal cycle of MPB at the sediment surface compares to that depicted by the remotely

sensed NDVI data. The simulated MPB biomass in the biofilm and its instantaneous PP are close to maximum values of biomass previously measured in biofilms developing at the surface of very fine sediments of the Brouage mudflat (24 ± 5 mg chl *a* m$^{-2}$; Herlory et al., 2004). Also, the mean time spent by MPB cells at the surface in the model (36.6 ± 12 min) is consistent with the estimate reported by Blanchard et al. (60 min; 2004). Blanchard et al. (2004) relate the mean time spent




by cells at the surface to the time at which the photosynthetic capacity at saturating light levels of epipelic diatoms sampled on the same mudflat started to decrease (Blanchard et al., 2004). In our study, the mean time spent by cells at the surface also fluctuates according the total MPB biomass, leading to an inverse relationship with the total MPB biomass. Once at the surface, the simulated MPB growth is regulated by the mass-specific photosynthetic rate in $\mu g\,C\,(\mu g\,chl\,a)^{-1}\,h^{-1}$ converted

into a growth rate ($h^{-1}$) using a variable C:Chl $a$ ratio. The resulting MPB growth rates simulated by the model ($0.24 \pm 0.07$ $d^{-1}$ with a range of values between 0.04 $d^{-1}$ and 0.43 $d^{-1}$) were consistent with observations made on epipelic diatoms (0.035 - 0.86 $d^{-1}$; Gould and Gallagher, 1990; Underwood and Smith, 1998; Scholz and Liebezeit, 2012). With respect to the simulated C:Chl $a$ ratio, it varies in a range of values between 16 and 75.5 $g\,C\,g\,chl\,a^{-1}$ within the range of observed values in mudflats (18.7 - 80 $g\,C\,g\,chl\,a^{-1}$; Guarini, 1998; Gould and Gallagher, 1990; de Jonge et al., 2012).

Contrary to Chl $a$ measurements, there were no PP measurements made in 2008 on the Brouage mudflat. For comparison, we use averages of mass-specific photosynthetic rates computed from previous measurements at different locations on the Brouage mudflat for different years (using $CO_2$ fluxes data measured in benthic chambers). Despite the year-to-year variability, the mean mass-specific photosynthetic rates simulated by the model during spring tides ($0.61 \pm 0.08$ $mg\,C\,(mg\,chl\,a)^{-1}\,h^{-1}$ in April, $0.64 \pm 0.12$ $mg\,C\,(mg\,chl\,a)^{-1}\,h^{-1}$ in May and $0.43 \pm 0.03$ $mg\,C\,(mg\,chl\,a)^{-1}\,h^{-1}$ in July) were in the range of measure-

ments for the same months ($1.6 \pm 1.1$ $mg\,C\,(mg\,chl\,a)^{-1}\,h^{-1}$ in April 2012, $0.28 \pm 0.11$ $mg\,C\,(mg\,chl\,a)^{-1}\,h^{-1}$ in May 2015 and $0.32 \pm 0.13$ $mg\,C\,(mg\,chl\,a)^{-1}\,h^{-1}$ in July 2015; pers.comm. from J. Lavaud). Morevover, simulated daily and yearly PP rates compared to measurements made across other European intertidal mudflats. The simulated annual ($118$ $g\,C\,m^{-2}\,y^{-1}$) and daily mean ($320 \pm 219$ $mg\,C\,m^{-2}\,d^{-1}$) PP rates were in the range of reported estimates ($140 \pm 82$ $g\,C\,m^{-2}\,y^{-1}$, N = 18 and $690 \pm 682$ $mg\,C\,m^{-2}\,d^{-1}$, N = 9, respectively; Underwood and Kromkamp, 1999). The model-data comparison suggests

that the model can resolve with confidence the main patterns of the MPB seasonal cycle.

The relative contribution of light, MST and grazing to the simulated MPB seasonal cycle resulted from the coupling of the fortnightly tidal cycle and seasonal solar irradiance and air temperature cycles. Such a coupling is reported in intertidal sediments in the Tagus estuary, Portugal (Serodio and Catarino, 1999). In the model, an emersion period takes place in the middle of the day during spring tides exposing the mud surface to a daily solar irradiance and temperature maximum. In

summer, when the seasonal maximum of daily solar irradiance and temperature is reached, the high simulated MST values translate into an enhanced thermo-inhibition of MPB growth and *P. ulvae* grazing pressure. The highest MPB thermo-inhibition in summer spring tides was also highlighted by Guarini et al. (1997) in the Brouage mudflat. During neap tides, light limits the MPB growth when emersion periods occur early in the morning and late in the afternoon at low daily light levels. The reduced PP of MPB at low light levels and MST values during neap tides compared to spring tides was also observed by Kwon et al.

(2014) on the Hwaseong mudflat, South Korea.

In the model, we do not consider any MPB limitation by inorganic nutrients. In the Brouage mudflat, Feuillet-Girard et al. (1997) highlighted the greater affinity of MPB to ammonium compared to nitrate. They observed a higher ammonium availability released from the sediment during summer, making unlikely the nutrient limitation responsible of the summer depression of MPB biomass. The high nutrient availability in summer can be attributed to faunal activities (bioturbation, bio-irrigation,

excretion; Feuillet-Girard et al., 1997; Heilskov et al., 2006; Laverock et al., 2011; Rakotomalala et al., submitted).





## 4.2 Role of mud surface temperature on the MPB and *P. ulvae* activity

On the Brouage mudflat, the simulated MST plays a major role in the MPB seasonal cycle. In spring, the simulated MST increases towards the MPB temperature optimum for photosynthesis ($T_{opt}$). Along with increasing light levels, it contributes to increase the mass-specific photosynthetic rate and triggers the onset of the MPB spring bloom. As soon as the simulated MST exceeds the MPB temperature optimum for photosynthesis, the MPB PP starts to decrease due to thermo-inhibition, particularly during spring tides. Because of the heat inertia of the surface sediment, the simulated MST decreases in fall slower than the solar irradiance. As a consequence, the simulated MST departs slower from the temperature optimum for photosynthesis than does the downward irradiance from the light saturation parameter ($E_k$). Despite decreasing solar irradiance in fall, the simulated MPB PP increases until November, when the simulated MPB growth rate is limited by too low light levels and MST values with respect to the MPB light saturation parameter (100 W m$^{-2}$) and temperature optimum for photosynthesis (18 °C), respectively.

Using the Production-Temperature (P-T) model from Blanchard et al. (1996), Blanchard et al. (1997) and Guarini et al. (2006) also suggested that the MPB PP was temperature-limited in summer on the Brouage mudflat. On a southern intertidal mudflat (Tagus Estuary, Portugal), Brito et al. (2013) suggested that thermo-inhibition was responsible for the summer MPB depression observed in NDVI times series in conditions of high sediment temperature (30 °C). In addition, the detrimental effect of MST ranging between 18 °C and 24 °C was shown in microcosms using fluorescence (Cartaxana et al., 2015).

In the model, the production is related to temperature according the P-T relationship of Blanchard et al. (1996). As a result, the occurrence and intensity of MPB thermo-inhibition depends on the MPB temperature optimum and maximum for photosynthesis used in the relationship. The set of parameters determines the thermal threshold and interval at which thermo-inhibition occurs. The sensitivity analysis shows that the annual PP is very sensitive to the temperature amplitude between the two parameters. The annual PP increases as the amplitude increases. On the Brouage mudflat, the MPB temperature optimum and maximum for photosynthesis were estimated to 25 °C and 38 °C, respectively, and assumed to be constant over the year (Blanchard et al., 1997). In our study, a lower MPB temperature optimum for photosynthesis value of 18 °C is required to simulate a spring bloom that compares to the NDVI time series. Such a temperature optimum also implies a more rapid onset and a higher MPB thermo-inhibition as the simulated MST increases in summer. Values of both MPB temperature optimum and maximum for photosynthesis are reported to vary by up to 10 °C (Table 5). To that respect, the MPB temperature optimum for photosynthesis is a key parameter in the model, because it constrains the onset of the MPB spring bloom and the thermo-inhibition span and intensity.

In addition, the strong heating and wind exposure of the mud surface is accompanied by pore water evaporation that results into desiccation and increased salinity (Coelho et al., 2009). A decrease of pore water content can induce even more detrimental effects within the cells through production of reactive oxygen species (Rijstenbil, 2003; Roncarati et al., 2008) potentially leading to the oxydation of the photosynthetic unit (Nishiyama et al., 2006). The motility of epipelic diatoms is supposed to be a strategy to avoid harmful conditions at the surface of cohesive sediments (Admiraal, 1984). However, Juneau et al. (2015) showed no significant negative effect of salt stress on the photosynthesis of immobile epipelic diatoms. Coelho et al.





(2009) highlighted the role of the rate of pore water content decrease in the field. While slow desiccation (reduction by 40% of the pore water content in 4.5 h) had no significant negative effect on the photosynthesis of microphytobenthic cells within the biofilm, fast desiccation (reduction by 40% of the pore water content in 2 h) resulted in desiccation and decreased the photosynthetic activity of MPB (Coelho et al., 2009). The speed of physiological adaptations by epipelic diatoms to cope with

salt stress could be overtaken by the rate of de-watering (Coelho et al., 2009). In addition to micro-migrations, epipelic diatoms produce extracellular polymeric substances (EPS) to temper the effect of desiccation and high salinity (Sauer et al., 2002). High sediment temperature (> 35 °C) is also known to reduce the motility of MPB diatoms and so their capacity to avoid harmful conditions at the sediment surface (Cohn et al., 2003; Laviale et al., 2015). To that respect, the use in the model of a MPB temperature optimum for photosynthesis lower than values reported in the litterature (Table 5) may overestimate the

thermo-inhibition process and, as such, promote low PP rates. Nevertheless, this negative feedback on PP may compensate the absence in the model of detrimental effects of desiccation on the microphytobenthic cells, which is a positive feedback on PP.

The simulated MST also rules the ingestion rate of MPB by the grazer *P. ulvae* in the model. The grazing parameters play an important role in the regulation of MPB annual production. Simulated PP rates increase as the value of the optimal temperature for grazing increases because the grazing optimum is not often reached in the model. Meanwhile, simulated PP rates decrease

with augmentation of the shape parameter of the temperature-related grazing ($\alpha_Z$) because of the increase of the grazing exponential terms resulting into a higher simulated grazing pressure. In the model, the ingestion rate increases when the MST exceeds the optimal temperature for grazing (fixed at 20 °C; Pascual and Drake, 2008). A high metabolism of benthic grazers promoted by high temperature conditions (up to 22 °C) and the resulting increase of the grazing pressure on benthic diatoms was observed by Sahan et al. (2007) on a mudflat in Netherlands.

In the model, we report 49 days in 2008 during which the simulated MST daily averaged over the emersion periods exceeded the optimal temperature for grazing ($T_{opt_Z}$, 20 °C). However, the simulated grazing pressure results into a decrease of the MPB PP for only 11 days. The remaining 38 days, grazing did not exceed the MPB PP. Such a behavior of the model highlights the key role played by the MST in simulating the MPB and *P. ulvae* growth rates. It also suggests that the seasonal MPB dynamics is likely to be strongly impacted by the interannual variability of the meteorological conditions (de Jonge et al., 2012; Benyoucef

et al., 2014), which regulates the simulated bottom-up (i.e. MPB thermo-regulation) and top-down (i.e. grazing) processes.

### 4.3 Effect of light on MPB photosynthesis

In the model, light is the most limiting factor throughout the year. The low irradiance during fall and winter limits the MPB photosynthesis as the irradiance is in average lower than the light saturation parameter ($E_k$). In spring, the increasing irradiance and MST translate into higher mass-specific photosynthetic rates than in fall-winter leading to the onset of the simulated MPB

spring bloom. In summer, photo-inhibition is not accounted for in the model as the simulated mean time spent by a MPB cells at the surface is lower than the time required to induce photo-inhibition at saturating light levels (Blanchard et al., 2004). As a consequence, light limits the simulated MPB growth only during neap tides, when the sediment emersion occurs at low light levels early and late in the day.



Photosynthesis is represented in the model by the Production-Irradiance (P-E) model of Platt and Jassby (1976). It relies on the photosynthetic capacity ($P_{max}^b$), the light saturation parameter ($E_k$) and the maximum light utilization coefficient ($\alpha = \frac{P_{max}^b}{E_k}$, Talling, 1957). Irradiance has no influence on the photosynthetic capacity and maximum light utilization coefficient (MacIntyre et al., 2002) in our study. Based on the work of (Blanchard et al., 1996), the photosynthetic capacity and maximum

light utilization coefficient vary in the model with the simulated MST. Therefore, the seasonal adjustment of photosynthesis to irradiance depends mainly on the photoacclimation status of MPB cells, which can be related to the light saturation parameter (Sakshaug et al., 1997). The light saturation parameter corresponds to the irradiance at which photosynthesis switches from light reactions (light absorption and photochemical energy conversion) to dark reactions (reductant utilization) (Sakshaug et al., 1997). It has been reported to vary seasonally in benthic microalgae (Blanchard and Cariou-Le Gall, 1994; Barranguet et al.,

1998; Light and Beardall, 2001; Pniewski et al., 2015; Barnett et al., 2015). Cells increase their light saturation parameter at high irradiance (summer) and reduce it with decreasing light levels (Sakshaug et al., 1997). In our study, as the light saturation parameter is set as constant throughout the year ($100 \, \mathrm{W\,m^{-2}}$), photoacclimation is simulated by the way of a variable C:Chl $a$ ratio.

During winter, low light acclimated cells have a lower C:Chl $a$ ratio due to an increase of the Chl $a$ content (MacIntyre et al.,

2002; Brunet et al., 2011). In summer, with the increasing irradiance and day length, high light-acclimated cells reduce their Chl $a$ content leading to a higher C:Chl $a$ ratio (MacIntyre et al., 2002; Brunet et al., 2011). In the model, solar irradiance shapes the simulated C:Chl a ratio (Eq. B9 in Appendix B2). The C:Chl $a$ ratio reaches a seasonal maximum value ($75.5 \, \mathrm{g\,C\,g\,chl}\,a^{-1}$) in summer when solar irradiance is the highest. Such a result is consistent with estimate ($80 \, \mathrm{g\,C\,g\,chl}\,a^{-1}$) reported in summer by de Jonge et al. (2012). In the model, given that the mass-specific photosynthetic rate ($\mathrm{\mu g\,C\,(\mu g\,chl}\,a)^{-1}\,\mathrm{h^{-1}}$) and the C:Chl

$a$ ratio are related to the growth rate ($\mathrm{h^{-1}}$), the growth rate increases as the C:Chl $a$ ratio decreases (low light acclimated cells). The seasonal variation of the simulated growth rate results from the combination of the variation of the photosynthetic capacity and maximum light utilization coefficient driven by the simulated MST and the variation of the C:Chl $a$ ratio with irradiance.

Finally, photoinhibition at high irradiance is not accounted for in the P-E model of Platt and Jassby (1976) used in the model. Epipelic diatoms achieve "micro-migrations" within the sediment to avoid harmful light conditions prevailing at the sediment

surface (Kromkamp et al., 1998; Perkins et al., 2001; Cartaxana et al., 2011). However, combined with high temperature conditions ($> 35 \, °\mathrm{C}$) at the sediment surface potentially leading to reduced cell motility (Cohn et al., 2003), epipelic diatoms can be photoinhibited (Laviale et al., 2015). In temperate intertidal mudflats, high light and temperature conditions occur during summer and their combined effect on MPB photosynthetic rate may explain the depression of MPB biomass observed in summer.

**4.4   Top-down regulation of MPB dynamics**

Grazing by meio- and macrobenthos is often suggested as the main driver of the MPB biomass depression observed in summer on intertidal mudflats (Cadée and Hegeman, 1974; Cariou-Le Gall and Blanchard, 1995; Sahan et al., 2007; Orvain et al., 2014). Weerman et al. (2011) showed experimentally a strong decrease of MPB biomass in the presence of macrofauna driven by direct grazing and by the absence of surface mud stabilisation due to bioturbation by deposit feeders.




In the model, *P. ulvae* grazing exceeds the MPB PP mainly in summer. The simulated grazing pressure corresponds to 6 synoptic events (between 1 and 4 days) of MPB biomass removal when the MST is higher than the temperature optimum for grazing. 90 % of these events occurs when the simulated MPB growth is already temperature-limited as the temperature optimum for grazing is higher than the temperature optimum for photosynthesis. Consequently, the grazing pressure by *P. ulvae*

adds to the effect of thermo-inhibition. Throughout 2008, such a combined effect takes place 2.7 % of the year. The simulated gain terms promoting the growth rate of MPB limited by thermo-inhibition do not compensate the loss terms dominated by the grazing pressure, which leads to a decrease of the MPB biomass. In a schematic model, Thompson et al. (2000) showed such a seasonal uncoupling between the grazing intensity by intertidal grazing molluscs and the microalgae abundance from observations made on a rocky shore of the Isle of Mann (United Kingdom). The authors conceptualized the role played by the

light and temperature stress on the microalgae productivity and by the temperature-promoted grazing in the depression of the microalgal standing stocks in summer.

The combined effect of grazing and thermo-inhibition translates into 14 % (15.8 g C) of the simulated annual MPB PP channeled towards the *P. ulvae* and secondary production. The simulated gross *P. ulvae* secondary production is 13.63 g C m$^{-2}$ y$^{-1}$, which represents 11.5 % of the simulated annual MPB PP (118 g C m$^{-2}$ y$^{-1}$). This fraction of PP transferred to *P. ulvae* sec-

ondary production is consistent with the average fraction reported by Asmus and Asmus (12.5 %; 1985) on intertidal sand bottom communities of the Island of Sylt in the North Sea.

In July, the simulated density of *P. ulvae* lies in the lower range of time-coincident measurements. As the simulated MST fairly agrees with time-coincident measurements, other factors may explain the likely underestimation by the model of the density and ingestion of *P. ulvae*. First, there may be a bias resulting from the monthly mean weight estimates used to simulate

the *P. ulvae* density (see Appendix B3). The monthly mean weights are based on samples gathered in 2014-2015 on the Aiguillon mudflat, in the vicinity of the Brouage mudflat (Fig. 1). Nevertheless, the seasonality of the *P. ulvae* density is similar on the two mudflats with a peak of density in late summer (Haubois et al., 2002), which suggests that such a bias is likely limited. In addition, apart from the four events of high ingestion, the simulated ingestion rates (0 - 50 ng chl $a$ ind$^{-1}$ h$^{-1}$) are consistent with ingestion rates measured in experiments with *P. ulvae* and benthic diatoms collected in our study area and

performed at temperature close to the optimal temperature for grazing in the model (15 - 20 °C; 0.75 - 52 ng chl $a$ ind$^{-1}$ h$^{-1}$; Blanchard et al., 2000; Haubois et al., 2005; Pascal et al., 2008). Concerning simulated peaks of ingestion, the maximum ingestion rate (334 ng chl $a$ ind$^{-1}$ h$^{-1}$) simulated during the peak of July 22 is coherent with the maximum ingestion rate (385 ng chl $a$ ind$^{-1}$ h$^{-1}$) experimentally measured by Coelho et al. (2011) in Portugal. Second, the wave- and tidal-induced shear stress on the bottom sediment may transport horizontally *P. ulvae* individuals across the mudflat. Such a process is not

accounted for in the model and may lead to an underestimation of the *P. ulvae* biomass and density. Finally, potential MPB grazing by fauna other than *P. ulvae* is represented in a simple way by a linear and generic loss term in the model whereas it might be a non-linear process that can vary seasonally (Pinckney et al., 2003). This closure term may be underestimated in the model.

With respect to meiofauna, Pinckney et al. (2003) suggested a more intense grazing by meiofauna in summer than in winter in

the Terrebonne Bay estuary (USA). Admiraal et al. (1983) estimated the meiofauna grazing at 300 mg C m$^{-2}$ d$^{-1}$ on a mudflat




of the Ems Dollard estuary (Netherlands). Comparable rates of meiofauna ingestion (58 - 189 $\mathrm{mg\,C\,m^{-2}\,d^{-1}}$) are reported for the Brouage mudflat (Montagna et al., 1995). Admiraal et al. (1983) concluded to a non significant effect of meiofauna grazing relative to the MPB production rates they measured. Nevertheless, their estimated grazing rate exceeds our simulated daily MPB production rates for 26 days in summer i.e. 29% of the time of the second phase in the model, suggesting that meiofauna

grazing could impact MPB. In addition, Pascal et al. (2008) compared ingestion rates by *P. ulvae* and a nematode community from the Brouage mudflat in experimental conditions. According to the abundance of organisms selected for the experiment of Pascal et al. (2008) and a constant C:Chl *a* ratio of 45 $\mathrm{g\,C\,g\,chl}\,a^{-1}$ (Guarini, 1998), the amount of Chl *a* ingested by nematodes per hour was only 1.5 % of the Chl *a* ingested by *P. ulvae* per hour in their experiment. However, in regard to the observed abundances on the field and without density-dependant effect on grazing rates, this theoritical amount of Chl *a* ingested by

nematodes increases to almost 50% of the Chl *a* ingested by *P. ulvae* in the study of Pascal et al. (2008). According to the measured biomass uptake by meiofauna (Montagna et al., 1995) and nematodes (Pascal et al., 2008) for the Brouage mudflat, an explicit representation of meiofauna ingestion in the model might magnify the simulated depletion of MPB biomass in summer months. The representation of grazing in the model can be improved. Nevertheless, the fair agreement between the simulated *P. ulvae* densities and biomass levels with time-limited but time-coincident observations suggests that overall the

model simulates with some confidence the grazing pressure on MPB.

### 4.5  Physical setting of the coupled model

The predictive ability of the physical-biological coupled model depends on the accuracy of the oceanic and meteorological forcings. The frequency of the water height and meteorological time series used to constrain the model is hourly while the model time step is 06 minutes. The lower frequency of the model forcings over a day partly explains the model-data discrepancies.

In addition, the weather station where meteorological data were acquired is located 30 $\mathrm{km}$ away from the Brouage study site. Local weather conditions may differ between the two sites, especially the global irradiance and wind speed used to simulate the MST and MPB growth rate. Global irradiance can be impacted by local cloud cover and the wind regime can be different due to local thermal winds. In the model, the timing of the emersion-immersion cycle is constrained by the observed water heights and bathymetric level. The bathymetric level used to compute the water height above the Brouage study site originates from a

digital elevation model with a 1-m horizontal resolution and a 15-cm vertical precision. Even if the Brouage mudflat is relatively flat (1:1000), ridges and runnels are present near the study site (Gouleau et al., 2000) and the topography is highly variable at a meter scale. Inaccuracies in the bathymetric level relative to the study site may translate into model-data discrepancies in terms of timing of the emersion-immersion cycle in the model. Given that the mud temperature model is constrained by the water height and meteorological data, it is sensitive to possible inaccuracies in the forcings that may impact the simulated

hourly dynamics of MPB and *P. ulvae*. Nevertheless, at the seasonal scale, the impact on the biological compartments of such inaccuracies in the forcings may be limited.



## 5 Conclusion and perspectives

This study is a first attempt to simulate the MPB seasonal cycle observed on a temperate intertidal mudflat and to quantify the relative contribution of both biotic and abiotic factors on the seasonal MPB dynamics. The physical-biological coupled model fairly compares to time-coincident remotely sensed and *in situ* data and provides key findings about the seasonality of MPB on

the Brouage mudflat (French Atlantic coast):

   – The 2008 MPB seasonal cycle consists in 3 phases: a spring bloom, a summer depression of the biomass levels, and a moderate peak of biomass in fall;

   – In winter and early spring, the seasonal mass-specific maximum photosynthetic rate mainly driven by the simulated MST and the seasonal low C:Chl *a* ratios lead to a seasonal maximum of MPB growth rate and to a MPB spring bloom;

– Thermo-inhibition (39.5 % of summer) and *P. ulvae* grazing (8.7 % of summer) are responsible for the MPB biomass depression occurring in summer. Grazing exceeds the MPB PP 90 % of the time when the MPB growth is already temperature-limited;

   – The combined effect of grazing and thermo-inhibition results into 14 % of the simulated annual MPB primary production channeled towards the *P. ulvae* secondary production through ingestion;

– The model is sensitive to MPB temperature parameters (temperature optimum and maximum for photosynthesis), to the MPB light saturation parameter and, to a lesser extent, to grazing parameters (the optimal temperature for grazing and the shape parameter of the temperature-related grazing function).

The seasonal MPB dynamics simulated by the model compares to time coincident times series of remotely sensed NDVI data hence providing a qualitative assessement of the model predictive ability. A next step would be to extend such a model-

satellite data comparison to a more quantitative assessment to validate the simulated levels of MPB Chl *a* concentration and PP. The recent advance of multispectral and hyperspectral remote sensing allows for the development of new algorithms to retrieve products of ecological interest for MPB. Brito et al. (2013) developed local empirical relationships relating synchronized NDVI data to *in situ* Chl *a* concentrations to retrieve from space estimates of Chl *a* concentration on a Portuguese intertidal mudflat. Efforts are also focused in using remote sensing reflectances from airbone hyperspectral data to assess MPB PP rates (Méléder

et al., 2018). Recently, and in the light of the work of Brito et al. (2013), Daggers et al. (2018) combined biomass derived from NDVI data with simulated photosynthetic capacity from environmental conditions (irradiance and air temperature) to map MPB PP on intertidal mudflats in Netherlands. Other promising methods in the estimation of PP in intertidal mudflat at the ecosystem scale are the non-invasive atmospheric and aquatic Eddy Covariance (EC) techniques. The atmospheric EC provides continuous and direct $CO_2$ flux measurements at the air-water and air-sediment interfaces during high and low tides,

respectively, across different time scales from hours to years (Baldocchi et al., 1988; Aubinet, Marc and Grelle, Achim and Ibrom, Andreas and Rannik, Üllar and Moncrieff, John and Foken, Thomas and Kowalski, Andy S and Martin, Philippe H and Berbigier, Paul and Bernhofer, Ch and others, 1999; Zemmelink et al., 2009; Polsenaere et al., 2012). Similarly, the aquatic



EC measures benthic $O_2$ fluxes at the sediment/water interface (Berg et al., 2003). Quantifying the MPB PP and biomass on intertidal mudflats is a prerequisite for further estimating the flux of biogenic carbon from the benthos to the pelagos. During the immersion period, MPB can be resuspended (9.7 $\mathrm{mg\,C}$ per high tide, i.e 3 % of the mean simulated production during low tides, Dupuy et al., 2014) and highly disturb the functionning of the benthic-pelagic ecosystem (Saint-Béat et al., 2014).

The study of air-water and sediment-water exchanges through simultaneous atmospheric and aquatic EC measurements could allow quantifying the importance of metabolic fluxes during immersion and emersion periods but also the coupled processes between the benthic and pelagic compartments such as MPB resuspension. Microphythobenthic community resuspension can significantly contribute to planktonic gross PP and, in turn, explain lower $CO_2$ fluxes from the water column to the atmosphere at high tide during the day than at night (Guarini et al., 2008; Polsenaere et al., 2012). To date, the modelling effort put on

the physically-driven (tides and waves) resuspension processes of MPB is still limited (see Mariotti and Fagherazzi, 2012). Accounting in models for sediment bottom shear stress mediated by hydrological forcings (current and waves) along with bioturbation processes could lead to more realistic predictions of the interannual MPB dynamics. Such a representation of the biologically and physically-driven benthic-pelagic interactions would be fully apprehended by the coupling of biological MPB models to high resolution ocean models. Such an approach would open the door to an accurate assessement of the vertical

and horizontal export flux of biogenic matter at the land-ocean interface and, more generally, of the contribution of productive biofilms in mudflats in the carbon cycle of the global coastal ocean.

*Code availability.*

*Data availability.*

*Code and data availability.*

**Appendix A: Mud temperature model**

The original version of the mud temperature model of Guarini et al. (1997) is simplified by only resolving the mud surface temperature $T_M(z_0, t)$ (K) which is governed by the following equation:

$$\rho_M C_{P_M} \frac{\partial T_M(z_0, t)}{\partial t} = f(T_M(z_0, t)), \tag{A1}$$

where $f(T_M(z_0, t))$ is the heat energy balance (HEB, $\mathrm{W\,m^{-2}}$) at the sediment surface $z_0$ (m) at time $t$ (s). This sediment

surface layer is 1-cm deep. The temperature (K) is assumed to be homogenous within the layer and is governed by the HEB (Harrison and Phizacklea, 1987; Piccolo et al., 1993). $\rho_M$ is the volumetric mass of mud ($\mathrm{kg\,m^{-3}}$). It is the sum of the water fraction and of the dry sediment fraction ($\rho_M = \rho_W \xi + \rho_S(1 - \xi)$ where $\rho_W$ and $\xi$ are the water volumetric mass ($\mathrm{kg\,m^{-3}}$)





and the porosity (%), respectively. $C_{P_M}$ is the specific heat capacity of mud at constant pressure ($\mathrm{J\,kg^{-1}\,K^{-1}}$):

$$C_{P_M} = \frac{\eta}{\mu\rho M}, \tag{A2}$$

where $\eta$ is the heat conductivity ($\mathrm{W\,m^{-1}\,K^{-1}}$) and $\mu$ the thermal diffusivity ($\mathrm{m^2\,s^{-1}}$). Heat exchange fluxes at the sediment interface are different according to the emersion-immersion cycle. During low tide, the HEB is governed by downward fluxes

of radiation from the sun ($R_S$, $\mathrm{W\,m^{-2}}$) and from the atmosphere ($R_{Atm}$, $\mathrm{W\,m^{-2}}$), by upward fluxes of radiation from the receiving surface ($R_M$, $\mathrm{W\,m^{-2}}$), by sensible heat fluxes by conduction due to mud-air temperature difference ($S_{Mud\to Air}$, $\mathrm{W\,m^{-2}}$) and by flux of evaporation ($V_M$, $\mathrm{W\,m^{-2}}$):

$$f\left(T_M(z_0,t)\right) = R_S + R_{Atm} - R_M - S_{Mud\to Air} - V_M \text{ with } V_M = \xi V_W, \tag{A3}$$

where $\xi$ is the mud porosity ($\xi \in [0,1]$, %) and $V_W$ the evaporative heat flux of seawater ($\mathrm{W\,m^{-2}}$). Details about formulas and

constants computation of each fluxes during emersion are given in Tables A1 and A2. During immersion, only a sensible heat flux from seawater to mud is taken into account because the overlying turbid seawater is assumed to limit the penetration of the downward irradiance (Harrison and Phizacklea, 1985; Losordo and Piedrahita, 1991). The sensible heat flux consists in the product of the conductivity and with a finite-difference approximation of the temperature gradient between mud and the overlying seawater (Losordo and Piedrahita, 1991):

$$f\left(T_M(z_0,t)\right) = S_{Mud\to Water} = -\frac{\eta}{h_W}\left(T_M\left(z_0,t\right) - T_W(t)\right) \text{ with } h_W = 0.2H, \tag{A4}$$

where $h_W$ corresponds to the the overlying water mixing height (m), a fraction of the total height ($H$ in m). The seawater temperature ($T_W$) is computed according to the heat exchange through heat conduction within the surface mixed layer ($z_{\alpha_{MLD}}$):

$$T_W(t) = \alpha_{MLD} T_W(z_{\alpha_{MLD}}, t) + (1 - \alpha_{MLD}) T_W(t) \text{ with } \alpha_{MLD} = 0.12\left(1 + \frac{U}{3}\right) \tag{A5}$$

where $T_W(z_{\alpha_{MLD}}, t)$ is the water temperature in the surface mixed layer (K). $\alpha_{MLD}$ is the depth of the mixed layer (m) according the wind speed $U$ ($\mathrm{m\,s^{-1}}$). The surface water temperature is also governed by the HEB at the water surface:

$$\rho_W C_{P_W} \frac{\partial T_W(z_{\alpha_{MLD}}, t)}{\partial t} = f\left(T_W(z_{\alpha_{MLD}}, t)\right), \tag{A6}$$

with $f\left(T_W(z_{\alpha_{MLD}}, t)\right) = R_S + R_{Atm} - R_W - S_{Air\to Water}, \tag{A7}$

where $\rho_W$ is the volumetric mass of water ($\mathrm{kg\,m^{-3}}$). $C_{P_W}$ is the specific heat capacity of seawater at constant pressure

($\mathrm{J\,kg^{-1}\,K^{-1}}$). The term $S_{Air\to Water}$ is the sensible heat flux mediated by thermal conduction due to water-air temperature difference. $R_W$ ($\mathrm{W\,m^{-2}}$) is the seawater upward radiation. $T_W$ (K) is initiliazed by the following equation:

$$T_W(t) = 17 + 5cos\left(2\pi \frac{day - 230}{year\ length}\right) + 273.15 \tag{A8}$$

where $day$ is the day of the year and the year length is in days. Details on parameters and constants are given in Tables A1 and

A2.



## Appendix B: Biological model

### B1 MPB migration scheme

A system of three partial differential equations describes the temporal dynamics of the MPB biomass within the surface biofilm ($S$), MPB biomass within the 1ˢᵗ cm of sediment ($F$), and biomass of MPB grazer *P. ulvae* ($Z$). The system drives the MPB migration scheme according to the diurnal and tidal cycles that constrain the biological-physical coupled model (Table 1). During the daytime emersion period:

$$\text{if } \tau > 0 \begin{cases} \frac{dS}{dt} = \left(r_F F + P^b S\right)\left(1 - \frac{S}{S_{max}}\right) - m_S S - min\left[IR\left(\frac{Z}{W_Z^{mean}}\right), S_{IR}\right] \times H\left(F, F_{mini}\right) \\ \frac{dF}{dt} = -r_F F\left(1 - \frac{S}{S_{max}}\right) + P^b S\left(\frac{S}{S_{max}}\right) - m_F F \\ \frac{dZ}{dt} = \gamma \times min\left[IR\left(\frac{Z}{W_Z^{mean}}\right), S_{IR}\right] \times H\left(F, F_{mini}\right) - m_Z Z \\ \frac{d\tau}{dt} = -1 \end{cases} \tag{B1}$$

$$\text{if } \tau \leq 0 \begin{cases} \frac{dS}{dt} = -r_S S - m_S S - min\left[IR\left(\frac{Z}{W_Z^{mean}}\right), S_{IR}\right] \times H\left(F, F_{mini}\right) \\ \frac{dF}{dt} = r_S S - m_F F \\ \frac{dZ}{dt} = \gamma \times min\left[IR\left(\frac{Z}{W_Z^{mean}}\right), S_{IR}\right] \times H\left(F, F_{mini}\right) - m_Z Z \\ \frac{d\tau}{dt} = -1 \end{cases} \tag{B2}$$

where $\tau$ (h) corresponds to the potential duration of the biofilm or the potential duration of the production period. It is computed at the end of each nighttime emersion and immersion periods for the next daytime emersion period (Eq. B4 and B6). When $\tau > 0$, the MPB cells migrate upward in the sediment from $F$ to $S$ compartment at a transfer rate of $r_F$ (h⁻¹). MPB stop migration when $S$ reaches saturation at $S_{max}$ (mg chl $a$ m⁻²). Primary production within the $S$ compartment regulated by the mass-specific photosynthetic rate $P^b$ (µg C (µg chl $a$)⁻¹ h⁻¹) is set to zero when $S = S_{max}$ according to the term $\left(1 - \frac{S}{S_{max}}\right)$, which represents the MPB space-limitation in $S$ compartment. The MPB biomass produced is therefore tranferred from $S$ to $F$ according to the term $P^b S\left(\frac{S}{S_{max}}\right)$ in the $F$ time derivative. In order to take into account for all the MPB biofilm biomass plus the biomass produced in the biofilm ($S*$), the $S*$ time derivative was computed as follows:

$$\frac{dS*}{dt} = \frac{dS}{dt} + P^b S\left(\frac{S}{S_{max}}\right). \tag{B3}$$

When $\tau \leq 0$, the MPB cells migrate downward in the sediment from the $S$ to $F$ compartment at a transfer rate of $r_S$ (h⁻¹). The terms $m_S$ and $m_F$ are loss rates (h⁻¹) repesenting MPB senescence and grazing by surface deposit feeders (on $S$) and subsurface deposit feeders (on $F$). $m_Z$ is a loss rate (h⁻¹) representing *P. ulvae* senescence (see Appendix B3).





During the night emersion period, the MPB cells migrate downward into the sediment from $S$ to $F$. *P.ulvae* grazes on MPB cells remaining in the biofilm ($S$):

$$\begin{cases} \frac{dS}{dt} = -r_S S - m_S S - min\left[IR\left(\frac{Z}{W_Z^{mean}}\right), S_{IR}\right] \times H\left(F, F_{mini}\right) \\ \frac{dF}{dt} = r_S S - m_F F \\ \frac{dZ}{dt} = \gamma \times min\left[IR\left(\frac{Z}{W_Z^{mean}}\right), S_{IR}\right] \times H\left(F, F_{mini}\right) - m_Z Z \\ \tau = \left(\frac{F}{S_{max}} + 1\right) \times \tau_s \end{cases} \qquad (B4)$$

According to Guarini et al. (2006, 2008), $\tau$ depends on the MPB biomass in the $F$ compartment relative to $S_{max}$ and on the average time spent at the surface by a unit of biomass equal to $S_{max}$ ($\tau_s$, h). It suggests the higher is the biomass in $F$, the longer $S$ will remain at saturation $S_{max}$. In order to obtain a MPB density-dependant effect, $\tau_s$ is mediated by the ratio of $F$ to a carrying capacity ($F_{max}$):

$$\tau_s = \tau_{s_{max}}\left[max\left(1 - \left(\frac{F}{F_{max}}\right), \tau_{s_{min}}\right)\right] \qquad (B5)$$

$\tau_{s_{max}}$ (1 h, Blanchard et al., 2004) is the maximum time spent at the surface by a unit of biomass equal to $S_{max}$. $\tau_{s_{min}}$ is the minimum value of $\tau_s$ set at 0.02 h. It is set according the minimum time $\tau$ needed to obtain a saturating biofilm i.e. 15 min (Herlory et al., 2004). From now on, the higher is $F$, the lower is the time period during which the biofilm is at the saturation level.

During the immersion period, MPB cells remaining in the biofilm finish their downward migration and *P. ulvae* does not exert any grazing pressure anymore:

$$\begin{cases} \frac{dS}{dt} = -r_S S - m_S S \\ \frac{dF}{dt} = r_S S - \nu_F F \\ \frac{dZ}{dt} = -m_Z Z \\ \tau = \left(\frac{F}{S_{max}} + 1\right) \times \tau_s \end{cases} \qquad (B6)$$

In addition to $m_S$ and $m_F$, $\nu_F$ is a loss rate (h$^{-1}$) that represents MPB cells senescence plus the chronic resuspension of MPB cells in the water column during immersion periods. Guarini et al. (2006, 2008) used to set $S$ to zero at the beginning of immersion periods. The authors considered that all the remaining cells at the surface were resuspended. In our study, the chronic resuspension of MPB cells from the $F$ compartment to the implicit water column is simulated using a higher loss rate ($\nu_F$) compared to losses during emersion ($m_S$ and $m_F$) instead of a potential biofilm complete resuspension. Parameter values are given in Table A3.

## B2 MPB primary production

The mass-specific photosynthetic rate $P^b$ ($\mu g\,C\,(\mu g\,chl\,a)^{-1}\,h^{-1}$) is regulated by temperature ($T$, °C) and by photosynthetically active radiation ($E$, W m$^{-2}$), which corresponds to 44 % of downward shortwave radiation (Britton and Dodd, 1976). The





model of Platt and Jassby (1976) is used to determine the production rate as a function of light:

$$P^b = P^b_{max} \times tanh\left(\frac{E}{E_k}\right),$$ (B7)

where $P^b_{max}$ is the photosynthetic capacity ($\mu$g C ($\mu$g chl $a$)$^{-1}$ h$^{-1}$) and $E_k$ is the light saturation parameter (W m$^{-2}$). $P^b_{max}$ depends on the mud surface temperature $T$ according to the relationship of Blanchard et al. (1996):

$$P^b_{max} = P^b_{MAX} \times \left(\frac{T_{max} - T}{T_{max} - T_{opt}}\right)^{\beta} \times e^{\left(-\beta \times \left[\frac{T_{max} - T}{T_{max} - T_{opt}} - 1\right]\right)},$$ (B8)

where $T_{max}$ (°C) and $T_{opt}$ (°C) are the maximum and optimal temperature for the photosynthesis, respectively. $\beta$ is a curvature coefficient that shapes the temperature-photosynthesis relationship. $P^b_{MAX}$ is the maximum value that takes $P^b_{max}$ at $T_{opt}$.

The mass-specific photosynthetic rate $P^b$ is expressed in $\mu$g C ($\mu$g chl $a$)$^{-1}$ h$^{-1}$. It is therefore necessary to convert it in terms of produced Chl $a$ to obtain a growth rate in h$^{-1}$. To that respect, we used a variable C:Chl $a$ ratio (g C g chl $a^{-1}$) on the

finding of de Jonge et al. (2012) on MPB. The ratio is computed according the formulation of Cloern et al. (1995) adapted for coastal pelagic diatoms (Sibert et al., 2010, 2011; Le Fouest et al., 2013):

$$\frac{Chla}{C} = \left(\frac{Chla}{C}\right)_{min} \times \left(1 + 4 \times e^{-0.5 \times \frac{E}{K_E}}\right),$$ (B9)

where $\left(\frac{Chla}{C}\right)_{min}$ is the minimum Chl $a$:C ratio (g chl $a$ g C$^{-1}$) and $K_E$, the half-saturation constant for light use (in Ein m$^{-2}$ d$^{-1}$).

The MPB primary production ($\mu$g C m$^{-2}$ h$^{-1}$) corresponds to the sum of the space-dependant production at the surface of the

biofilm (*i.e.* the $P^b S\left[1 - \frac{S}{S_{max}}\right]$ term) and of the biomass produced and directly transfered from $S$ to $F$ (*i.e.* the $P^b S\left[\frac{S}{S_{max}}\right]$ term). Consequently, it can be simplified by:

$$production = P^b S\left(1 - \frac{S}{S_{max}}\right) + P^b S\left(\frac{S}{S_{max}}\right) = P^b S$$ (B10)

The constants values are given in Table A3.

## B3  Grazer *P. ulvae*

$S$ is explicitely grazed by the mud snail *Peringia ulvae* ($Z$, mg C m$^{-2}$). The grazing rate is regulated by the individual ingestion rate of snails ($IR$, ng chl $a$ ind$^{-1}$ h$^{-1}$) and $Z$ expressed in terms of density (ind m$^{-2}$). Density is computed as the ratio of $Z$ (mg C m$^{-2}$) over the mean individual weight ($W^{mean}_Z$, mg C) linearly interpolated on simulation time scale (6 min, Table A4). Grazing rate can not exceed the biomass in $S$ per hour ($S_{IR}$, mg chl $a$ m$^{-2}$ h$^{-1}$) by means of the minimum function $min\left[IR\left(\frac{Z}{W^{mean}_Z}\right), S_{IR}\right]$. The grazing is also limited through a heavyside function ($H$) including a feeding threshold ($F_{mini}$,

mg chl $a$ m$^{-2}$). Only a fraction ($\gamma$, %) of the MPB biomass grazed by $Z$ is assimilated into new $Z$ biomass.

The individual ingestion rate (ng chl $a$ ind$^{-1}$ h$^{-1}$) by *P. ulvae* is calculated according to the formulation of Haubois et al. (2005) for adult snails. It depends on the total MPB biomass and on the mud surface temperature:

$$IR = min\left(0.015 \times (F + S)^{1.72} \times e^{\alpha_Z(T - T_{opt_Z})}, IR_{max}\right),$$ (B11)





where $T_{opt_Z}$ (°C) is the optimal temperature for grazing. $IR_{max}$ is the maximal observed individual ingestion rate. $\alpha_Z$ (no unit) is a curvature parameter. The Chl $a$ uptake is converted into carbon unit according to the C:Chl $a$ ratio described previously. The term $(F + S)$ is expressed in $\mu\mathrm{g\,g\,dry\,sed}^{-1}$. The biomass expressed in $\mathrm{mg\,chl}\,a\,\mathrm{m}^{-2}$ is converted into $\mu\mathrm{g\,g\,dry\,sed}^{-1}$ as follows:

$$[Chla]\,(\mu\mathrm{g\,g\,dry\,sed}^{\text{-}1}) = \frac{[Chla]\,(\mathrm{mg\,chl}\,a\,\mathrm{m}^{-2})}{\varphi} \times thickness_{sed}, \tag{B12}$$

where $\varphi$ is the sediment bulk density in $\mathrm{g\,l}^{-1}$ and $thickness_{sed}$ is the sediment thickness i.e. $1\,\mathrm{cm}$. Finally, the mortality rate of $Z$ is a quadratic density-dependant mortality rate:

$$m_Z = m_Z^{min} Z, \tag{B13}$$

where $m_Z^{min}$ is the minimum mortality rate ($\mathrm{h}^{-1}$). The constants values are given in Table A3.

*Author contributions.*

*Competing interests.* The authors declare that they have no conflict of interest.

*Disclaimer.*

*Acknowledgements.* This research was funded by the Centre national d'études spatiales (CNES), the Centre National de la Recherche Scientifique (CNRS, LEFE-EC2CO program), the Région Nouvelle-Aquitaine and the European Union (CPER/FEDER) and the Groupement d'Intérêt Public (GIP) Seine-Aval PHARESEE project. RS was supported by a PhD fellowship from the French Ministry of Higher Education, Research and Innovation. The authors acknowledge Meteo France for providing meteorological data and the Institut National de l'Information Géographique et Forestière (IGN) and the Service Hydrographique et Océanographique de la Marine (SHOM) for providing the digital elevation model of Charente Maritime LITTO3D®. We thank Johann Lavaud (UMI Takuvik, CNRS & Université Laval, Canada) for providing primary production data from measurements made at the study site in 2012 and 2015. We also thank Thierry Guyot (LIENSs) for his help in designing Fig. 2. This research is part of fulfillment of the requirements for a PhD degree (RS) at the Université de La Rochelle, France.



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



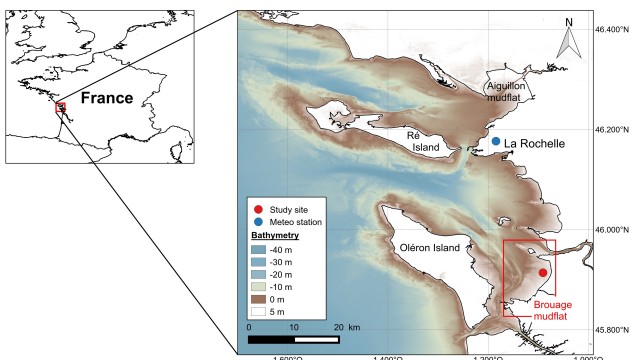

**Figure 1.** Bathymetric map of the Pertuis Charentais (source: French marine service for hydrography and oceanography (SHOM)) and location of the main intertidal mudflats. The study site is represented by a red full point and the Meteo France weather station is represented by a blue full point.



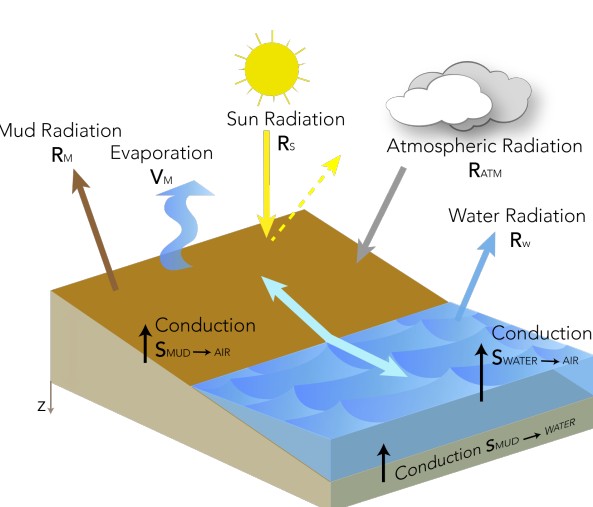

**Figure 2.** Conceptual scheme of heat exchange at the mud surface in the intertidal zone. Fluxes contributing to heat energy balance are represented by arrows during emersion and immersion periods. Modified from Guarini et al. (1997).





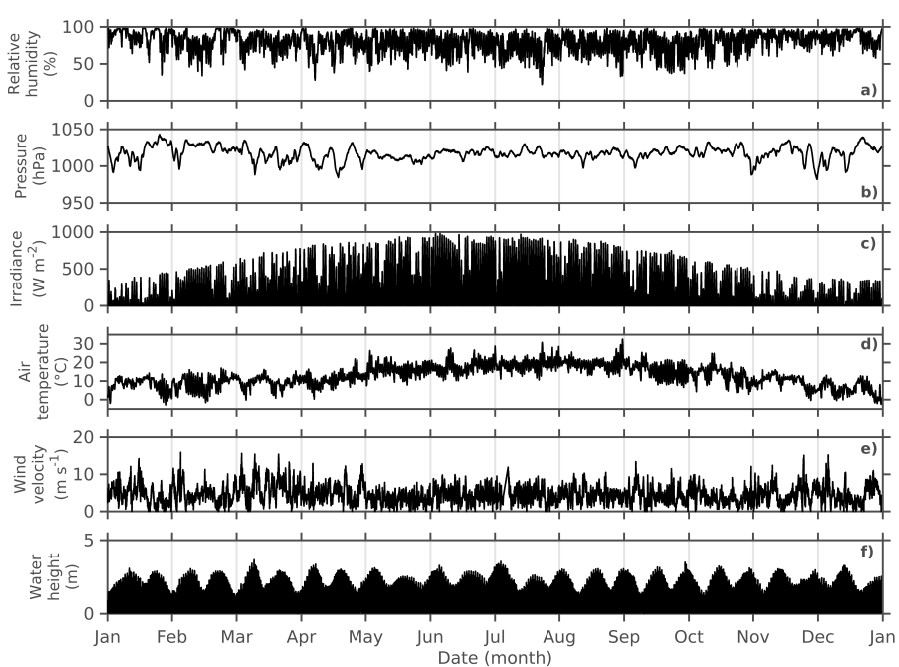

**Figure 3.** Annual cycle of forcings in 2008. a) Relative humidity. b) Atmospheric pressure above the sea. c) Global irradiance. d) Air temperature in the shade. e) Wind velocity. f) Water height at the study site. Meteorological data comes from the weather station located near the airport of La Rochelle and the water height was measured at the tide gauge of La Rochelle-La Pallice corrected by the bathymetry of the study site.





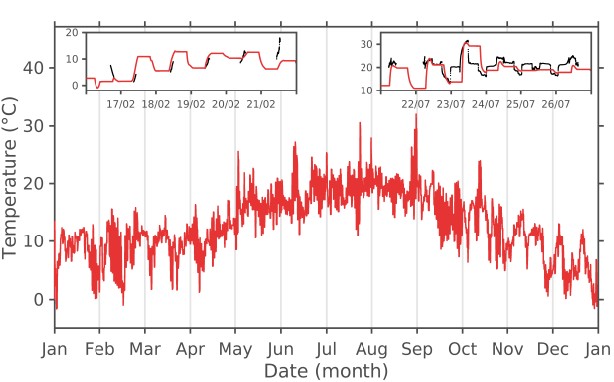

**Figure 4.** Observed (black points) and simulated (red lines) mud surface temperature in 2008.





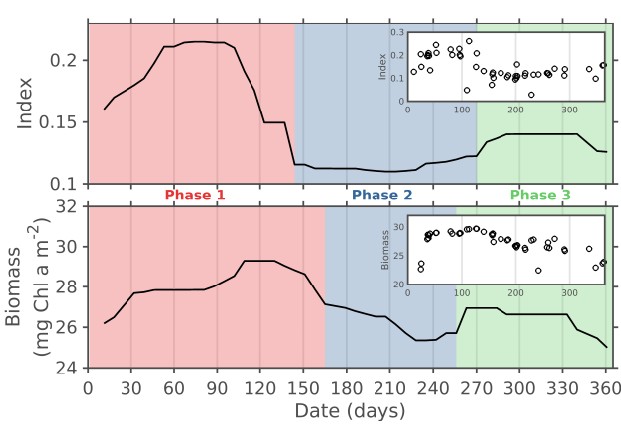

**Figure 5.** Normalized difference vegetation index and simulated biofilm daily maximum biomass in 2008. Original data (small plots) have been regularized and filtered with running medians (window size = 7). NDVI is calculated at the pixel corresponding to the study site. Phases are determined according the amplitude of the sign change of the second order derivative.





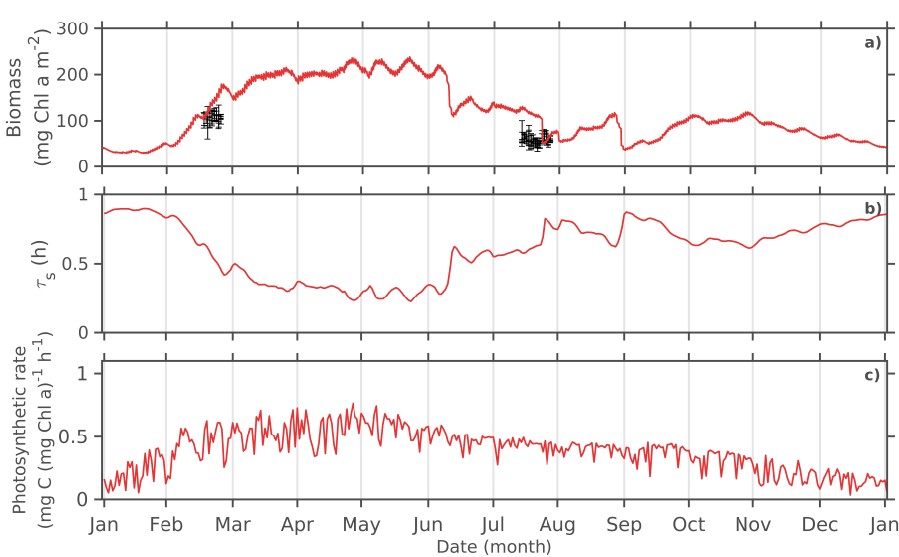

**Figure 6.** Seasonal cycle of the 2008: **a)** simulated total MPB biomass; **b)** average time spent by a unit of MPB biomass at the sediment surface; **c)** average mass-specific photosynthetic rate during daytime low tides. Black error bars correspond to observations ($\pm$ sd).



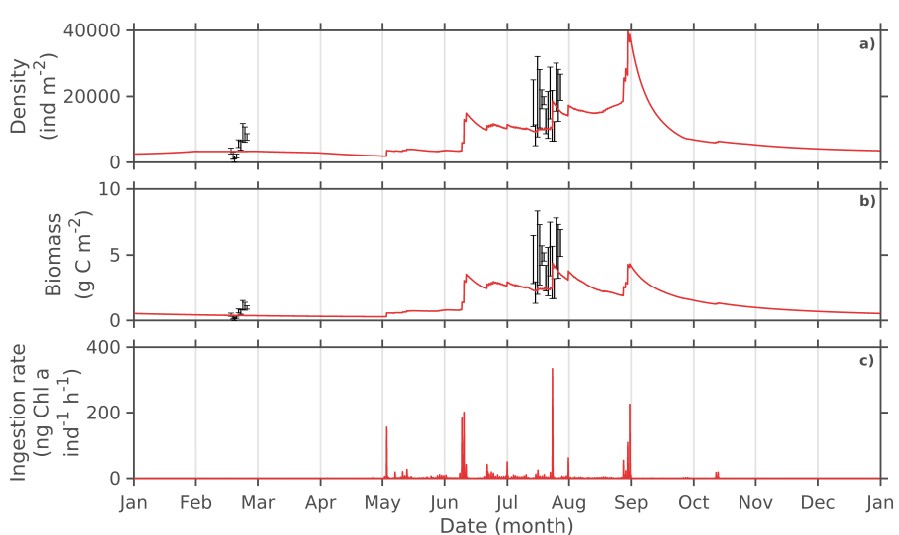

**Figure 7.** Seasonal cycle of the 2008: **a)** Simulated *P. ulvae* density; **b)** simulated biomass of *P. ulvae*; **c)** individual ingestion rate by *P. ulvae*.

Black error bars correspond to observations (± sd).





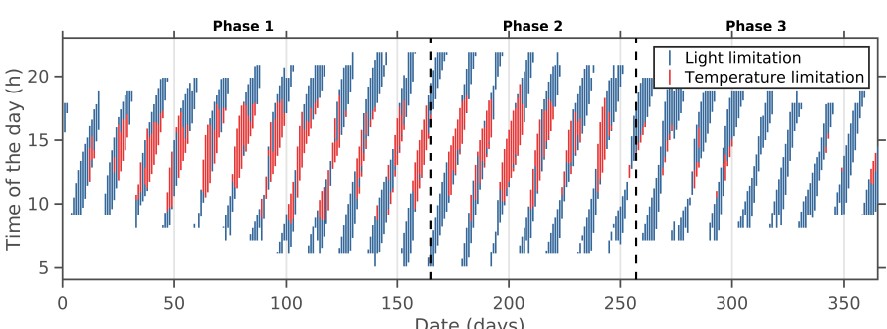

**Figure 8.** Seasonal cycle of daily hours of limitation by light and temperature over daytime emersion periods in 2008.





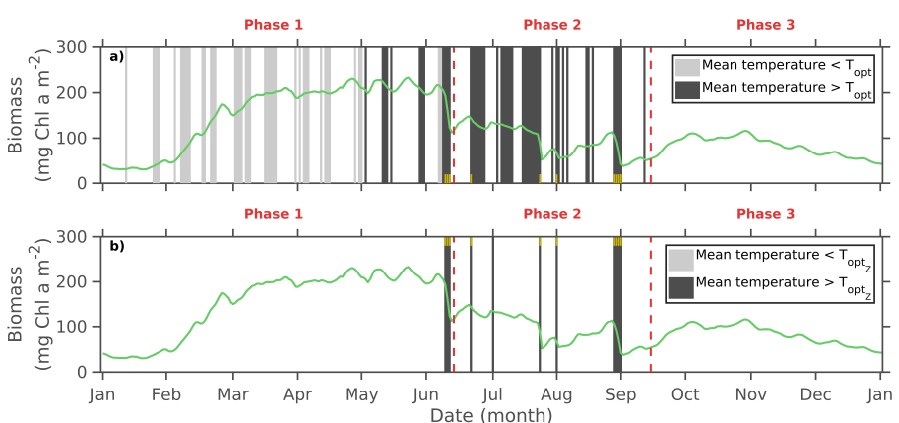

**Figure 9.** Seasonal cycle of the MPB biomass (green full line) along with the daily limiting driver in terms of duration for temperature and in terms of biomass produced vs. ingestion by grazing in 2008: a) Days dominated by temperature limitation, b) days dominated by grazing pressure. Temperature conditions are averaged over the daytime emersion period. The yellow bars indicate the match between temperature and grazing -limited days.





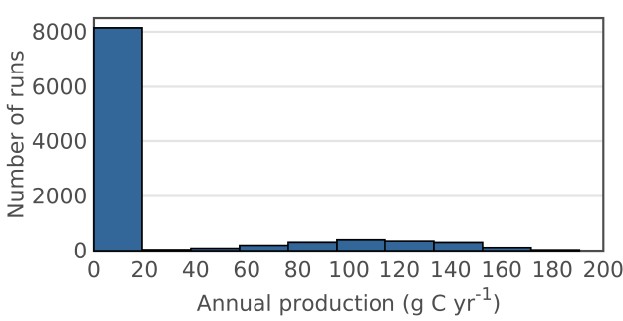

**Figure 10.** Sensitivity analysis of the simulated annual MPB PP.





**Table 1.** Conceptual schemes and differential equations of the biological model including the MPB biomass within the sediment $1^{st}$ cm ($F$), the MPB biomass within the biofilm ($S$) and the biomass of *P. ulvae* ($Z$). The upper case corresponds to daytime emersion periods, when MPB cells migrate at the sediment surface (①) to produce and transfer biomass to the sediment $1^{st}$ cm (②). The middle case corresponds to day or night time emersion period when MPB cells migrate down to the sediment $1^{st}$ cm (③). The lower case corresponds to immersion periods, when MPB cells are chronically resuspended from the $1^{st}$ cm to the water column (④) and the remaining MPB cells within the biofilm finish their downward migration (③). *P. ulvae* grazing is only active during emersion periods (righ side up on schemes)(modified from Guarini et al., 2008).

| Scheme | Cases | Equations |
|---|---|---|
| 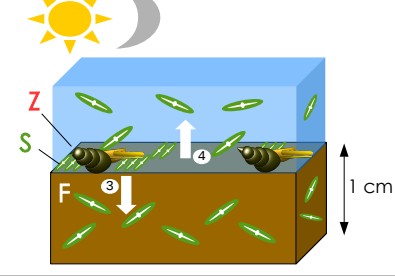 | Day<br>Low tide<br>$\tau > 0$ | $\frac{dS}{dt} = (r_F F + P^b S)\left(1 - \frac{S}{S_{max}}\right) - m_S S$ $-min\left[IR\left(\frac{Z}{W_Z^{mean}}\right), S_{IR}\right] \times H(F, F_{mini})$<br>$\frac{dF}{dt} = -r_F F\left(1 - \frac{S}{S_{max}}\right) + P^b S\left(\frac{S}{S_{max}}\right) - m_F F$<br>$\frac{dZ}{dt} = \gamma \times min\left[IR\left(\frac{Z}{W_Z^{mean}}\right), S_{IR}\right] \times H(F, F_{mini}) - m_Z Z$<br>$\frac{d\tau}{dt} = -1$ |
|  | Day<br>Low tide<br>$\tau \leq 0$ | $\frac{dS}{dt} = -r_S S - m_S S - min\left[IR\left(\frac{Z}{W_Z^{mean}}\right), S_{IR}\right] \times H(F, F_{mini})$<br>$\frac{dF}{dt} = r_S S - m_F F$<br>$\frac{dZ}{dt} = \gamma \times min\left[IR\left(\frac{Z}{W_Z^{mean}}\right), S_{IR}\right] \times H(F, F_{mini}) - m_Z Z$<br>$\frac{d\tau}{dt} = -1$ |
|  | Night<br>Low tide | $\frac{dS}{dt} = -r_S S - m_S S - min\left[IR\left(\frac{Z}{W_Z^{mean}}\right), S_{IR}\right] \times H(F, F_{mini})$<br>$\frac{dF}{dt} = r_S S - m_F F$<br>$\frac{dZ}{dt} = \gamma \times min\left[IR\left(\frac{Z}{W_Z^{mean}}\right), S_{IR}\right] \times H(F, F_{mini}) - m_Z Z$<br>$\tau = \left(\frac{F}{S_{max}} + 1\right) \times \tau_s$ |
|  | High tide | $\frac{dS}{dt} = -r_S S - m_S S$<br>$\frac{dF}{dt} = r_S S - \nu_F F$<br>$\frac{dZ}{dt} = -m_Z Z$<br>$\tau = \left(\frac{F}{S_{max}} + 1\right) \times \tau_s$ |





**Table 2.** Contribution of light and temperature limitation during the three phases of the MPB seasonal cycle.

| Phase | Temperature | Light |
|---|---|---|
| Phase 1 | 33.3 % | 66.7 % |
| Phase 2 | 39.5 % | 60.5 % |
| Phase 3 | 0 % | 100 % |



**Table 3.** Ranges of values for the random selection of parameters value used in the Monte-Carlo sensitivity analysis

| Parameter | Unit | Range | Reference |
|---|---|---|---|
| $T_{opt}$ (temperature optimum for photosynthesis) | °C | [15; 40] | Blanchard et al. (1997); Hubas et al. (2006); Morris and Kromkamp (2003); Rakotomalala et al. (submitted) |
| $T_{max}$ (temperature maximum for photosynthesis) | °C | [$T_{opt}$+1; 40] | Same as $T_{opt}$ |
| $E_k$ (light saturation parameter) | $\mathrm{W\,m^{-2}}$ | [2.5; 180] | Blanchard and Cariou-Le Gall (1994); Barranguet et al. (1998); Light and Beardall (2001); Pniewski et al. (2015); Barnett et al. (2015) and references within |
| $K_E$ (half-saturation constant for light use) | $\mathrm{Ein\,m^{-2}\,d^{-1}}$ | [1; 20] | Sibert et al. (2011); Le Fouest et al. (2013) |
| $T_{opt_Z}$ (optimal temperature for grazing) | °C | [15; 30] | Pascual and Drake (2008) |
| $\alpha_Z$ (shape parameter of the temperature related grazing) | - | [0; 0.8] | Present study |

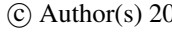



**Table 4.** Metrics obtained from the model sensitivity analysis on annual MPB PP.

| | \multicolumn{7}{c}{Sustainable primary production runs} | | | | | | | \multicolumn{7}{c}{Vanishing primary production runs} | | | | | |
|---|---|---|---|---|---|---|---|---|---|---|---|---|---|---|
| | $T_{opt}$ | $T_{max}$ | $T_{opt_Z}$ | $\alpha_Z$ | $E_k$ | $K_E$ | $T_{amp}$ | $T_{opt}$ | $T_{max}$ | $T_{opt_Z}$ | $\alpha_Z$ | $E_k$ | $K_E$ | $T_{amp}$ |
| $R$ | -0.24 | 0.22 | 0.16 | -0.11 | -0.56 | -0.015 | 0.39 | -0.35 | -0.12 | -0.014 | 0.0071 | 0.022 | -0.041 | 0.44 |
| Mean | 19.00 | 34.00 | 22.00 | 0.39 | 77.00 | 12.00 | 15.00 | 29.00 | 34.00 | 22.00 | 0.40 | 95.00 | 10.00 | 4.90 |
| $\sigma_{norm}$ | 0.14 | 0.14 | 0.19 | 0.60 | 0.64 | 0.43 | 0.28 | 0.22 | 0.16 | 0.19 | 0.58 | 0.54 | 0.54 | 0.73 |
| Min | 15.00 | 20.00 | 15.00 | 0.00 | 2.50 | 1.00 | 4.50 | 15.00 | 16.00 | 15.00 | 0.00 | 2.50 | 1.00 | 0.051 |
| Max | 27.00 | 40.00 | 30.00 | 0.80 | 180.00 | 20.00 | 25.00 | 40.00 | 41.00 | 30.00 | 0.80 | 180.00 | 20.00 | 20.00 |

$R$ is the Pearson's correlation coefficient between annual production values from the different runs with the parameters values associated. $T_{amp}$ corresponds to the difference between $T_{max}$ and $T_{opt}$. $\sigma_{norm}$ is the normalized standard deviation i.e. the standard deviation divided by the mean.



**Table 5.** Temperature optimum and maximum for photosynthesis ($T_{opt}$ and $T_{max}$, respectively; °C).

| Location | $T_{opt}$ | $T_{max}$ | Reference |
|---|---|---|---|
| Marennes-Oléron (France) | 25 | 38 | Blanchard et al. (1997) |
| Roscoff (France) | 21 | 32.5 | Hubas et al. (2006) |
| Ems Dollard (Netherlands) | 30 | 40 | Morris and Kromkamp (2003) |
| Aiguillon Cove (France) | 20 | 30 | Rakotomalala et al. (submitted) |
| Marennes-Oléron (France) | 18 | 38 | Present study |



**Table A1.** Equations of the processes involved in the sediment temperature model

| Process | Symbol meaning |
| --- | --- |
| **Atmospheric and solar radiation** | |
| $R_{sth} = R_0 sin(h)(1-A)$ | $R_{sth}$ : cloudless sky theoritical solar radiation |
| formulated by Brock (1981) | $R_0$ : solar constant |
| | $h$ : sun height |
| | $A$ : albedo |
| $R_{Atm} = \varepsilon_A \sigma T_A^4 (\zeta - k)$ | $\varepsilon_A$: emissivity of air |
| | $\sigma$: Stephan-Boltzman constant |
| | $T_A$ : measured air temperature |
| | $\zeta$: constant ($2 \geq \zeta \geq 1$) |
| | $k$: attenuation coefficient |
| $\varepsilon_A = 0.937 \times 10^{-5} T_A^2$ | |
| $k = \frac{R_S}{R_{sth}}$ | $R_S$: solar radiation |
| $sin(h) = sin(\delta)sin(\phi) + cos(\delta)cos(\phi)cos(AH)$ | $\delta$: declination of the sun |
| | $\phi$: latitude of the area |
| | $AH$: true horary angle |
| **Mud radiation** | |
| $R_M = \varepsilon_M \sigma T_M^4(z_0, t)$ | $\varepsilon_M$: emissivity of mud |
| **Conduction** | |
| $S_{Mud \to Air} = \rho_A C_{P_A} C_{h_{M \to A}} (1+U)(T_M(z_0,t) - T_A)$ | $\rho_A$: air volumetric mass |
| formulated by Pond et al. (1974) | $C_{P_A}$: specific heat of air at constant pressure |
| | $C_{h_{M \to A}}$: bulk transfer coefficient for conduction between mud and air |
| | $U$: wind speed measured at 10 m |
| $S_{Air \to Water} = \rho_A C_{P_A} C_{h_{A \to W}} (1+U)(T_W(t) - T_A)$ | $C_{h_{A \to W}}$: bulk transfer coefficient for conduction between air and water |
| **Evaporation** | |
| $V_W = \rho_A L_V C_V (1+U)\left[q_S\left(1 - \frac{q_A}{q_S}\right)\right]$ | $L_V$: latent heat evaporation |
| | $C_V$: bulk transfer coefficient for evaporation |
| | $q_S$: specific humidity of saturated air at water temperature |
| | $q_A$: absolute air humidity |
| $L_V = [2500.84 - 2.35(T_E - 273.15)] \times 10^3$ | $T_E$: temperature of interstitial water (in equilibrium with mud temperature) |
| formulated by Van Bavel and Hillel (1976) | |
| $q_S = \frac{\lambda p_{sat}^V}{p_{Atm} - (1-\lambda)p_{sat}^V}$ | $\lambda$: ratio between mass constant for dry air and mass constant for the vapor |
| | $p_{sat}^V$: vapor pressure in saturation at interstitial water temperature |
| | $p_{Atm}$: atmospheric pressure |
| $p_{sat}^V = exp\{2.3\left[\frac{7.5(T_E - 273.15)}{237.3 + (T_E - 273.15)} + 0.76\right]\}$ | |
| **Water radiation** | |
| $R_W = \sigma T_W^4(t)$ | |

$k$ is imposed to 1 if greater than 1. During night periods, $k$ is an average of the values 2 h before the night.



**Table A2.** Parameters in the mud surface temperature model

| Parameter | Description | Value | Unit |
|---|---|---:|---|
| **General equations** | | | |
| $\eta$ | Conductivity | 0.8 | $\mathrm{W\,m^{-1}\,K^{-1}}$ |
| $\rho_S$ | Soil volumetric mass | 2650 | $\mathrm{kg\,m^{-3}}$ |
| $\rho_W$ | Water volumetric mass | 1000 | $\mathrm{kg\,m^{-3}}$ |
| $\xi$ | Mud porosity | 0.62 | % |
| $\mu$ | Thermal diffusivity | $0.48 \times 10^{-6}$ | $\mathrm{m^2\,s^{-1}}$ |
| **Solar radiations** | | | |
| $R_0$ | Solar constant | 1353 | $\mathrm{W\,m^{-2}}$ |
| $A$ | Albedo | 0.08 | - |
| **Atmospheric radiations** | | | |
| $\sigma$ | Stephan-Boltzman | $5.67 \times 10^{-8}$ | $\mathrm{W\,m^{-2}\,K^{-4}}$ |
| $\zeta$ | Constant | Radiation on water : 1.05 | - |
| | Radiation on mud : 1.9 | | - |
| **Mud radiation** | | | |
| $\varepsilon_M$ | Mud emissivity | 0.96 | - |
| **Conduction** | | | |
| $\rho_A$ | Air volumetric mass | 1.2929 | $\mathrm{kg\,m^{-3}}$ |
| $C_{P_A}$ | Air specific heat | 1003 | $\mathrm{J\,kg^{-1}\,K^{-1}}$ |
| $C_{P_W}$ | Water specific heat | 4180 | $\mathrm{J\,kg^{-1}\,K^{-1}}$ |
| $C_{h_{M \to A}}$ | Mud-air bulk coefficient | 1 | - |
| $C_{h_{A \to W}}$ | Air-water bulk coefficient | 0.014 | - |
| **Evaporation** | | | |
| $C_V$ | Bulk coefficient | 0.0014 | - |
| $\lambda$ | Constant ratio | 0.621 | - |



**Table A3.** Biological model parameters

| Symbol | Description | Value | Unit | Source |
|---|---|---|---|---|
| **MPB** | | | | |
| $r_S$ | Transfer rate of biomass from $S$ to $F$ | 10 | h$^{-1}$ | Guarini et al. (2008) |
| $r_F$ | Transfer rate of biomass from $F$ to $S$ | 1 | h$^{-1}$ | Guarini et al. (2008) |
| $m_S$ | Loss rate of biomass of $S$ | 0.001 | h$^{-1}$ | Guarini et al. (2008) |
| $m_F$ | Loss rate of biomass of $F$ during emersion period | 0.001 | h$^{-1}$ | Guarini et al. (2008) |
| $\nu_F$ | Loss rate of biomass of $F$ during immersion period | 0.0035 | h$^{-1}$ | Present study |
| $F_{min}$ | Minimum biomass of $F$ | 25 | mg chl $a$ m$^{-2}$ | Present study |
| $F_{max}$ | Maximum biomass of $F$ | 300 | mg chl $a$ m$^{-2}$ | Guarini et al. (1998) |
| $S_{max}$ | Maximum biomass of $S$ | 25 | mg chl $a$ m$^{-2}$ | Guarini et al. (2000) |
| $\tau_{s_{min}}$ | Minimum time spent by a unit of $S_{max}$ at the surface | 0.02 | h | Present study |
| $\tau_{s_{max}}$ | Maximum time spent by a unit of $S_{max}$ at the surface | 1 | h | Blanchard et al. (2004) |
| $E_k$ | Light saturation parameter | 100 | W m$^{-2}$ | Guarini et al. (2000) |
| $P_{MAX}^b$ | Maximum photosynthetic capacity in April | 11.18 | µg C (µg chl $a$)$^{-1}$ h$^{-1}$ | Blanchard et al. (1997) |
| | Maximum photosynthetic capacity in June | 7.56 | µg C (µg chl $a$)$^{-1}$ h$^{-1}$ | Blanchard et al. (1997) |
| | Maximum photosynthetic capacity in September | 5.81 | µg C (µg chl $a$)$^{-1}$ h$^{-1}$ | Blanchard et al. (1997) |
| | Maximum photosynthetic capacity in December | 3.04 | µg C (µg chl $a$)$^{-1}$ h$^{-1}$ | Blanchard et al. (1997) |
| $T_{opt}$ | Optimum temperature for photosynthesis | 18 | °C | Present study |
| $T_{max}$ | Maximum temperature for photosynthesis | 38 | °C | Blanchard et al. (1997) |
| $\beta$ | Shape parameter of the P-T relationship in April | 3.90 | - | Blanchard et al. (1997) |
| | Shape parameter of the P-T relationship in June | 2.80 | - | Blanchard et al. (1997) |
| | Shape parameter of the P-T relationship in September | 1.76 | - | Blanchard et al. (1997) |
| | Shape parameter of the P-T relationship in December | 1.03 | - | Blanchard et al. (1997) |
| $K_E$ | Half-saturation constant for light use | 20 | Ein m$^{-2}$ d$^{-1}$ | Present study |
| $\left(\frac{Chla}{C}\right)_{min}$ | Minimum Chl $a$: C ratio | 0.0125 | g chl $a$ g C$^{-1}$ | Present study |
| **Grazer *P. ulvae*** | | | | |
| $\alpha_Z$ | Shape parameter of the temperature related grazing | 0.55 | - | Present study |
| $T_{opt_Z}$ | Optimum temperature for grazing | 20 | °C | Pascual and Drake (2008) |
| $IR_{max}$ | Maximum individual ingestion rate | 385 | ng chl $a$ ind$^{-1}$ h$^{-1}$ | Coelho et al. (2011) |
| $m_Z^{min}$ | Minimum mortality rate of *P. ulvae* | $6 \times 10^{-7}$ | h$^{-1}$ | Present study |
| $\gamma$ | Assimilation rate | 0.55 | % | Kofoed (1975) |
| **Sediment** | | | | |
| $\varphi$ | Mean bulk density of sediment | 520 | g l$^{-1}$ | Present study |





**Table A4.** Mean individual weight of *P. ulvae* (mg C)

| Month | J | F | M | A | M | J | J | A | S | O | N | S |
|---|---|---|---|---|---|---|---|---|---|---|---|---|
| Weight | 0.21 | 0.13 | 0.11 | 0.11 | 0.15 | 0.22 | 0.26 | 0.23 | 0.10 | 0.23 | 0.19 | 0.15 |