# Peer review of "On biotic and abiotic drivers of the microphytobenthos seasonal cycle in a temperate intertidal mudflat: a modelling study"

_Biogeosciences, 2018_

## Referee Comment (RC1) · P. Cook (Referee) · 20 Jul 2018

The manuscript details a biophysical model to describe primary production, biomass and grazing of MPB on a tidal flat. The work found that the annual cycle of biomass could be reasonably described by light, temperature and grazing inhibition. A key strength of the model is that included a detailed consideration of MPB motility which is a key process in the MPB growth and survival. The factors controlling annual MPB biomass patterns are also relatively poorly understood and attempts to model this complicated process are an important contribution to our understanding and will hopefully stimulate further research to address key uncertainties.

[Figure]

The model showed that a combination of temperature limitation of MPB growth and grazing could explain the dip in biomass observed over summer. The temperature inhibition value used to achieve this outcome had to be 'tuned' to match the data and the temperature inhibition used for MPB was lower than in previous studies, which was concerning. How sensitive is the annual pattern to this temp? Could you show a run where Topt is ~20 C?

Another weakness of the study was that it had a rather limited data set for validation. I am surprised the authors were not able to find a study site with a larger time series of data for grazers and directly measured MPB biomass. Given the importance of the physical model, it was also disappointing that there was very little temperature data, which is very easy to collect. The limited data available also seemed to disagree in pattern and magnitude a lot more than I would have regarded as acceptable for a physical parameter. This data weakness, was somewhat compensated for by the discussion which placed the model inputs and outputs within the context of the literature giving an overall confidence in the general applicability of the model.

One potentially very important factor missing from the model was resuspension. In my mind, this is potentially a very important factor controlling MPB biomass. In Table 1 it seems to be implied this is used in the model (process 4) and also there is later mention of a generic loss term (pg 15 line 31). In the conclusions and perspectives part of the manuscript, it then goes on to say resuspension is not included in the model. Could the authors please clarify what the generic loss term is? I also suggest the discussion of resuspension be included earlier on in the discussion, rather than being raised right at the end. I would also like to see this discussion expanded a little. At present, it really only addresses possible PP by MPB during resuspension, it does not address how much MPB might be exported. The possible resuspension and export of MPB should be discussed and omission from the model justified. Is it possible the loss of biomass is just resuspension on a few windy days?

The manuscript was generally well written and the ideas well constructed. There were

a few spelling and grammatical issues. I have noted a few below, but it would be easiest for the authors to use a spell checker to find these.

Minor comments Pg 6 line 7. Consisted of

Pg 6 l19. Could clarify a little better that (1st cm) means 1st cm of sediment.

pg 8 line 32 onwards. This is a little confusing. first it is stated that grazing is mostly limiting, then it says days where MPB biomass consumed was larger than that produced occurred only 8.7% of the time.

Pg 10 line 31 developing

Pg 12 line 9 delete too

Pg 12 line 15 detrimental

Pg 12 line 26, in that respect (check all uses of this)

Pg 12 line 32 oxidation

Pg 13 line 4. This sentence just repeats the last one, delete

Pg 15 line 13. This implies a very high growth efficiency (13.63/15.8). Can this be correct? Or do they graze other food sources too?

Figure 5. I don't understand why there is a small plot (original data) for biomass. If this is from the model, it should be more continuous? Or perhaps you have only extracted the same days as the NDVI data? Why not show all the data?

Figure 9b. Caption could specify days dominated by grazing pressure when temperature is greater than grazing optimum (T > Toptz). I found this a little hard to understand at first.

---

## Referee Comment (RC2) · Anonymous Referee #2 · 2 Aug 2018

Review of 'On biotic and abiotic drivers of the microphytobenthos seasonal cycle in a temparate intertidal mudflat: a modelling study' by Savelli et al.

Summary The manuscript describes a box model of combined mudflat temperature dynamics and benthic diatom growth. Snails are included as grazers. The model was constructed mostly from existing elements. The model was applied to a mudflat site on the Atlantic coast of southern France, using forcing from a meteorological station and tide gauge data for the year 2008. Simulated mud surface temperatures, diatom biomass and snail biomass and abundance were compared with two short data sets from a few weeks in February and July. Also, visual comparison was made between

simulated diatom biomass and a vegetation index derived from remote sensing data. The sensitivity of the model results to variations of a subset of the model parameters was assessed by a set of monte-carlo simulations analysed by calculating correlation coefficients between the varied parameters and diatom primary production, showing primary production collapse for most of the monte-carlo simulations, and fairly weak correlations for the simulations that did not collapse. The reference run was analised predominantly in terms of the seasonal cycle, limiting effects of light and temperature in relation to submergence/exposure and grazing, showing that temperature inhibition occurs around mid-day, light limitation in the early morning and late afternoon, and predominantly low levels of grazing except for a few episodic events.

General comments

I found this an interesting and well-written manuscript, which is appropriate for the journal. However, I have a number of major concerns regarding i) the validation, ii) the sensitivity analysis, iii) water temperature calculations, iv) snail model.

i) Validation

The two short periods of in-situ data are not sufficient to constrain the seasonal cycle. With these data, many other potential modelled seasonal cycles, including constant values (straight lines) could be equally valid results. The authors mention a monthly data set of chlorophyll observations from the same mud flat covering March 1992 to February 1993. A simulation for this period should be included and compared with the observations. The remote-sensing data are not really a substitute for this because they may have their own issues, and in the current manuscript are not the same variable.

ii) Sensitivity analysis

The sensitivity analysis leaves me puzzled. Why calculate correlation coefficients which assume linearity if the model equations are clearly non-linear? For which areas of the varied parameter space does the primary production collapse? Why? Why

was this subset of parameters selected and not others? Instead of randomised monte-carlo simulations and questionable statistics, I would think a series of graphs where primary production is plotted as a function of a varied parameter (with others set to reference values) would be much more instructive, or should in the least be used to analyse what happens in the monte-carlo simulations. Such an approach could even be used to restrict the range of variation of the parameters such that not so many simulations collapse (if the collapsing simulations occur towards the extremities of the parameter space). These new ranges could then also be critically compared with the ranges reported in the literature.

iii) Water temperature calculations

Eq. A7 should include S(mud_to_water). Also, the heating/cooling of the water column should be related to the instantaneous water depth. I can't find this in the equations. Are these just issues with the representation in the manuscript, or is the heat balance model flawed? This should be corrected.

iv) Snail model

I'm puzzled by the few sharp peaks in ingestion. Is this realistic behaviour or an artifact of the model? If the latter, could it be related to the exponent in Eq B11, which can change sign depending on the temperature? This seems odd from a mathematical perspective. Was this kind of behaviour of the equation envisaged/included in the range of values considered in the publication in which this relationship was proposed?

I have a number of additional concerns and suggestions that I will detail below. Overall, I recommend major revisions or rejection with encouragement to re-submit.

Further important issues

-It is suggested (p. 6, l. 10) that the mud temperature (fig 4) closely follows the air temperature (fig 3). This is difficult to see. Please include the air temperature in fig 4 for better comparison.

-The point above triggers the question if a full mud temperature model is necessary. This question could be easily addressed by driving the microphytobentos model with the air temperature (or air temperature when exposed and water temperature when submerged) and comparing the results with the reference simulation.

-p. 7, l. 30-p. 8, l. 2. 4xsignificant. I disagree. These differences are not significant, because the model mean is within the confidence interval of the observations.

-p. 8, light limitation. The definition is confusing. Also during the night, light is the limiting factor. Please use the full 24 hr period, not just daylight hours to represent this.

-Discussion. The authors provide a substantial number of numeric comparisons with published results throughout the discussion. This information is very difficult to digest in this way. Please compile a table of all these data/values/references, and present as part of the results.

-Table 1. I'm not sure if figures are allowed within a table - check journal requirements. This table doesn't seem to contain new information compared with the text (appendix B1). Ensure there is no duplication (delete table?).

-Appendix B1. dZ/dt is identical in the three cases. Please print only once. Also B4 is identical to B2 except for the formulation for tau - find an alternative way to present this without duplication.

Detail

-I've spotted several typos - please use a spell checker.

-p. 1, l. 10. events of biomass reduction when

-p. 1, l. 14. export flux: from, to?

-p. 2, l. 10. migrate towards

-p. 2, l. 27. De Jonge

-p. 3, l. 6. the composition of the benthic diatom assemblage?

-p. 4, l. 4. each set of measurements

-p. 4, l. 7. The reference to fig 3 occurs before the first ref to fig 2. Swap figures.

-p. 4, l. 7. consisted of

-p. 4, l. 25. spatially averaged

-p. 5, l. 31. This sentence is unclear.

-p. 6, l. 13. This sentence is unclear.

-p. 6, l. 15. First use, write out the names of the variables. Why these - there are many others (Table A3)?

-p. 6, l. 23-24. This sentence is unclear.

-Fig 5. Label graphs. Also plot 'original data' in the main figure for better comparison. Rephrase caption to make it clear what these original data are.

-p. 7. l. 2-5. Fall bloom. This seems less evident in the 'original data'? Is that true and if so why?

-p. 7, l. 28. one month

-Figure 9. The white colour is missing from the legends. For graph b, there is no grey. Is this actually the case or an issue with the figure? It seems that in graph b, T_opt was plotted, not T_opt_z?

-p. 9, l. 3. limiting

-p. 9, l. 17. key: why are these 'key' (and how is that defined)?

-p. 12, l. 29. Here, a section starts on salinity (it's not entirely clear to me where this ends). This is the first mention of salinity, and as far as I understand salinity is not represented in the model. So this paragraph seems a bit out of place. Either delete, or

argue why salinity was not included in the model in Methods, and then move this bit to a separate heading in the discussion.

-p. 30, l. 31-35. This contradicts statements in Results.

-Figure 8. Why does the vertical axis start at 5? The plot seems to suggests that this truncates the data in mid-summer?

-emersion (is it emergence?) and immersion are easily confused, please use exposure and submergence.

-Figure 1. the font size used for latitude and longitude may be too small.

---

## Referee Comment (RC3) · Anonymous Referee #3 · 5 Aug 2018

Savelli et al. Biogeosciences 2018 325

This is an interesting and well presented study that models the annual cycle of microhpytobenthos biomass and activity on intertidal mudflats on the Atlantic coast of France, and, using key environmental drivers of temperature and irradiance, and the biotic interaction derived from gastropod grazing (Peringia, prev. known as Hydrobia), show a three phase response over the annual cycle. The modelling is well presented and characterised, based on an extensive literature review of available sources and variables. The modelling supports a set of well accepted (if not always well described in the quantitative literature) assumptions about the main drivers of daily and annual MPB

production (light availability, temperature stress) and summer depressions of biomass due to grazing pressure. To some extent this is to be expected, as the model is constructed with a number of a priori assumptions, so providing the mathematics works, then once would expect to see the patterns that are produced. The modelling produces results consistent with earlier work, and a general acceptance that annual primary production on such mudflat systems is somewhere of the order of 120-150 g C m-2 y-1. The only outcome that I found did not fit my preconceived understanding on MPB dynamics was that Peringia grazing actually only had a significant effect on very few days over the summer. That is a surprise.

The weakness with the study is the lack of a parallel data set of primary variables (biomass, production, grazing density and pressure) with which to validate the model. There are a couple of periods when the model is shown to approximate to some corresponding field data, though there is an annual NVDI data set used to support the surface biomass aspects. The rest of the model is not validated. I think this is a problem because the extensive discussion implicitly relies on the model being correct, and then provides an interesting and well referenced discussion around various driving factors and other factors that may play a role. I think the authors need to validate their model using some other data sets, perhaps from some of the other mudflat systems that they have (and are) working on within the Atlantic / Channel seaboard, or resolved at finer temporal scales to demonstrate the robustness of the assumptions under pinning the model. After all, if the model works on one mudflat, it ought to be applicable to other similar systems, and this would really demonstrate its value to others workers in the field.

Some more specific points are made below:

Overall, what are the error terms around the modelled responses? The figures show some significant error terms in the existing field data, but no errors around the model outcomes.

P3, L15. "in the light of current knowledge….role still unclear". I think there is a very extensive set of literature on the roles of abiotic and biotic factors for MPB dynamics, so this statement portrays a false sense of uncertainty. P5, L32. Given an extensive literature (some of which is mentioned in the discussion) about resuspension of MPB by wind/wave action, why was this not included in the model? The wind data were available, and weather-induced and tidal wash-away effects are shown to be significant in removing MPB biomass?

Figure 5 is an important figure. It needs to be made clear in the legend that this refers to S*. Why when the NVDI signal varies by over 100% in the course of the year, does the S* value only vary by at most 6-7%. Though the "pattern" looks the same (what is the correlation or correspondence between the two annual cycles?), the order of magnitude of change does not. How can this be, when they are assumed to be measuring the same thing?

P7, L9 onwards. The variable Ts is dependent on overall biomass, but then the outcomes of this seem counter-intuitive to what we know about biofilms and cell microcycling. Cells appear to spend the time they need at the surface to photosynthesise and accumulate enough carbon, while minimising their risk of photodamage. So each cell spending 54 minutes at the surface during January and August, while only 12 minutes in April, appears to be an outcome of an underlying assumption about biomass, rather than an understanding about diatom photophysiology and behaviour?

P7, L20, clarify if this is the assumed intrinsic growth rate?

P8, L14 onwards. This section appears to be saying that during the summer periods, the biofilms are light limited, because there are longer days? If this just a mathematical artefact? After all, an individual cell only needs some many quanta of light to meet its photosynthetic requirements, and with variable migration, lower biomass and longer days, why would individual cells be light limited?

P8 L30 and P15,L1 This is the one area I found surprising, given the number of published accounts of strong inverse correlations between Peringia (Hydrobia) abundance and biomass on NW European mudflats. Particularly when the authors have said in an earlier paragraph that during phase 2 light was limiting, which would make the biomass response even more susceptible to being grazed down? How convinced are the authors that this is a true situation, or is the model not capturing the real impact of grazers during this phase?

P13, L6, see Steele et al. Biofouling 30, 987 – 998 for a detailed study of EPS and desiccation on diatom photosynthetic capacity

P18, L3. What happens if a resuspension element is included in the model (Dupuy et al gives 3%, Blanchard et al 2006, in In J. Kromkamp [ed.], Functioning of microphytobenthos in estuaries: Proceedings of the microphytobenthos symposium, Amsterdam, The Netherlands, August 2003. Royal Netherlands Academy of Arts and Sciences, and Hanlon et al. 2006 Limnol. Oceanogr. 51: 79-93, provide other values, and de Jonge and van Beusekom (op. cit) provide some critical wind speeds)?

---

## Author Comment (AC1) · 18 Sep 2018

We gratefully thank referee #1 for his constructive comments with respect to our manuscript. In order to improve the manuscript with respect to these comments, we amended the manuscript as suggested by the referee wherever it was possible.

The modifications made in the manuscript are based on a new model run, which includes the model improvements suggested by the three reviewers. They include the mud temperature model, the P. ulvae grazing mathematical formulation and the setting of the mean time spent by a MPB cell at the sediment surface. As a result, the simulated data presented here are modified compared to the initial version of the

manuscript.

1. "The temperature inhibition value used to achieve this outcome had to be 'tuned' to match the data and the temperature inhibition used for MPB was lower than in previous studies, which was concerning. How sensitive is the annual pattern to this temp? Could you show a run where Topt is ~20 C?"

The different microphytobenthic Topt estimation techniques and contrasting in situ versus laboratory conditions explain the range of values reported in the literature (from 20 to 30 °C). In the model run where Topt was set to 20°C the seasonality of MPB is very similar to that depicted by the model ran with Topt value set to 18°C. However, while the simulated MPB biomass was maintained during winter 2008-2009 with a Topt value of 18 °C, it vanished with a Topt value of 20 °C. Such a result suggests that at our study site a Topt value of 18 °C allows MPB to cope with cold winter conditions and relatively low mud surface temperatures.

In the discussion section (4.2), additional detrimental effects of high temperature on epipelic diatoms are discussed. Those effects are not considered in laboratory-based estimates of Topt given in Blanchard et al. (1997) because the authors worked with a suspension of MPB cells. In 2008, PAM measurements showed a lower photosynthetic activity in July than in February on the Brouage mudflat (pers. com. S. Lefebvre and J. Lavaud). The authors suggest that micromigrations are lowered by high mud surface temperature (and so by the pore water evaporation and resulting increase of salinity) and that MPB is unable to avoid harmful light conditions at mud surface temperature even lower than 20 °C. MPB cells are therefore photo-inhibited via high temperature-related processes. As photo-inhibition is not accounted for in the Production-Irradiance relationship used in the model, a lower Topt value than the range estimated in the literature is a way to account for the negative feedback of photo-inhibition on the photosynthetic rate triggered by high temperature-related processes within the sediment.

We modified the Discussion section 4.2 (p. 13, l 8-11) to make it clearer:

[Figure]

" As the detrimental effects of high salinity levels is not explicitly accounted for in the model as a forcing variable, they are implicitly accounted for through temperature-related mechanisms, i.e. a Topt value lower than values reported in the literature (Table 5). Such an approach overestimates the thermo-inhibition process and, as such, promotes low PP rates that implicitly reproduces in the model the detrimental effects of desiccation on the microphytobenthic cells."

2. "Another weakness of the study was that it had a rather limited data set for validation. I am surprised the authors were not able to find a study site with a larger time series of data for grazers and directly measured MPB biomass. "

The lack of validation data was pointed out by the three referees. We agree with this comment. Located 1.7 km from the shore our study site is remote. It is, however, the most studied site in the area but the sampling variables and protocols vary from year-to-year. We hence made the choice to use 2008 data from the French national project VASIREMI as it is unique in the area in terms of space and time coincident in situ measurements of both physical (sediment temperature) and biological (MPB and grazer biomass) variables during two contrasting seasons. In addition, high resolution atmospheric and oceanic forcings required to constrain the model are available for 2008.

To cope with the lack of data, we used two datasets of in situ MPB Chl a concentration available for the same station. The two datasets cover the spring, summer and winter seasons in 2012 and 2013. We added a new Figure (R1, attached to the answer to referees) to show the MPB seasonal cycle in terms of Chl a concentration based on the 2008, 2012 and 2013 data.

We added a new sentence in the Materials and Methods section (2.1.1) as follows: "In addition to the 2008 dataset, we used data of in situ MPB Chl a concentration collected within the 1st cm of sediment at the same station in April 19 – 22, 2012, July 05, 2012, November 14, 2012, February 11, 2013 and April 10, 2013). The sampling protocol is

fully detailed in Lavergne et al. (2017)."

The new figure (R1, attached to the answer to referees) aims at showing the observed seasonal cycle of MPB Chl a at our study site based on the data available, i.e. a 3-year dataset (2008, 2012, 2013). A new paragraph was added in the Results section (3.2) as follows:

"Two distinct periods were identified from in situ Chl a measurements in the sediment first cm (Fig. R1). The observed seasonal cycle of Chl a was characterised by a spring bloom and by a decrease of Chl a concentration in summer. Given the few available measurements in autumn, the seasonal MPB dynamics at this season remained uncertain.The maximum of Chl a concentration reached during the spring bloom ranged between 234 and 306 mg Chl a m-2, which is consistent with the concentration simulated by the model in 2008."

3. "Given the importance of the physical model, it was also disappointing that there was very little temperature data, which is very easy to collect. The limited data available also seemed to disagree in pattern and magnitude a lot more than I would have regarded as acceptable for a physical parameter."

We took into account the referees #1 and #2 comments regarding the sediment temperature model. Based on Guarini et al. (1997) and Phizacklae (1987), we modified the physical model by setting a rapid equilibrium between mud surface temperature and the temperature of the overlying water layer, i.e. the simulated mud surface temperature is now set to water temperature during immersion periods. In the new model run, the root mean squared error (RMSE) between the observed and simulated MST values for the two 2008 periods (RMSE=1.81 °C) is reduced twice compared to the previous model run (RMSE=3.98 °C). In addition, the Pearson's correlation coefficient is higher in the new model run (r=0.93) compared to the previous model run (r=0.76). The referees comments helped to significantly improve the model capability to simulate the MST.

4. "In the conclusions and perspectives part of the manuscript, it then goes on to say resuspension is not included in the model. Could the authors please clarify what the generic loss term is?"

We agree with referees #1 and #3 that the resuspension process is not explicitly detailed in the manuscript. As there are no data available of current velocity on the sea bed in 2008 at our study station, we did not infer on hydrodynamically-related resuspension processes of MPB. In the model, we assumed a constant rate of MPB cells resuspended during immersion periods. During immersion periods, the generic loss term (vF, 0.003 h-1) includes the chronic resuspension and MPB senescence processes. During emersion periods, the loss term is lower (mF, 0.001 h-1) as it only represents the MPB senescence.

5. "I also suggest the discussion of resuspension be included earlier on in the discussion, rather than being raised right at the end. I would also like to see this discussion expanded a little. At present, it really only addresses possible PP by MPB during resuspension, it does not address how much MPB might be exported. The possible resuspension and export of MPB should be discussed and omission from the model justified."

We agree with referee #1 that some text on how much MPB might be exported and on the justifications about our mathematical formulation were lacking in the discussion. In the original version of the manuscript, we included this part in the perspectives section, because we currently work on the MPB resuspension mechanisms and related physical processes to be further included in the model. The referee #1 comment was hence taken into account by adding some text in the Results and Discussion sections.

The Results section was modified as follows: "In the model, the linear loss terms applied to the MPB biomass simulated within the 1st cm translated into a yearly averaged resuspension of 1.7 $\pm$ 0.3% of the averaged MPB biomass in the 1st cm of sediment during high tides. Over the year, 25 % of the simulated MPB production during low

tides was resuspended, which corresponded to a total annual resuspension of 31.6 g C m-2. "

The Discussion section was modified as follows: "The short-term daily dynamics of MPB is regulated by resuspension events (Blanchard et al., 2002). The intensity of resuspension of MPB into the water column can be either chronic or catastrophic according to the flow velocity and the sediment stabilization (Marrioti and Fagherrazzi, 2012). Catastrophic events can locally resuspend all the MPB biomass as the resuspended sediment layer is thicker than the vertical distribution of MPB biomass (Marrioti and Fagherrazzi, 2012). The repeated occurrences of such events over several days can shape the seasonal cycle of MPB by lowering the biomass of photosynthetically competent MPB. In their model, Guarini et al., (2008) introduced a chronic resuspension of all the MPB biomass remaining in the biofilm when tidal floods occurred. In their parametrization, the MPB biomass remains at the sediment surface according to a mean time spent at the surface (equivalent to tau in our study). In our model, the chronic resuspension of MPB biomass is formulated by a linear loss term of the MPB biomass within the first cm (0.002 h-1). In the absence of MPB biomass deposition, the total simulated MPB biomass which is resuspended into the water column represents 25% of the simulated benthic MPB annual production. Such a value supports the fact that benthic MPB production contributes significantly to the pelagic food web (Perissinotto et al., 2003; Krumme et al., 2008). In the light of the work of Marrioti and Fagherrazzi (2012), resuspension and deposition are key mechanisms that need to be related to fauna bioturbation, sediment characteristics (e.g. nature and stabilization) and hydrodynamics (Marrioti and Fagherrazzi, 2012). Such an approach requires the availability of waves and current data to estimate the bed shear stress and modulate the intensity of resuspension (from chronic to catastrophic events), which are not available at our study site for 2008. "

6. "Is it possible the loss of biomass is just resuspension on a few windy days?"

Bed shear stress induced by physical factors (i.e. current and wave orbital velocities,

bed roughness) and sediment stabilization control the resuspension of sediment and associated MPB (Tolhurst et al., 2003). Dupuy et al. (2014) showed that benthic diatoms are resuspended at a friction velocity of 3 cm s-1. This critical friction velocity for diatoms resuspension can be lower than the tidal current velocity without the action of wind during spring tides on sheltered mudflats according to the simulations of Le Hir et al., (2000). In addition, the impact of grazing activity by benthic deposit feeders has to be considered. Bioturbation generates a fluff layer of sediment-organic matrix, which is resuspended at a lower critical friction velocity (1 cm s-1 for P. ulvae bioturbated fluff layer; Orvain et al., 2004). Chronic resuspension of MPB cells can therefore occur with no wind, as shown by Guarini et al. (2008). Furthermore, waves and winds interact with tidal currents. When considering an angle between the waves and the current direction for the bed shear stress calculation (Soulsby, 1997), the wave forcing can be antagonistic, synergetic or neutral on the current bed shear stress according to the tidal and the wave conditions. Resuspension can hence occur without any action of winds.

7. "The manuscript was generally well written and the ideas well constructed. There were a few spelling and grammatical issues. I have noted a few below, but it would be easiest for the authors to use a spell checker to find these."

Spelling and grammatical issues have been checked according to the referee comment.

8. "Pg 4 l19. Could clarify a little better that (1st cm) means 1st cm of sediment."

It has been changed by (1st cm of sediment).

9. "Pg 8 line 32 onwards. This is a little confusing. first it is stated that grazing is mostly limiting, then it says days where MPB biomass consumed was larger than that produced occurred only 8.7% of the time."

The sentence is confusing. Considering the modification made on the grazing formulation in the model, the results are now different from the original version of the manuscript. Grazing is now mostly significant during phase 1. The sentence was modified as follows:

"The top-down control by P. ulvae occurred mostly during phase 1, when the ingested MPB biomass exceeded the MPB PP during 7% of the time (Fig. 9b)."

10. "Pg 15 line 13. This implies a very high growth efficiency (13.63/15.8). Can this be correct? Or do they graze other food sources too?"

The initial sentence is confusing: "The combined effect of grazing and thermo-inhibition translates into 22 % (5.9 g C) of the simulated annual MPB PP channelled towards P. ulvae gross secondary production. The simulated P. ulvae gross secondary production is 27 g C m−2 y−1".

On the 5.9 g C grazed by P. ulvae during events of combined grazing and thermo-inhibition only 55% (assimilation rate $\gamma$, Annexe B) was assimilated by P. ulvae, which corresponds to 3.2 g C. The sentence was hence modified as follows:

"The combined effect of grazing and thermo-inhibition translates into 22 % (5.9 g C) of the simulated annual MPB PP channelled towards the P. ulvae compartment before assimilation and gross secondary production. The simulated annual P. ulvae gross secondary production is 27 g C m−2 y−1."

11. "Figure 5. I don't understand why there is a small plot (original data) for biomass. If this is from the model, it should be more continuous? Or perhaps you have only extracted the same days as the NDVI data? Why not show all the data?"

For the model/satellite comparison, the MPB biomass simulated in the biofilm was extracted for the same days than the NDVI data. In order to filter both NDVI and simulated MPB biomass data, the resulting time series were first regularised to obtain regular time intervals between all points. Small plots were hence showed to illustrate the shape of the original time series before the numerical treatment. The figure and the caption were improved according to the comment of referee #2 to make it clearer (small plots were removed and the extracted points were overlaid on the main plots).

12. "Figure 9b. Caption could specify days dominated by grazing pressure when temperature is greater than grazing optimum (T > ToptZ). I found this a little hard to understand at first."

The information given in the legend was included and detailed in the figure caption to make it clearer.

Captions of attached figures:

Fig. R1: Daily averaged in situ MPB biomass sampled in the sediment 1st cm at the study station on the Brouage mudflat in 2008 (black full dots), 2012 (grey full dots) and 2013 (blue full dots). Error bars correspond to the standard deviation.

Fig. R2: Univariate sensitivity analysis of the simulated MPB annual production to: a) the temperature optimum for MPB growth (Topt); b) the temperature maximum for MPB growth (Tmax); c) the light saturation parameter (Ek); d) the half-saturation constant for light use (KE); e) the optimal temperature for grazing (ToptZ); f) the shape parameter of the temperature related grazing (alphaZ). r is the Spearman's correlation coefficient (asterisk inform when $p < 0.05$) and N is the number of tested values for each biological constant.

Fig. R3: Difference between the observed air temperature and the simulated mud surface temperature (°C) in 2008.

Fig. R4: Seasonal cycle of the MPB simulated biomass in the 1st cm of sediment in the presence (red full line) and in the absence of P. ulvae (red dashed line) in 2008.

[Figure]

**Fig. 1.**

**Fig. 2.**

a) N =13, r = -1*
b) N =9, r = 1*
c) N =72, r = -1*
d) N =20, r = 0.9985*
e) N =19, r = 1*
f) N =15, r = -1*

Annual production (g C yr$^{-1}$)

Temperature (°C)

Irradiance (W m$^{-2}$)

Irradiance (Ein m$^{-2}$ d$^{-1}$)

Temperature (°C)

Dimensionless

[Figure]

Fig. 3.

[Figure]

**Fig. 4.**

---

## Author Comment (AC2) · 18 Sep 2018

We gratefully thank referee #2 for her/his constructive comments with respect to our manuscript. In order to improve the manuscript with respect to these comments, we amended the manuscript as suggested by the referee wherever it was possible.

The modifications made in the manuscript are based on a new model run, which includes the model improvements suggested by the three reviewers. They include the mud temperature model, the P. ulvae grazing mathematical formulation and the setting of the mean time spent by a MPB cell at the sediment surface. As a result, the simulated data presented here are modified compared to the initial version of the

manuscript.

1. "The two short periods of in-situ data are not sufficient to constrain the seasonal cycle. With these data, many other potential modelled seasonal cycles, including constant values (straight lines) could be equally valid results. The authors mention a monthly data set of chlorophyll observations from the same mud flat covering March 1992 to February 1993. A simulation for this period should be included and compared with the observations. The remote-sensing data are not really a substitute for this because they may have their own issues, and in the current manuscript are not the same variable."

The lack of validation data was pointed out by the three referees. We agree with this comment. Located 1.7 km from the shore our study site is remote. It is, however, the most studied site in the area but the sampling variables and protocols vary from year-to-year. We hence made the choice to use 2008 data from the French national project VASIREMI as it is unique in the area in terms of space and time coincident in situ measurements of both physical (sediment temperature) and biological (MPB and grazer biomass) variables during two contrasting seasons. In addition, high resolution atmospheric and oceanic forcings required to constrain the model are available for 2008, which is not the case for 1992 and 1993.

To cope with the lack of data, we used two datasets of in situ MPB Chl a concentration available for the same station. The two datasets cover the spring, summer and winter seasons in 2012 and 2013. We added a new Figure (R1, attached to the answer to referees) to show the MPB seasonal cycle in terms of Chl a concentration based on the 2008, 2012 and 2013 data.

We added a new sentence in the Materials and Methods section (2.1.1) as follows: "In addition to the 2008 dataset, we used data of in situ MPB Chl a concentration collected within the 1st cm of sediment at the same station in April 19 – 22, 2012, July 05, 2012, November 14, 2012, February 11, 2013 and April 10, 2013). The sampling protocol is

fully detailed in Lavergne et al. (2017)."

The new figure (R1, attached to the answer to referees) aims at showing the observed seasonal cycle of MPB Chl a at our study site based on the data available, i.e. a 3-year dataset (2008, 2012, 2013). A new paragraph was added in the Results section (3.2) as follows:

"Two distinct periods were identified from in situ Chl a measurements in the sediment 1st cm (Fig. R1). The observed seasonal cycle of Chl a was characterised by a spring bloom and by a decrease of Chl a concentration in summer. Given the few available measurements in autumn, the seasonal MPB dynamics at this season remained uncertain.The maximum of Chl a concentration reached during the spring bloom ranged between 234 and 306 mg Chl a m-2, which is consistent with the concentration simulated by the model in 2008."

With respect to the satellite data, we agree with the referee #2 that the remotely-sensed NDVI and simulated Chl a concentration data cannot be quantitatively compared as they are not the same variable. However, the Spearman's correlation coefficient between the NDVI and the simulated Chl a in the biofilm is 0.58 (p < 0.05). The NDVI/simulated Chl a relationship is therefore qualitatively reliable and can inform on the MPB seasonality. At a constant Chl a concentration, the Chl a pigments would absorb more light in summer than in winter because of the package effect. The remotely-sensed NDVI would hence be expected to be higher in summer than in winter for a same biomass. However, based on field measurements, the NDVI is observed to be higher in winter (March) than in summer suggesting that the package effect of the Chl a pigments has no influence on the NDVI seasonality.

2. "The sensitivity analysis leaves me puzzled. Why calculate correlation coefficients which assume linearity if the model equations are clearly non-linear? For which areas of the varied parameter space does the primary production collapse? Why? Why was this subset of parameters selected and not others? Instead of randomised monte-carlo

simulations and questionable statistics, I would think a series of graphs where primary production is plotted as a function of a varied parameter (with others set to reference values) would be much more instructive, or should in the least be used to analyse what happens in the Monte-Carlo simulations. Such an approach could even be used to restrict the range of variation of the parameters such that not so many simulations collapse (if the collapsing simulations occur towards the extremities of the parameter space). These new ranges could then also be critically compared with the ranges reported in the literature."

We took into account the referee #2 comment about the non-linearity in the model between the biological constants and MPB annual production. We performed the same sensitivity analysis but we used the Spearman's correlation coefficient, which is adapted for non-linear relationships. The results of the sensitivity analysis are similar to that from the original version of the manuscript. The model remains mainly sensitive to the temperature parameters related to the MPB growth rate and, to a lesser extent, to the light saturation parameters (Ek) and the half saturation constant for light use (KE).

In addition, we took into account the referee #2 comment by varying in the model one biological constant at once in the range of reported values found in the literature. The relationships between the biological constants and the simulated annual MPB production are presented in a new figure (Fig. R2, attached to the answer to referees). It results from the new analysis that the simulated annual MPB production is also mainly sensitive to Topt, Ek et KE. In addition, the MPB production collapses when Topt is higher than 25 °C. As Tmax remained constant when Topt varied, this new analysis highlights the role of Tamp as in the Monte-Carlo analysis. The two analyses hence show very similar results.

3. "Eq. A7 should include S(mud_to_water). Also, the heating/cooling of the water column should be related to the instantaneous water depth. I can't find this in the equations. Are these just issues with the representation in the manuscript, or is the heat balance model flawed? This should be corrected. " "It is suggested (p. 6, l.

10) that the mud temperature (fig 4) closely follows the air temperature (fig 3). This is difficult to see. Please include the air temperature in fig 4 for better comparison. " "The point above triggers the question if a full mud temperature model is necessary. This question could be easily addressed by driving the microphytobentos model with the air temperature (or air temperature when exposed and water temperature when submerged) and comparing the results with the reference simulation."

We took into account the referees #1 and #2 comments regarding the sediment temperature model. Based on Guarini et al. (1997) and Phizacklae (1987), we modified the physical model by setting a rapid equilibrium between mud surface temperature and the temperature of the overlying water layer, i.e. the simulated mud surface temperature is now set to water temperature during immersion periods. In the new model run, the root mean squared error (RMSE) between the observed and simulated MST values for the two 2008 periods (RMSE=1.81 °C) is reduced twice compared to the previous model run (RMSE=3.98 °C). In addition, the Pearson's correlation coefficient is higher in the new model run (r=0.93) compared to the previous model run (r=0.76). The referees comments helped to significantly improve the model capability to simulate the MST.

Regarding the definition of the total water depth, we agree with referee #2 that the description on how it is considered in the model was not clear in the Appendix section of the original version of the manuscript. In the model, the total water depth is represented as two fractions (set by the alpha coefficient) of the whole water column (i.e. from the top to the sea bed). The alpha coefficient sets the top fraction of the total water depth that is influenced by the atmospheric forcings (i.e. equivalent to the mixed layer depth). The heat balance between water and air is resolved in the model within this top layer while the bottom layer set by the remaining fraction of the total water depth remains at the water temperature computed at the previous time step of the model run. The simulated water temperature of the whole water column results from the mixing between the two layers. We modified the description of the total water depth in the

model and provide more details in the Appendix A.

With respect to the air and mud temperature relationship, we modified the original sentence (p6, l21) to make it clearer: "The simulated temperature of surface mud followed the seasonal cycle of air temperature (Pearson's r = 0.85, p<0.05; Fig. 3d and Fig. 4)." Figure R3 shows the air-mud temperature difference. The mean difference computed from the absolute air-mud differences is 2.14±2.3 °C, which reflects the high differences between the air and mud temperature at the synoptic scale. As a consequence, the air temperature forcing cannot be used to constrain the MPB growth as it departs too much from the simulated mud temperature.

4. "I'm puzzled by the few sharp peaks in ingestion. Is this realistic behaviour or an artifact of the model? If the latter, could it be related to the exponent in Eq B11, which can change sign depending on the temperature? This seems odd from a mathematical perspective. Was this kind of behaviour of the equation envisaged/included in the range of values considered in the publication in which this relationship was proposed?"

Referees #2 and #3 pointed out possible issues regarding the simulated ingestion rate of P. ulvae. In order to improve the ingestion rate simulated in the model when the simulated mud surface temperature exceeds the optimal temperature for grazing, we formulated the ingestion-temperature relationship according to a Holling type III sigmoid mathematical function (see Gentleman et al., 2003), which accounts for the effect of mud temperature. The new equation is described in the Appendix:

"The individual ingestion rate IR (ng Chl a ind−1 h−1) of P. ulvae is calculated using a Holling type III sigmoid mathematical function accounting for the effect of mud temperature T (°C):

IR = IRmax * ( T^alphaZ / (T^alphaZ + ((ToptZ+10)/2)^alphaZ) * H(S − 0.5),

where ToptZ (°C) is the optimal temperature for grazing. IRmax is the maximal observed individual ingestion rate. alphaZ (no unit) is a curvature parameter. The right

part is a heaviside function H related to the biomass in the biofilm.

The maximal individual ingestion rate IRmax (ng Chl a ind−1 h−1) is calculated according to the formulation of Haubois et al. (2005) for adult snails. It depends on the total MPB biomass:

IR = 0.015 × (F + S)^1.72

The Chl a uptake rate is converted into carbon unit according to the C:Chl a ratio described previously. The term (F + S) is expressed in $\mu$g g dry sed−1. The biomass expressed in mg Chl a m−2 is converted into $\mu$g g dry sed−1 as follows: [Chla] ($\mu$g g dry sed-1 ) =[Chla]^1.2605 (mg Chl a m−2) / rho_S * thickness_sed,

where rho_S is the sediment bulk density in g l−1 and thickness_sed is the sediment thickness i.e. 1 cm. The Chl a concentration is scaled by the exponent 1.2605 in order to reach a maximal observed ingestion rate of 385 ng Chl a ind−1 h−1 (Coehlo et al., 2011) when the Chl a concentration converges towards a maximal observed value (300 mg Chl a m−2, Guarini (1998))."

With the new grazing formulation, an increase of the simulated mud surface temperature towards the optimal temperature for grazing results into an increase of the ingestion rate until it reaches a plateau at its maximal value. This maximal value is determined by the simulated MPB biomass within the first cm of sediment according to the relationship of Haubois et al., (2005). In contrast with the original model run, the new grazing mathematical function dampens the sharp peaks of ingestion and, as such, is more realistic with respect to previous works (Blanchard et al., 2000; Haubois et al., 2005; Pascal et al., 2008).

5. "p. 7, l. 30-p. 8, l. 2. 4xsignificant. I disagree. These differences are not significant, because the model mean is within the confidence interval of the observations."

In the manuscript, we give the mean±standard deviation. The data distributions are skewed and asymmetric. This is the reason why we used an appropriate nonparametric test (i.e. Mann-Whitney) instead of the confidence interval. The non-parametric analysis tests the means taking into account the skewness and asymmetry of the data.

6. "p. 8, light limitation. The definition is confusing. Also during the night, light is the limiting factor. Please use the full 24 hr period, not just daylight hours to represent this."

As mentioned by the referee #2, MPB is also light-limited during the night in the model. However, as the MPB production occurs only during the daytime emersion periods, we computed the limitation terms that constrain primary production only during the daytime emersion periods.

7. "Discussion. The authors provide a substantial number of numeric comparisons with published results throughout the discussion. This information is very difficult to digest in this way. Please compile a table of all these data/values/references, and present as part of the results."

We agree with the referee #2 comment. To make the discussion clearer, we included a new table in the Results section.

8. "Table 1. I'm not sure if figures are allowed within a table - check journal require-ments. This table doesn't seem to contain new information compared with the text (appendix B1). Ensure there is no duplication (delete table?)."

We agree with the referee comment that the differential equations appear both in the Appendix B and within Table 1. Nevertheless, Table 1 provides a clear and synthetic view of the simulated ecological processes that may provide the reader a rapid under-standing of the model. In turn, the Appendix B section provides more details on the mathematical functions used in the differential equations shown in Table 1. The journal editorial support confirmed that figures can be inserted within a table.

9. "Appendix B1. dZ/dt is identical in the three cases. Please print only once. Also B4 is identical to B2 except for the formulation for tau - find an alternative way to present

this without duplication."

In each case, the differential equations are mathematically and numerically linked to each others as each scalar refers to other scalars. This is why they must be all shown in each presented case to help the reader clearly understand how the biomass flows between the model compartments. In addition, it permits the reader to focus on one case independently of the others.

10. "p. 1, l. 14. export flux: from, to? "

The sentence was modified as follows: "The model ability to infer on biotic and abiotic mechanisms driving the seasonal MPB dynamics could open the door to a new assessment of the export flux of biogenic matter from the coast to the open ocean and, more generally, of the contribution of productive intertidal biofilms to the coastal carbon cycle."

11. "p. 4, l. 7. The reference to fig 3 occurs before the first ref to fig 2. Swap figures."

The two figures were swapped.

12. "p. 5, l. 31. This sentence is unclear. "

To make it clearer, the sentence was modified as follows: "The variable SâĹŮ represents the S compartment that incorporated the S instantaneous production of biomass, which was directly transferred to the F compartment (mg chl a m$-2$)."

13. "p. 6, l. 13. This sentence is unclear. "

The sentence was modified as follows: "We performed a sensitivity analysis to quantify how simultaneous variations of key biological constants might impact the simulated MPB production. "

14. "p. 6, l. 15. First use, write out the names of the variables. Why these - there are many others (Table A3)?"

[Figure]

The names of the constants was added: "A Monte-Carlo fixed sampling method (Hammersley and Handscomb, 1964) was used to randomly select values of the temperature optimum for photosynthesis (Topt), the temperature maximum for photosynthesis (Tmax), the optimal temperature for grazing (ToptZ), the shape parameter of the temperature related grazing (alphaZ), the light saturation parameter (Ek) and the half-saturation constant for light use (KE) within observed ranges (Table 3)."

We selected these biological constants, because they are direct inputs in the mathematical functions that enter in the calculation of the simulated MPB production rate and P. ulvae ingestion rate. Other biological parameters as beta and pbMAX were not included in the sensitivity analysis since they vary seasonally in the model.

15. "p. 6, l. 23-24. This sentence is unclear. "

By seasonal amplitude we mean the difference between the maximum value and the minimum value for the time period considered.

The sentence was modified as follows: "The amplitude (i.e. the difference between the maximum and the minimum value) of the simulated mud temperature was higher in summer-fall (32.1 °C) than in winter-spring (18.1 °C). "

16. "Fig 5. Label graphs. Also plot 'original data' in the main figure for better comparison. Rephrase caption to make it clear what these original data are. "

We overlaid the original data on the two panel of the figure. The figure caption was modified as follows: "Seasonal cycle of the 2008: a) Normalized difference vegetation index (NDVI) and b) simulated daily maximum of the MPB biomass in the biofilm. Original extracted data (black circles) are overlaid. The black full lines represent the original extracted data regularized and filtered with running medians (window size = 7). NDVI is calculated at the pixel corresponding to the study site. Phases are determined according to the amplitude of the sign change of the second order derivative."

17. "p. 7. l. 2-5. Fall bloom. This seems less evident in the 'original data'? Is that true

and if so why? "

In fall, less satellite scenes were available than in spring and summer. Nevertheless, NDVI estimates retrieved in fall showed higher values in autumn than in late summer suggesting a moderate fall bloom. Such a pattern is also simulated by the model. In addition, such a fall bloom is also evidenced in Fig. 6a by the simulated MPB biomass within the sediment first cm.

18. "Figure 9. The white colour is missing from the legends. For graph b, there is no grey. Is this actually the case or an issue with the figure? It seems that in graph b, T_opt was plotted, not T_opt_z? "

Panels a and b represent the time periods during which either mud surface temperature or grazing is the most limiting term for MPB growth in the model. While on panel a the time periods in white represent the light limitation periods, they correspond in panel b to periods during which grazing is non-limiting.

The new figure based on results of the improved model run now presents grey colour.

To make it clearer, the figure caption was modified as follows: "Seasonal cycle of the MPB biomass (green full line), and time occurrence and duration of the daily limiting term in 2008: a) time periods when MPB is limited by mud temperature, b) time periods when MPB is limited by grazing. Mud temperature data are averaged over the daytime emersion period. The dashed vertical lines delimit the 3 phases shown in Fig. 5. The red colour on vertical bars indicates a match between temperature- and grazing-limited time periods."

19. "p. 9, l. 17. key: why are these 'key' (and how is that defined)?"

We selected these biological constants, because they are direct inputs in the mathematical functions that enter in the calculation of the simulated MPB production rate and P. ulvae ingestion rate. We modified the sentence as follows: "A total of 10000 model runs (N) was performed, in which a set of biological constants ($T_{opt}$, $T_{max}$, $T_{optZ}$, alphaZ, Ek and KE) were randomly selected within reported observed ranges (Table 3). These biological constants were chosen, because they are direct inputs in the mathematical functions that enter in the calculation of the simulated MPB production rate and P. ulvae ingestion rate."

20. "p. 12, l. 29. Here, a section starts on salinity (it's not entirely clear to me where this ends). This is the first mention of salinity, and as far as I understand salinity is not represented in the model. So this paragraph seems a bit out of place. Either delete, or argue why salinity was not included in the model in Methods, and then move this bit to a separate heading in the discussion.

Even if the salinity is not explicitly represented as a forcing variable in the physical model, we discussed the detrimental effect on the MPB cells and growth rate of high salinity levels induced by a strong heating of the mud surface and subsequent high temperatures. This is the reason why this part is developped in the Discussion section 4.2 on the role of mud surface temperature on MPB.

We modified the end of this paragraph (p. 13, l 8-11) to make it clearer:

" As the detrimental effects of high salinity levels is not explicitly accounted for in the model as a forcing variable, they are implicitly accounted for through temperature-related mechanisms, i.e. a Topt value lower than values reported in the literature (Table 5). Such an approach overestimates the thermo-inhibition process and, as such, promotes low PP rates that implicitly reproduces in the model the detrimental effects of desiccation on the microphytobenthic cells."

21. "p. 30, l. 31-35. This contradicts statements in Results."

If this comment refers to p. 14 l. 31-35, P. ulvae grazing is considered as limiting in the Results section only when the amount of MPB biomass grazed by P. ulvae is higher than the amount of biomass produced by MPB in the model. As such, a significant effect of grazing is simulated during 12 days in 2008. However, grazing can impact the

MPB biomass even when it is not the most limiting term for MPB growth. We show on a new figure (Fig. R4, attached to the answer to referees) the simulated MPB total biomass with and without P. ulvae grazing in the model. It clearly shows that grazing, even if it is not the most limiting term, impacts the MPB dynamics during the whole summer. As suggested in the literature, the MPB biomass is much lower in summer in presence than in the absence of P. ulvae (Sahan et al., 2007; Weerman et al., 2011).

22. "Figure 8. Why does the vertical axis start at 5? The plot seems to suggests that this truncates the data in mid-summer? "

The vertical axis was extended to range from 0 to 23h.

23. "emersion (is it emergence?) and immersion are easily confused, please use exposure and submergence."

We understand that the two terms can be easily confused. However, they are commonly accepted and widely used and accepted amongst the community (e.g. Admiraal and Peletier, 1980; Underwood and Kromkamp, 1999).

24. Figure 1. the font size used for latitude and longitude may be too small.

The font size was increased.

Captions of attached figures:

Fig. R1: Daily averaged in situ MPB biomass sampled in the sediment 1st cm at the study station on the Brouage mudflat in 2008 (black full dots), 2012 (grey full dots) and 2013 (blue full dots). Error bars correspond to the standard deviation.

Fig. R2: Univariate sensitivity analysis of the simulated MPB annual production to: a) the temperature optimum for MPB growth (Topt); b) the temperature maximum for MPB growth (Tmax); c) the light saturation parameter (Ek); d) the half-saturation constant for light use (KE); e) the optimal temperature for grazing (ToptZ); f) the shape parameter of the temperature related grazing (alphaZ). r is the Spearman's correlation coefficient
(asterisk inform when p < 0.05) and N is the number of tested values for each biological constant.

Fig. R3: Difference between the observed air temperature and the simulated mud surface temperature (°C) in 2008.

Fig. R4: Seasonal cycle of the MPB simulated biomass in the 1st cm of sediment in the presence (red full line) and in the absence of P. ulvae (red dashed line) in 2008.
* * *
[Figure]

**Fig. 1.**

[Figure]

**Fig. 2.**

[Figure]

[Figure]

**Fig. 3.**

[Figure]

**Fig. 4.**

---

## Author Comment (AC3) · 18 Sep 2018

We gratefully thank referee #3 for her/his constructive comments with respect to our manuscript. In order to improve the manuscript with respect to these comments, we amended the manuscript as suggested by the referee wherever it was possible.

The modifications made in the manuscript are based on a new model run, which includes the model improvements suggested by the three reviewers. They include the mud temperature model, the P. ulvae grazing mathematical formulation and the setting of the mean time spent by a MPB cell at the sediment surface. As a result, the simulated data presented here are modified compared to the initial version of the

manuscript.

1. "The only outcome that I found did not fit my preconceived understanding on MPB dynamics was that Peringia grazing actually only had a significant effect on very few days over the summer. That is a surprise." "P8 L30 and P15,L1 This is the one area I found surprising, given the number of published accounts of strong inverse correlations between Peringia (Hydrobia) abundance and biomass on NW European mudflats. Particularly when the authors have said in an earlier paragraph that during phase 2 light was limiting, which would make the biomass response even more susceptible to being grazed down? How convinced are the authors that this is a true situation, or is the model not capturing the real impact of grazers during this phase?"

In the model, P. ulvae grazing is considered as limiting in the Results section only when the amount of MPB biomass grazed by P. ulvae is higher than the amount of biomass produced by MPB in the model. As such, a significant effect of grazing is simulated during 12 days in 2008. However, grazing can impact the MPB biomass even when it is not the most limiting term for MPB growth. We show on a new figure (Fig. R4, attached to the answer to referees) the simulated MPB total biomass with and without P. ulvae grazing in the model. It clearly shows that grazing, even if it is not the most limiting term, impacts the MPB dynamics during the whole summer. As suggested in the literature, the MPB biomass is much lower in summer in presence than in the absence of P. ulvae (Sahan et al., 2007; Weerman et al., 2011).

2. "I think the authors need to validate their model using some other data sets, perhaps from some of the other mudflat systems that they have (and are) working on within the Atlantic / Channel seaboard, or resolved at finer temporal scales to demonstrate the robustness of the assumptions under pinning the model. After all, if the model works on one mudflat, it ought to be applicable to other similar systems, and this would really demonstrate its value to others workers in the field."

The lack of validation data was pointed out by the three referees. We agree with this

comment. Located 1.7 km from the shore our study site is remote. It is, however, the most studied site in the area but the sampling variables and protocols vary from year-to-year. We hence made the choice to use 2008 data from the French national project VASIREMI as it is unique in the area in terms of space and time coincident in situ measurements of both physical (sediment temperature) and biological (MPB and grazer biomass) variables during two contrasting seasons. In addition, high resolution atmospheric and oceanic forcings required to constrain the model are available for 2008.

To cope with the lack of data, we used two datasets of in situ MPB Chl a concentration available for the same station. The two datasets cover the spring, summer and winter seasons in 2012 and 2013. We added a new Figure (R1, attached to the answer to referees) to show the MPB seasonal cycle in terms of Chl a concentration based on the 2008, 2012 and 2013 data.

We added a new sentence in the Materials and Methods section (2.1.1) as follows: "In addition to the 2008 dataset, we used data of in situ MPB Chl a concentration collected within the 1st cm of sediment at the same station in April 19 – 22, 2012, July 05, 2012, November 14, 2012, February 11, 2013 and April 10, 2013). The sampling protocol is fully detailed in Lavergne et al. (2017)."

The new figure (R1, attached to the answer to referees) aims at showing the observed seasonal cycle of MPB Chl a at our study site based on the data available, i.e. a 3-year dataset (2008, 2012, 2013). A new paragraph was added in the Results section (3.2) as follows:

"Two distinct periods were identified from in situ Chl a measurements in the sediment first cm (Fig. R1). The observed seasonal cycle of Chl a was characterised by a spring bloom and by a decrease of Chl a concentration in summer. Given the few available measurements in autumn, the seasonal MPB dynamics at this season remained uncertain.The maximum of Chl a concentration reached during the spring bloom ranged

between 234 and 306 mg Chl a m-2, which is consistent with the concentration simulated by the model in 2008."

Applying the model to other intertidal systems requires year- and site-specific atmospheric and oceanic forcings along with multiparametric data to initiate and validate the model, which is not trivial to set up. However, we agree with the referee that the model portability to other mudflats should be envisaged as it would provide support to the model predicting capacity.

3. "Overall, what are the error terms around the modelled responses? The figures show some significant error terms in the existing field data, but no errors around the model outcomes."

In situ data include replicates at a same sampling time, which permits to compute the standard deviation around the mean. Such an approach is not possible with the model as a unique solution is estimated at each time step by the way of the numerical integration. A numerical model is by nature a mathematical approximation of a true state. As such, it will always depart from a true solution, which is difficult to quantify as it depends on the model complexity and the number of degrees of freedom. Some uncertainty is first introduced in the model through the quality of the atmospheric and oceanic forcings. In addition, the model relies on the choice of mathematical functions and constants, which is based, however, on a theoretical background gathered from observations in the field and/or laboratory. The choice of the parameters values and functions also introduces some uncertainty in the numerical estimates. A way of quantifying this uncertainty and the relevance of a model structure is to perform a sensitivity analysis. We present in the manuscript such an analysis, reinforced by our response to the comment made by the referee #2. It results that the model is sensitive to the choice of the temperature- and light-related constants. More data, including remote sensing data, will be further required to quantitatively assess an error around the model predictions. Nevertheless, the model/data comparison we show in our study and that uses time-limited but time-coincident situ data covering physical (mud temperature) and bio-

logical (MPB and P. ulvae) variables brings some confidence to a reasonable predictive capability of the coupled model.

4. "P3, L15. "in the light of current knowledge. . ..role still unclear". I think there is a very extensive set of literature on the roles of abiotic and biotic factors for MPB dynamics, so this statement portrays a false sense of uncertainty. "

We modified the sentence as follows: "The role of each individual abiotic or biotic factor involved in the MPB short term dynamics is well documented (eg. Feuillet-Girard, 1994; Admiraal, 1977; Vieira et al., 2016; Blanchard and Cariou-Le Gall, 1994; Barranguet et al., 1998; Light and Beardall, 2001; Pniewski et al., 2015; Barnett et al., 2015; Blanchard et al., 1997; Cartaxana et al., 2015; Coelho et al., 2009; Weerman et al., 2011; Pinckney et al., 2003; Admiraal et al., 1983; Montagna et al., 1995; Blanchard et al., 2002; Dupuy et al., 2014). However, and in the light of the current knowledge, the quantitative contribution of combined factors in the seasonal MPB dynamics remains uncertain."

5. "Figure 5 is an important figure. It needs to be made clear in the legend that this refers to S*. Why when the NVDI signal varies by over 100% in the course of the year, does the S* value only vary by at most 6-7%. Though the "pattern" looks the same (what is the correlation or correspondence between the two annual cycles?), the order of magnitude of change does not. How can this be, when they are assumed to be measuring the same thing?"

With respect to the satellite data, we agree with the referee #3 that the remotely-sensed NDVI and simulated Chl a concentration data cannot be quantitatively compared as they are not the same variable. However, the Spearman's correlation coefficient between the NDVI and the simulated Chl a concentration in the biofilm is 0.58 ($p < 0.05$). The NDVI/simulated Chl a relationship is therefore qualitatively reliable and can inform on the MPB seasonality. At a constant Chl a concentration, the Chl a pigments would absorb more light in summer than in winter because of the package effect. The

remotely-sensed NDVI would hence be expected to be higher in summer than in winter for a same biomass. However, based on field measurements, the NDVI is observed to be higher in winter (March) than in summer suggesting that the package effect of the Chl a pigments has no influence on the NDVI seasonality.

Furthermore, the simulated biofilm saturates quickly in terms of biomass at the sediment surface. Such a pattern therefore tempers short terms variations of the MPB dynamics at the sediment surface retrieved by the NDVI.

6. "P7, L9 onwards. The variable Ts is dependent on overall biomass, but then the outcomes of this seem counter-intuitive to what we know about biofilms and cell microcycling. Cells appear to spend the time they need at the surface to photosynthesise and accumulate enough carbon, while minimising their risk of photodamage. So each cell spending 54 minutes at the surface during January and August, while only 12 minutes in April, appears to be an outcome of an underlying assumption about biomass, rather than an understanding about diatom photophysiology and behaviour?"

The mathematical formulation was chosen to introduce an effect of carrying capacity in the simulated MPB dynamics. However, we agree with the referee #3 that, in terms of photophysiology and light requirements for the photosynthesis and inorganic carbon fixation, the mathematical formulation used in the model is counter-intuitive. To that respect, we replaced the initial mathematical formulation of Ts by the one from Guarini et al., (2008), which assumes a constant Ts value (Ts=1 h) over the year.

7. "P7, L20, clarify if this is the assumed intrinsic growth rate?

The simulated growth rate is not the intrinsic growth rate. It is obtained from the product between the simulated production rate in mg C mg Chl a-1 h-1 and the simulated Chla:C ratio to get the production rate in mg Chl a mg Chl a-1 h-1 or h-1. As such, the simulated growth rate does not include loss terms and is hence a gross growth rate. The sentence was hence modified as follows: "The annual mean of the MPB gross growth rate simulated within the biofilm was $0.25 \pm 0.07$ d $-1$ with a range of values

between 0.05 d −1 and 0.41 d −1 ."

8. "P8, L14 onwards. This section appears to be saying that during the summer periods, the biofilms are light limited, because there are longer days? If this just a mathematical artefact? After all, an individual cell only needs some many quanta of light to meet its photosynthetic requirements, and with variable migration, lower biomass and longer days, why would individual cells be light limited? "

We agree that individual cells are supposed to meet their photosynthetic requirements more easily in summer than in winter. In the model, the simulated light limitation takes into account the effect of low tides occurrence over the daytime periods (i.e. variable light levels) and the temperature conditions (i.e. optimal or not compared to the temperature optimum for MPB growth).

On the first hand, light is limiting in the model during daytime emersion periods in summer when the daytime emersion periods occur early/late in the daytime period during neap tides. The simulated MPB migrates towards the sediment surface but is exposed to low light levels during dawn and dusk compared to spring tides conditions when the emersion periods occur in the middle of the day at high light levels.

On the other hand, the simulated light limitation during daytime emersion periods in summer also relies on the simulated mud surface temperature. Despite favourable light levels during daytime emersion periods, the simulated mud surface temperature can be close to the temperature optimum for MPB growth and can hence promote microphytobenthic growth in relatively low light conditions.

The text was modified as follows: "In phase 2, light was the most limiting factor (60%, Table 2). The increasing daytime duration allowed MPB to grow on two daytime emersion periods at the beginning and at the end of the daytime period during neap tides. However, the simulated MPB is exposed to relatively low light levels during dawn and dusk compared to spring tides conditions, when the emersion periods occur in the middle of the day at relatively high light levels."

9. "P13, L6, see Steele et al. Biofouling 30, 987 – 998 for a detailed study of EPS and desiccation on diatom photosynthetic capacity

We thank the referee #3 for the reference. The positive effects of EPS on diatoms is much more developed than in the previous cited reference. We hence replaced it by that of Steele et al. (2014).

10. "P18, L3. What happens if a resuspension element is included in the model (Dupuy et al gives 3%, Blanchard et al 2006, in In J. Kromkamp [ed.], Functioning of microphytobenthos in estuaries: Proceedings of the microphytobenthos symposium, Amsterdam, The Netherlands, August 2003. Royal Netherlands Academy of Arts and Sciences, and Hanlon et al. 2006 Limnol. Oceanogr. 51: 79-93, provide other values, and de Jonge and van Beusekom (op. cit) provide some critical wind speeds)?"

We agree with referees #1 and #3 that the resuspension process is not explicitly detailed in the manuscript. As there are no data available of current velocity on the sea bed in 2008 at our study station, we did not infer on hydrodynamically-related resuspension processes of MPB. In the model, we assumed a constant rate of MPB cells resuspended during immersion periods. During immersion periods, the generic loss term ($vF$, 0.003 h-1) includes the chronic resuspension and MPB senescence processes. During emersion periods, the loss term is lower ($mF$, 0.001 h-1) as it only represents the MPB senescence.

We agree with referee #1 that some text on how much MPB might be exported and on the justifications about our mathematical formulation were lacking in the discussion. In the original version of the manuscript, we included this part in the perspectives section, because we currently work on the MPB resuspension mechanisms and related physical processes to be further included in the model. The referee #1 comment was hence taken into account by adding some text in the Results and Discussion sections.

The Results section was modified as follows: "In the model, the linear loss terms applied to the MPB biomass simulated within the first cm translated into a yearly averaged

resuspension of $1.7 \pm 0.3\%$ of the averaged MPB biomass in the sediment 1st cm during high tides. Over the year, 25 % of the simulated MPB production during low tides was resuspended, which corresponded to a total annual resuspension of 31.6 g C m-2. "

The Discussion section was modified as follows: "The short-term daily dynamics of MPB is regulated by resuspension events (Blanchard et al., 2002). The intensity of resuspension of MPB into the water column can be either chronic or catastrophic according to the flow velocity and the sediment stabilization (Marrioti and Fagherrazzi, 2012). Catastrophic events can locally resuspend all the MPB biomass as the resuspended sediment layer is thicker than the vertical distribution of MPB biomass (Marrioti and Fagherrazzi, 2012). The repeated occurrences of such events over several days can shape the seasonal cycle of MPB by lowering the biomass of photosynthetically competent MPB. In their model, Guarini et al., (2008) introduced a chronic resuspension of all the MPB biomass remaining in the biofilm when tidal floods occurred. In their parametrization, the MPB biomass remains at the sediment surface according to a mean time spent at the surface (equivalent to tau in our study). In our model, the chronic resuspension of MPB biomass is formulated by a linear loss term of the MPB biomass within the first cm (0.002 h-1). In the absence of MPB biomass deposition, the total simulated MPB biomass which is resuspended into the water column represents 25% of the simulated benthic MPB annual production. Such a value supports the fact that benthic MPB production contributes significantly to the pelagic food web (Perissinotto et al., 2003; Krumme et al., 2008). In the light of the work of Marrioti and Fagherrazzi (2012), resuspension and deposition are key mechanisms that need to be related to fauna bioturbation, sediment characteristics (e.g. nature and stabilization) and hydrodynamics (Marrioti and Fagherrazzi, 2012). Such an approach requires the availability of waves and current data to estimate the bed shear stress and modulate the intensity of resuspension (from chronic to catastrophic events), which are not available at our study site for 2008. "

Bed shear stress induced by physical factors (i.e. current and wave orbital velocities, bed roughness) and sediment stabilization control the resuspension of sediment and associated MPB (Tolhurst et al., 2003). Dupuy et al. (2014) showed that benthic diatoms are resuspended at a friction velocity of 3 cm s-1. This critical friction velocity for diatoms resuspension can be lower than the tidal current velocity without the action of wind during spring tides on sheltered mudflats according to the simulations of Le Hir et al., (2000). In addition, the impact of grazing activity by benthic deposit feeders has to be considered. Bioturbation generates a fluff layer of sediment-organic matrix, which is resuspended at a lower critical friction velocity (1 cm s-1 for P. ulvae bioturbated fluff layer; Orvain et al., 2004). Chronic resuspension of MPB cells can therefore occur with no wind, as shown by Guarini et al. (2008). Furthermore, waves and winds interact with tidal currents. When considering an angle between the waves and the current direction for the bed shear stress calculation (Soulsby, 1997), the wave forcing can be antagonistic, synergetic or neutral on the current bed shear stress according to the tidal and the wave conditions. Resuspension can hence occur without any action of winds.

Captions of attached figures:

Fig. R1: Daily averaged in situ MPB biomass sampled in the sediment 1st cm at the study station on the Brouage mudflat in 2008 (black full dots), 2012 (grey full dots) and 2013 (blue full dots). Error bars correspond to the standard deviation.

Fig. R2: Univariate sensitivity analysis of the simulated MPB annual production to: a) the temperature optimum for MPB growth (Topt); b) the temperature maximum for MPB growth (Tmax); c) the light saturation parameter (Ek); d) the half-saturation constant for light use (KE); e) the optimal temperature for grazing (ToptZ); f) the shape parameter of the temperature related grazing (alphaZ). r is the Spearman's correlation coefficient (asterisk inform when $p < 0.05$) and N is the number of tested values for each biological constant.

Fig. R3: Difference between the observed air temperature and the simulated mud

surface temperature (°C) in 2008.

Fig. R4: Seasonal cycle of the MPB simulated biomass in the 1st cm of sediment in the presence (red full line) and in the absence of P. ulvae (red dashed line) in 2008.
* * *
[Figure]

**Fig. 1.**

**Fig. 2.**

[Figure]

**Fig. 3.**

[Figure]

**Fig. 4.**

---

## Author Response (AR1)

Raphaël Savelli
LIttoral ENvironnement et Sociétés (LIENSs) - UMR 7266
Université de la Rochelle, Bâtiment ILE
2, rue Olympe de Gouges
17000 La Rochelle
France
Email: raphael.savelli1@univ-lr.fr

La Rochelle, October 12, 2018

Object: Revision of the manuscript bg-2018-325

Dear Editor,

Please find attached a revised version of the manuscript entitled "On biotic and abiotic drivers of the microphytobenthos seasonal cycle in a temperate intertidal mudflat: a modelling study" by R. Savelli, C. Dupuy, L. Barillé, A. Lerouxel, K. Guizien, A. Philippe, P. Bocher, P. Polsenaere, and V. Le Fouest. Based on your recommendations about the manuscript # bg-2018-325, we thank you to allow us providing a revised version of the manuscript which takes into account all the reviewers' comments. Following your request, we provide below a point-by-point response to the reviewers and a list of all relevant changes made in the manuscript. The changes corresponding to the major comments of reviewers are coloured in red in the revised version.

Yours sincerely,

Raphaël Savelli

**Answer to referee #1**

We gratefully thank referee #1 for his constructive comments with respect to our manuscript. In order to improve the manuscript with respect to these comments, we amended the manuscript as suggested by the referee wherever it was possible.

The modifications made in the manuscript are based on a new model run, which includes the model improvements suggested by the three reviewers. They include the mud temperature model, the *P. ulvae* grazing mathematical formulation and the setting of the mean time spent by a MPB cell at the sediment surface. As a result, the simulated data presented here are modified compared to the initial version of the manuscript.

**1. "The temperature inhibition value used to achieve this outcome had to be 'tuned' to match the data and the temperature inhibition used for MPB was lower than in previous studies, which was concerning. How sensitive is the annual pattern to this temp? Could you show a run where Topt is ~20 C?"**

The different microphytobenthic $T_{opt}$ estimation techniques and contrasting *in situ versus* laboratory conditions explain the range of values reported in the literature (from 20 to 30 °C). In the model run where $T_{opt}$ was set to 20°C the seasonality of MPB is very similar to that depicted by the model ran with $T_{opt}$ value set to 18°C. However, while the simulated MPB biomass was maintained during winter 2008-2009 with a $T_{opt}$ value of 18 °C, it vanished with a $T_{opt}$ value of 20 °C. Such a result suggests that at our study site a $T_{opt}$ value of 18 °C allows MPB to cope with cold winter conditions and relatively low mud surface temperatures.

In the discussion section (4.2), additional detrimental effects of high temperature on epipelic diatoms are discussed. Those effects are not considered in laboratory-based estimates of $T_{opt}$ given in Blanchard et al. (1997) because the authors worked with a suspension of MPB cells. In 2008, PAM measurements showed a lower photosynthetic activity in July than in February on the Brouage mudflat (pers. com. S. Lefebvre and J. Lavaud). The authors suggest that micromigrations are lowered by high mud surface temperature (and so by the pore water evaporation and resulting increase of salinity) and that MPB is unable to avoid harmful light conditions at mud surface temperature even lower than 20 °C. MPB cells are therefore photo-inhibited via high temperature-related processes. As photo-inhibition is not accounted for in the Production-Irradiance relationship used in the model, a lower

$T_{opt}$ value than the range estimated in the literature is a way to account for the negative feedback of photo-inhibition on the photosynthetic rate triggered by high temperature-related processes within the sediment.

We modified the Discussion section 4.2 (p. 14, l 6-10) to make it clearer:

"As the detrimental effects of high salinity levels is not explicitly accounted for in the model, they are implicitly accounted for through temperature-related mechanisms, i.e. an optimum of temperature for MPB growth lower than values reported in the literature (Table 6). Such an approach overestimates the thermo-inhibition process and, as such, promotes low PP rates that implicitly reproduces in the model the detrimental effects of desiccation on the microphytobenthic cells."

**2. "Another weakness of the study was that it had a rather limited data set for validation. I am surprised the authors were not able to find a study site with a larger time series of data for grazers and directly measured MPB biomass. "**

The lack of validation data was pointed out by the three referees. We agree with this comment. Located 1.7 km from the shore our study site is remote. It is, however, the most studied site in the area but the sampling variables and protocols vary from year-to-year. We hence made the choice to use 2008 data from the French national project VASIREMI as it is unique in the area in terms of space and time coincident *in situ* measurements of both physical (sediment temperature) and biological (MPB and grazer biomass) variables during two contrasting seasons. In addition, high resolution atmospheric and oceanic forcings required to constrain the model are available for 2008.

To cope with the lack of data, we used two datasets of *in situ* MPB Chl *a* concentration available for the same station. The two datasets cover the spring, summer and winter seasons in 2012 and 2013. We added a new Figure (Fig. 5) to show the MPB seasonal cycle in terms of Chl *a* concentration based on the 2008, 2012 and 2013 data.

We added a new sentence in the Materials and Methods section (p 4, l 24-26) as follows: "In addition to the 2008 dataset, we used data of *in situ* MPB Chl *a* concentration collected within the 1st cm of sediment at the same station in April 19 - 22, 2012, July 05, 2012, November 14, 2012, February 11, 2013 and April 10, 2013. The sampling protocol is fully detailed in Lavergne et al. (2017)."

The new figure (Fig. 5) aims at showing the observed seasonal cycle of MPB Chl *a* at our study site based on the data available, i.e. a 3-year dataset (2008, 2012, 2013). A new paragraph was added in the Results section (p 7, l 5-8) as follows:

"Based on *in situ* Chl *a* measurements sampled in the sediment $1^{st}$ cm in 2008 and 2012-2013, the observed seasonal cycle of Chl *a* was characterised by concentrations increasing from February to April, when the values were the highest (234-306 mg chl *a* $m^{-2}$ ; Fig. 5). Then the Chl *a* concentration decreased to reach a seasonal minimum in July (48-191 mg chl *a* $m^{-2}$ ; Fig. 5)."

And p 7, l 10-12:

"The simulated seasonal maximum and minimum of MPB biomass during spring and summer were consistent with the observations of 2008 and 2012-2013 (Fig. 5)."

**3. "Given the importance of the physical model, it was also disappointing that there was very little temperature data, which is very easy to collect. The limited data available also seemed to disagree in pattern and magnitude a lot more than I would have regarded as acceptable for a physical parameter."**

We took into account the referees #1 and #2 comments regarding the sediment temperature model. Based on Guarini et al. (1997) and Phizacklae (1987), we modified the physical model by setting a rapid equilibrium between mud surface temperature and the temperature of the overlying water layer, i.e. the simulated mud surface temperature is now set to water temperature during immersion periods. In the new model run, the root mean squared error (RMSE) between the observed and simulated MST values for the two 2008 periods (RMSE=1.81 °C) is reduced twice compared to the previous model run (RMSE=3.98 °C). In addition, the Pearson's correlation coefficient is higher in the new model run (r=0.93) compared to the previous model run (r=0.76). The referees comments helped to significantly improve the model capability to simulate the MST.

**4. "In the conclusions and perspectives part of the manuscript, it then goes on to say resuspension is not included in the model. Could the authors please clarify what the generic loss term is?"**

We agree with referees #1 and #3 that the resuspension process is not explicitly detailed in the manuscript. As there are no data available

of current velocity on the sea bed in 2008 at our study station, we did not infer on hydrodynamically-related resuspension processes of MPB. In the model, we assumed a constant rate of MPB cells resuspended during immersion periods. During immersion periods, the generic loss term ($v_F$, 0.003 h$^{-1}$) includes the chronic resuspension, MPB senescence processes and the grazing by subsurface deposit feeders. During emersion periods, the loss term is lower ($m_F$, 0.001 h$^{-1}$) as it only represents the MPB senescence and the grazing by subsurface deposit feeders.

**5. "I also suggest the discussion of resuspension be included earlier on in the discussion, rather than being raised right at the end. I would also like to see this discussion expanded a little. At present, it really only addresses possible PP by MPB during resuspension, it does not address how much MPB might be exported. The possible resuspension and export of MPB should be discussed and omission from the model justified."**

We agree with referee #1 that some text on how much MPB might be exported and on the justifications about our mathematical formulation were lacking in the discussion. In the original version of the manuscript, we included this part in the perspectives section, because we currently work on the MPB resuspension mechanisms and related physical processes to be further included in the model. The referee #1 comment was hence taken into account by adding some text in the Results and Discussion sections.

The Results section was modified as follows (p 8, l 6-9): "In the model, a linear loss term representing the resuspension process was applied to the MPB biomass simulated within the 1$^{st}$ cm of sediment (F compartment). In average over a high tide, 1.7 ± 0.3 % of the simulated MPB biomass was resuspended. With respect to primary production, 25 % of the MPB primary production simulated during low tides was resuspended, which corresponded in the model to a total annual resuspension of 31.6 g C m$^{-2}$."

The Discussion section was modified as follows (p 12, l 19-34): "The short-term daily dynamics of MPB is also regulated by resuspension events (Blanchard et al., 2002). The intensity of resuspension of MPB into the water column can be either chronic or catastrophic according to the flow velocity and the sediment stabilisation (Mariotti and Fagherazzi, 2012). Catastrophic events can locally resuspend all the MPB biomass as the resuspended sediment layer is thicker than the vertical distribution of MPB biomass (Mariotti and Fagherazzi, 2012). The

repeated occurrences of such events over several days could contribute to shape the seasonal cycle of MPB by lowering the biomass of photosynthetically competent MPB. In their model, Guarini et al. (2008) introduced a chronic resuspension of all the MPB biomass remaining in the biofilm when tidal floods occurred. In their parametrisation, the MPB biomass remains at the sediment surface according to a mean time spent at the surface (equivalent to τs in our study). In our study, the chronic resuspension of MPB biomass is formulated by a linear loss term of the MPB biomass within the 1$^{st}$ cm (0.002 h$^{-1}$). In the absence of MPB biomass deposition, the total simulated MPB biomass that is resuspended into the water column represents 25 % of the simulated benthic MPB annual production. Such a value brings support to a significant contribution of the benthic MPB production to the pelagic food web (Perissinotto et al., 2003; Krumme et al., 2008). In light of the work of Mariotti and Fagherazzi (2012), resuspension and deposition are key mechanisms that need to be related to fauna bioturbation, sediment characteristics (e.g. nature and stabilisation) and hydrodynamics (Mariotti and Fagherazzi, 2012). Such an approach requires the availability of waves and current data to estimate the bed shear stress and modulate the intensity of resuspension (from chronic to catastrophic events), which are not available at our study site for 2008."

**6. "Is it possible the loss of biomass is just resuspension on a few windy days?"**

Bed shear stress induced by physical factors (i.e. current and wave orbital velocities, bed roughness) and sediment stabilisation control the resuspension of sediment and associated MPB (Tolhurst et al., 2003). Dupuy et al. (2014) showed that benthic diatoms are resuspended at a friction velocity of 3 cm s$^{-1}$. This critical friction velocity for diatoms resuspension can be lower than the tidal current velocity without the action of wind during spring tides on sheltered mudflats according to the simulations of Le Hir et al., (2000). In addition, the impact of grazing activity by benthic deposit feeders has to be considered. Bioturbation generates a fluff layer of sediment-organic matrix, which is resuspended at a lower critical friction velocity (1 cm s$^{-1}$ for *P. ulvae* bioturbated fluff layer; Orvain et al., 2004). Chronic resuspension of MPB cells can therefore occur with no wind, as shown by Guarini et al. (2008). Furthermore, waves and winds interact with tidal currents. When considering an angle between the waves and the current direction for the bed shear stress calculation (Soulsby, 1997), the wave forcing can be antagonistic, synergetic or neutral on the current bed shear stress according to the tidal and the wave conditions. Resuspension can hence occur without any action of winds.

**7. "The manuscript was generally well written and the ideas well constructed. There were a few spelling and grammatical issues. I have noted a few below, but it would be easiest for the authors to use a spell checker to find these."**

Spelling and grammatical issues have been checked according to the referee comment.

**8. "Pg 4 l19. Could clarify a little better that (1st cm) means 1st cm of sediment."**

It has been changed by (1$^{st}$ cm of sediment) (p 4, l 19-20).

**9. "Pg 8 line 32 onwards. This is a little confusing. first it is stated that grazing is mostly limiting, then it says days where MPB biomass consumed was larger than that produced occurred only 8.7% of the time."**

The original sentence is confusing and the text was improved to make this part clearer. The improvements made on the grazing formulation in the model led to new model outputs with respect to the original version of the manuscript. In order to better highlight the impact of grazing, we replaced the panel b of figure 10 by the simulated time series of *P. ulvae* daily ingestion rate with the MPB daily production rate overlaid (Fig. 10b). The impact of grazing with respect to MPB primary production is now shown more clearly. The entire paragraph was modified as follows (p 9, l 14-23):

"With respect to grazing, the simulated biomass grazed by *P. ulvae* was compared to the simulated MPB biomass produced over the daytime emersion period (Fig. 10b). During phase 1, the ingested MPB biomass exceeded the MPB PP during 11 days (Fig. 10b). The simulated peaks of ingestion rate during these days varied between ~ 20 and 90 ng chl *a* ind$^{-1}$ h$^{-1}$ (Fig. 8c), which was consistent with the reported values from laboratory measurements (0.75-385 ng chl *a* ind$^{-1}$ h$^{-1}$ ; Table 3). The daily-averaged *P. ulvae* ingestion:MPB production ratio was lower but more variable in phase 1 (0.31 ± 0.45) than in phase 2 (0.47 ± 0.18) (Fig. 10b). Phase 1 was characterised by a marked and synoptic impact of grazing at high MPB biomass levels. By contrast, grazing was moderate but more sustained in phase 2. Grazing contributed with thermo-inhibition to maintain relatively low levels of MPB biomass (Fig. 10). As the ingestion rate of *P. ulvae* was related to the MPB biomass and to the MST, the peaks of grazing simulated in spring resulted from both the high MPB biomass

accumulated during the bloom and the MST close to the temperature optimum for grazing by *P. ulvae* ($T_{optz}$)."

**10. "Pg 15 line 13. This implies a very high growth efficiency (13.63/15.8). Can this be correct? Or do they graze other food sources too?"**

The initial sentence is confusing: "The combined effect of grazing and thermo-inhibition translates into 22 % (5.9 g C) of the simulated annual MPB PP channelled towards *P. ulvae* gross secondary production. The simulated *P. ulvae* gross secondary production is 27 g C $m^{-2}$ $y^{-1}$."

On the 5.9 g C grazed by *P. ulvae* during events of combined grazing and thermo-inhibition only 55% (assimilation rate γ, Annexe B) was assimilated by *P. ulvae*, which corresponded to 3.2 g C. With respect to the new model outputs, the sentence was not relevant anymore and was hence removed.

**11. "Figure 5. I don't understand why there is a small plot (original data) for biomass. If this is from the model, it should be more continuous? Or perhaps you have only extracted the same days as the NDVI data? Why not show all the data?"**

For the model/satellite comparison, the MPB biomass simulated in the biofilm was extracted for the same days than the NDVI data. In order to filter both NDVI and simulated MPB biomass data, the resulting time series were first regularised to obtain regular time intervals between all points. Small plots were hence showed to illustrate the shape of the original time series before the numerical treatment. The figure 7 and the caption were improved according to the comment of referee #2 to make it clearer (small plots were removed and the extracted points were overlaid on the main plots).

**12. "Figure 9b. Caption could specify days dominated by grazing pressure when temperature is greater than grazing optimum (T > Toptz). I found this a little hard to understand at first."**

The information given in the legend was included and detailed in the figure caption to make it clearer but it only concerns the Fig. 10a, as the figure 10b was changed.

**New cited references:**

Blanchard, G., Simon-Bouhet, B., and Guarini, J.-M.: Properties of the dynamics of intertidal microphytobenthic biomass, Journal of the Marine Biological Association of the United Kingdom, 82, 1027–1028, 2002.

Lavergne, C., Agogué, H., Leynaert, A., Raimonet, M., De Wit, R., Pineau, P., Bréret, M., Lachaussée, N., and Dupuy, C.: Factors influencing prokaryotes in an intertidal mudflat and the resulting depth gradients, Estuarine, Coastal and Shelf Science, 189, 74–83, 2017.

**Answer to referee #2**

We gratefully thank referee #2 for her/his constructive comments with respect to our manuscript. In order to improve the manuscript with respect to these comments, we amended the manuscript as suggested by the referee wherever it was possible.

The modifications made in the manuscript are based on a new model run, which includes the model improvements suggested by the three reviewers. They include the mud temperature model, the *P. ulvae* grazing mathematical formulation and the setting of the mean time spent by a MPB cell at the sediment surface. As a result, the simulated data presented here are modified compared to the initial version of the manuscript.

**1. "The two short periods of in-situ data are not sufficient to constrain the seasonal cycle. With these data, many other potential modelled seasonal cycles, including constant values (straight lines) could be equally valid results. The authors mention a monthly data set of chlorophyll observations from the same mud flat covering March 1992 to February 1993. A simulation for this period should be included and compared with the observations. The remote-sensing data are not really a substitute for this because they may have their own issues, and in the current manuscript are not the same variable."**

The lack of validation data was pointed out by the three referees. We agree with this comment. Located 1.7 km from the shore our study site is remote. It is, however, the most studied site in the area but the sampling variables and protocols vary from year-to-year. We hence made the choice to use 2008 data from the French national project VASIREMI as it is unique in the area in terms of space and time coincident *in situ* measurements of both physical (sediment temperature) and biological (MPB and grazer biomass) variables during two contrasting seasons. In addition, high resolution atmospheric and oceanic forcings required to constrain the model are available for 2008.

To cope with the lack of data, we used two datasets of *in situ* MPB Chl *a* concentration available for the same station. The two datasets cover the spring, summer and winter seasons in 2012 and 2013. We added a

new Figure (Fig. 5) to show the MPB seasonal cycle in terms of Chl *a* concentration based on the 2008, 2012 and 2013 data.

We added a new sentence in the Materials and Methods section (p 4, l 24-26) as follows: "In addition to the 2008 dataset, we used data of in situ MPB Chl *a* concentration collected within the 1st cm of sediment at the same station in April 19 - 22, 2012, July 05, 2012, November 14, 2012, February 11, 2013 and April 10, 2013. The sampling protocol is fully detailed in Lavergne et al. (2017)."

The new figure (Fig. 5) aims at showing the observed seasonal cycle of MPB Chl *a* at our study site based on the data available, i.e. a 3-year dataset (2008, 2012, 2013). A new paragraph was added in the Results section (p 7, l 5-8) as follows:

"Based on *in situ* Chl *a* measurements sampled in the sediment 1st cm in 2008 and 2012-2013, the observed seasonal cycle of Chl *a* was characterised by concentrations increasing from February to April, when the values were the highest (234-306 mg chl *a* m$^{-2}$ ; Fig. 5). Then the Chl *a* concentration decreased to reach a seasonal minimum in July (48-191 mg chl *a* m$^{-2}$; Fig. 5)."

And p 7, l 10-12:

"The simulated seasonal maximum and minimum of MPB biomass during spring and summer were consistent with the observations of 2008 and 2012-2013 (Fig. 5)."

With respect to the satellite data, we agree with the referee #2 that the remotely-sensed NDVI and simulated Chl *a* concentration data cannot be quantitatively compared as they are not the same variable. However, the Spearman's correlation coefficient between the NDVI and the simulated Chl *a* in the biofilm is 0.58 (p < 0.05). The NDVI/simulated Chl *a* relationship is therefore qualitatively reliable and can inform on the MPB seasonality. At a constant Chl *a* concentration, the Chl *a* pigments would absorb more light in summer than in winter because of the package effect. The remotely-sensed NDVI would hence be expected to be higher in summer than in winter for a same biomass. However, based on field measurements, the NDVI is observed to be higher in winter (March) than in summer suggesting that the package effect of the Chl *a* pigments has no influence on the NDVI seasonality.

**2. "The sensitivity analysis leaves me puzzled. Why calculate correlation coefficients which assume linearity if the model equations are clearly non-linear? For which areas of the varied parameter space does the primary production collapse? Why? Why was this subset of parameters selected and not others? Instead of randomised monte-carlo simulations and questionable statistics, I would think a series of graphs where primary production is plotted as a function of a varied parameter (with others set to reference values) would be much more instructive, or should in the least be used to analyse what happens in the monte-carlo simulations. Such an approach could even be used to restrict the range of variation of the parameters such that not so many simulations collapse (if the collapsing simulations occur towards the extremities of the parameter space). These new ranges could then also be critically compared with the ranges reported in the literature."**

We took into account the referee #2 comment about the non-linearity in the model between the biological constants and MPB annual production. We performed the same sensitivity analysis but we used the Spearman's correlation coefficient, which is adapted for non-linear relationships. The results of the sensitivity analysis are similar to that from the original version of the manuscript. The model remains mainly sensitive to the temperature parameters related to the MPB growth rate and, to a lesser extent, to the light saturation parameters ($E_k$) and the half saturation constant for light use ($K_E$).

In addition, we took into account the referee #2 comment by varying in the model one biological constant at once in the range of reported values found in the literature. The relationships between the biological constants and the simulated annual MPB production are presented in a new figure (Fig. 12). It results from the new analysis that the simulated annual MPB production is also mainly sensitive to $T_{opt}$, $E_k$ et $K_E$. In addition, the MPB production collapses when $T_{opt}$ is higher than 25 °C. As $T_{max}$ remained constant when $T_{opt}$ varied, this new analysis highlights the role of $T_{amp}$ as in the Monte-Carlo analysis. The two analyses hence show very similar results. The both are presented in Results section 3.5 (p 9-11).

**3. "Eq. A7 should include S(mud_to_water). Also, the heating/cooling of the water column should be related to the instantaneous water depth. I can't find this in the equations. Are these just issues with the representation in the manuscript, or is the heat balance model flawed? This should be corrected. "**
**"It is suggested (p. 6, l. 10) that the mud temperature (fig 4) closely follows the air temperature (fig 3). This is difficult to see. Please include the air temperature in fig 4 for better comparison. "**
**"The point above triggers the question if a full mud temperature model is necessary. This question could be easily addressed by driving the microphytobentos model with the air temperature (or air temperature when exposed and water temperature when submerged) and comparing the results with the reference simulation."**

We took into account the referees #1 and #2 comments regarding the sediment temperature model. Based on Guarini et al. (1997) and Phizacklae (1987), we modified the physical model by setting a rapid equilibrium between mud surface temperature and the temperature of the overlying water layer, i.e. the simulated mud surface temperature is now set to water temperature during immersion periods. In the new model run, the root mean squared error (RMSE) between the observed and simulated MST values for the two 2008 periods (RMSE=1.81 °C) is reduced twice compared to the previous model run (RMSE=3.98 °C). In addition, the Pearson's correlation coefficient is higher in the new model run (r=0.93) compared to the previous model run (r=0.76). The referees comments helped to significantly improve the model capability to simulate the MST.

Regarding the definition of the total water depth, we agree with referee #2 that the description on how it is considered in the model was not clear in the Appendix section of the original version of the manuscript. In the model, the total water depth is represented as two fractions (set by the alpha coefficient) of the whole water column (i.e. from the top to the sea bed). The alpha coefficient sets the top fraction of the total water depth that is influenced by the atmospheric forcings (i.e. equivalent to the mixed layer depth). The heat balance between water and air is resolved in the model within this top layer while the bottom layer set by the remaining fraction of the total water

depth remains at the water temperature computed at the previous time step of the model run. The simulated water temperature of the whole water column results from the mixing between the two layers. We modified the description of the total water depth in the model and provide more details in the Appendix A (p 19, l 23-28 and p 20, l 1-13).

With respect to the air and mud temperature relationship, we modified the original sentence (p 6, l 31) to make it clearer: "The simulated MST followed the seasonal cycle of air temperature (Pearson's r = 0.85, p-value < 0.05; Fig. 2d and Fig. 4)." Figure R1 (attached to the answer to referee #2) shows the air-mud temperature difference. The mean difference computed from the absolute air-mud differences is 2.14±2.3 °C, which reflects the high differences between the air and mud temperature at the synoptic scale. As a consequence, the air temperature forcing cannot be used to constrain the MPB growth as it departs too much from the simulated mud temperature.

**4. "I'm puzzled by the few sharp peaks in ingestion. Is this realistic behaviour or an artifact of the model? If the latter, could it be related to the exponent in Eq B11, which can change sign depending on the temperature? This seems odd from a mathematical perspective. Was this kind of behaviour of the equation envisaged/included in the range of values considered in the publication in which this relationship was proposed?"**

Referees #2 and #3 pointed out possible issues regarding the simulated ingestion rate of *P. ulvae*. In order to improve the ingestion rate simulated in the model when the simulated mud surface temperature exceeds the optimal temperature for grazing, we formulated the ingestion-temperature relationship according to a sigmoid mathematical function, which accounts for the effect of mud temperature. The new equation is described in the Appendix B3 (p 23, l 1-13):

"The individual ingestion rate (ng chl *a* ind$^{-1}$ h$^{-1}$ ) by *P. ulvae* is calculated using a sigmoid mathematical function accounting for the effect of mud temperature T (°C):

$$IR = IR_{max} * ( T^{\alpha Z} / (T^{\alpha Z} + ((T_{optZ}+10)/2)^{\alpha Z}),$$

where $T_{optZ}$ (°C) is the optimal temperature for grazing. $IR_{max}$ is the maximal observed individual ingestion rate. $\alpha_Z$ (no unit) is a curvature parameter. The maximal individual ingestion rate $IR_{max}$ (ng chl *a* ind$^{-1}$ h$^{-1}$) is calculated according to the formulation of Haubois et al. (2005) for adult snails. $IR_{max}$ depends on the total MPB biomass:

The maximal individual ingestion rate $IR_{max}$ (ng chl *a* ind$^{-1}$ h$^{-1}$) is calculated according to the formulation of Haubois et al. (2005) for adult snails. It depends on the total MPB biomass:

$$IR_{max} = 0.015 \times (F + S)^{1.72}$$

The Chl *a* uptake rate is converted into carbon unit according to the C:Chl *a* ratio described previously. The term (F + S) is expressed in µg chl *a* g dry sed$^{-1}$. The biomass expressed in mg chl *a* m$^{-2}$ is converted into µg chl *a* g dry sed$^{-1}$ as follows:

[Chl*a*] (µg chl *a* g dry sed$^{-1}$) =[Chl*a*]$^{1.2605}$ (mg chl *a* m$^{-2}$) / $\rho_S$ * thickness$_{sed}$,

where $\rho_S$ is the sediment bulk density in g l$^{-1}$ and thickness$_{sed}$ is the sediment thickness i.e. 1 cm. The Chl *a* concentration is scaled by the exponent 1.2605 in order to reach a maximal observed ingestion rate of 385 ng chl *a* ind$^{-1}$ h$^{-1}$ (Coelho et al., 2011) when the Chl *a* concentration converges towards a maximal observed value (300 mg chl *a* m$^{-2}$, Guarini, 1998)."

With the new grazing formulation, an increase of the simulated mud surface temperature towards the optimal temperature for grazing results into an increase of the ingestion rate until it reaches a plateau at its maximal value. This maximal value is determined by the simulated MPB biomass within the first cm of sediment according to the relationship of Haubois et al., (2005). In contrast with the original model run, the new grazing mathematical function dampens the sharp peaks of ingestion and, as such, is more realistic with respect to previous works (Blanchard et al., 2000; Haubois et al., 2005; Pascal et al., 2008).

**5. "p. 7, l. 30-p. 8, l. 2. 4xsignificant. I disagree. These differences are not significant, because the model mean is within the confidence interval of the observations."**

In the manuscript, we give the mean±standard deviation. The data distributions are skewed and asymmetric. This is the reason why we used an appropriate non-parametric test (i.e. Mann-Whitney) instead of the confidence interval. The non-parametric analysis tests the means taking into account the skewness and asymmetry of the data.

**6. "p. 8, light limitation. The definition is confusing. Also during the night, light is the limiting factor. Please use the full 24 hr period, not just daylight hours to represent this."**

As mentioned by the referee #2, MPB is also light-limited during the night in the model. However, as the MPB production occurs only during the daytime emersion periods, we computed the limitation terms that constrain primary production only during the daytime emersion periods.

**7. "Discussion. The authors provide a substantial number of numeric comparisons with published results throughout the discussion. This information is very difficult to digest in this way. Please compile a table of all these data/values/references, and present as part of the results."**

We agree with the referee #2 comment. To make the discussion clearer, we included a new table (Table 3) in the Results section and we compared the simulated values with the literature (p 7, l 32-34 and p 8, l 1-5):

"Biological parameters simulated by the model were compared to observed ranges reported in the literature (Table 3). The yearly-averaged value of S∗ simulated by the model ($27.2 \pm 3.6$ mg chl $a$ m$^{-2}$) was in agreement with the value given by Herlory et al. ($24 \pm 5$ mg chl $a$ m$^{-2}$ ; 2004). The yearly-averaged MPB gross growth rate ($\mu$) simulated within the biofilm was $0.25 \pm 0.07$ d$^{-1}$ with values ranging between $0.05$ d$^{-1}$ and $0.41$ d$^{-1}$, which compared to the observed growth rate ($0.035$-$0.86$ d$^{-1}$; Table 3). In the model, the MPB growth rate was related to the C:Chl $a$ ratio (see Eq. B8 in Appendix B2). The simulated C:Chl $a$ ratio (16 and 75.5 g C g chl $a^{-1}$ ) varied between the observed range ($18.7$-$80$ g C g chl $a^{-1}$ ; Table 3). The simulated annual and daily MPB PP rates ($127$ g C m$^{-2}$ y$^{-1}$ and $369 \pm 281$ mg C m$^{-2}$ d$^{-1}$ , respectively) were also consistent with the reported in situ estimates ($142 \pm 82$ g C m$^{-2}$ y$^{-1}$ and $690 \pm 682$ mg C m$^{-2}$ d$^{-1}$, respectively)."

**8. "Table 1. I'm not sure if figures are allowed within a table - check journal requirements. This table doesn't seem to contain new information compared with the text (appendix B1). Ensure there is no duplication (delete table?)."**

We agree with the referee comment that the differential equations appear both in the Appendix B and within Table 1. Nevertheless, Table 1 provides a clear and synthetic view of the simulated ecological processes that may provide the reader a rapid understanding of the model. In turn, the Appendix B section provides more details on the mathematical functions used in the differential equations shown in Table 1. The journal editorial support confirmed that figures can be inserted within a table.

**9. "Appendix B1. dZ/dt is identical in the three cases. Please print only once. Also B4 is identical to B2 except for the formulation for tau - find an alternative way to present this without duplication."**

In each case, the differential equations are mathematically and numerically linked to each others as each scalar refers to other scalars. This is why they must be all shown in each presented case to help the reader clearly understand how the biomass flows between the model compartments. In addition, it permits the reader to focus on one case independently of the others.

**10. "p. 1, l. 14. export flux: from, to? "**

The sentence was modified as follows (p 1, l 13-15): "The model ability to infer on biotic and abiotic mechanisms driving the seasonal MPB dynamics could open the door to a new assessment of the export flux of biogenic matter from the coast to the open ocean and, more generally, of the contribution of productive intertidal biofilms to the coastal carbon cycle."

**11. "p. 4, l. 7. The reference to fig 3 occurs before the first ref to fig 2. Swap figures."**

The two figures were swapped.

**12. "p. 5, l. 31. This sentence is unclear. "**

To make it clearer, the sentence was modified as follows (p 6, l 4-5): "The variable S∗ represented the S compartment that incorporated the S instantaneous production of biomass (mg chl *a* m$^{-2}$), which is directly transferred to F."

**13. "p. 6, l. 13. This sentence is unclear. "**

The sentence was modified as follows (p 6, l 19-20): "We performed a sensitivity analysis to quantify how simultaneous variations of key biological constants might impact the simulated MPB production."

**14. "p. 6, l. 15. First use, write out the names of the variables. Why these - there are many others (Table A3)?"**

The names of the constants was added (p 6, l 20-23): "A Monte-Carlo fixed sampling method (Hammersley and Handscomb, 1964) was used to randomly select values of the temperature optimum for photosynthesis ($T_{opt}$), the temperature maximum for photosynthesis ($T_{max}$), the optimal temperature for grazing ($T_{optZ}$), the shape parameter of the temperature related grazing ($\alpha_Z$), the light saturation parameter ($E_k$) and the half-saturation constant for light use ($K_E$) within observed ranges (Table 2)."

We selected these biological constants, because they are direct inputs in the mathematical functions that enter in the calculation of the simulated MPB production rate and *P. ulvae* ingestion rate. Other biological parameters as $\beta$ and $p^b_{MAX}$ were not included in the sensitivity analysis since they vary seasonally in the model.

**15. "p. 6, l. 23-24. This sentence is unclear. "**

By seasonal amplitude we mean the difference between the maximum value and the minimum value for the time period considered.

The sentence was modified as follows (p 6, l 33 and p 7, l 1-2): " The amplitude (i.e. the difference between the seasonal maximum and the minimum value) of the simulated mud temperature was higher in summer-fall (32.1 °C) than in winter-spring (18.1 °C). "

**16. "Fig 5. Label graphs. Also plot 'original data' in the main figure for better comparison. Rephrase caption to make it clear what these original data are. "**

We overlaid the original data on the two panel of the figure. The figure 7 caption was modified as follows: "Seasonal cycle of the 2008 **(a)** Normalised difference vegetation index (NDVI), and **(b)** simulated daily maximum of the MPB biomass (mg chl *a* m$^{-2}$) in the biofilm. Original extracted data (black circles) are overlaid. The black full lines represent the original extracted data regularised and filtered with running medians (window size = 7). The NDVI was calculated at the pixel corresponding to the study site. Phases were determined according to the amplitude of the sign change of the second order derivative."

**17. "p. 7. l. 2-5. Fall bloom. This seems less evident in the 'original data'? Is that true and if so why? "**

In fall, less satellite scenes were available than in spring and summer. Nevertheless,  NDVI estimates retrieved in fall showed higher values in autumn than in late summer suggesting a moderate fall bloom. Such a pattern is also simulated by the model. In addition, such a fall bloom is also evidenced in Fig. 6a by the simulated MPB biomass within the sediment first cm.

**18. "Figure 9. The white colour is missing from the legends. For graph b, there is no grey. Is this actually the case or an issue with the figure? It seems that in graph b, T_opt was plotted, not T_opt_z? "**

Panels a represent the time periods during which mud surface temperature is the most limiting term for MPB growth in the model. On panel a the time periods in white represent the light limitation periods. In order to better highlight the impact of grazing, we replaced the panel b of figure 10 by the simulated time series of *P. ulvae* daily ingestion rate with the MPB daily production rate overlaid (Fig. 10b). The impact of grazing with respect to MPB primary production is now shown more clearly.

To make it clearer, the figure caption was modified as follows: "Seasonal cycle of the 2008 **(a)** simulated MPB biomass (mg chl *a* m$^{-2}$, green full line) with time occurrence and duration of the simulated

temperature limitation term when daily-averaged mud surface temperature during emersion periods was lower (grey vertical bars) or higher (black vertical bars) than the optimal temperature for MPB growth ($T_{opt}$), and **(b)** simulated daily primary production rate (mg C m$^{-2}$ d$^{-1}$) and *P. ulvae* ingestion rate (mg C m$^{-2}$ d$^{-1}$). The dashed vertical lines delimit the 3 phases shown in Fig. 7."

**19. "p. 9, l. 17. key: why are these 'key' (and how is that defined)?"**

We selected these biological constants, because they are direct inputs in the mathematical functions that enter in the calculation of the simulated MPB production rate and *P. ulvae* ingestion rate. We modified the sentence as follows (p 9, l 32-34): "A total of 10,000 model runs (N) was performed, in which a set of biological constants ($T_{opt}$, $T_{max}$, $T_{optZ}$ , $\alpha_Z$ , $E_k$ and $K_E$) was randomly selected within the reported observed ranges (Table 2). These biological constants were chosen, because they were direct inputs in the mathematical functions used in the calculation of the simulated MPB production rate and *P. ulvae* ingestion rate."

**20. "p. 12, l. 29. Here, a section starts on salinity (it's not entirely clear to me where this ends). This is the first mention of salinity, and as far as I understand salinity is not represented in the model. So this paragraph seems a bit out of place. Either delete, or argue why salinity was not included in the model in Methods, and then move this bit to a separate heading in the discussion.**

Even if the salinity is not explicitly represented as a forcing variable in the physical model, we discussed the detrimental effect on the MPB cells and growth rate of high salinity levels induced by a strong heating of the mud surface and subsequent high temperatures. This is the reason why this part is developped in the Discussion section 4.2 on the role of mud surface temperature on MPB.

We modified the end of this paragraph  (p. 14, l 6-10) to make it clearer:

"As the detrimental effects of high salinity levels is not explicitly accounted for in the model, they are implicitly accounted for through temperature-related mechanisms, i.e. an optimum of temperature for

MPB growth lower than values reported in the literature (Table 6). Such an approach overestimates the thermo-inhibition process and, as such, promotes low PP rates that implicitly reproduces in the model the detrimental effects of desiccation on the microphytobenthic cells."

**21. "p. 14, l. 31-35. This contradicts statements in Results."**

The improvements made on the grazing formulation in the model led to new model outputs with respect to the original version of the manuscript. In order to better highlight the impact of grazing, we replaced the panel b of figure 10 by the simulated time series of *P. ulvae* daily ingestion rate with the MPB daily production rate overlaid (Fig. 10b). The impact of grazing with respect to MPB primary production is now shown more clearly. The entire paragraph was modified as follows (p 9, l 14-23):

"With respect to grazing, the simulated biomass grazed by *P. ulvae* was compared to the simulated MPB biomass produced over the daytime emersion period (Fig. 10b). During phase 1, the ingested MPB biomass exceeded the MPB PP during 11 days (Fig. 10b). The simulated peaks of ingestion rate during these days varied between ~ 20 and 90 ng chl *a* ind$^{-1}$ h$^{-1}$ (Fig. 8c), which was consistent with the reported values from laboratory measurements (0.75-385 ng chl *a* ind$^{-1}$ h$^{-1}$ ; Table 3). The daily-averaged *P. ulvae* ingestion:MPB production ratio was lower but more variable in phase 1 (0.31 ± 0.45) than in phase 2 (0.47 ± 0.18) (Fig. 10b). Phase 1 was characterised by a marked and synoptic impact of grazing at high MPB biomass levels. By contrast, grazing was moderate but more sustained in phase 2. Grazing contributed with thermo-inhibition to maintain relatively low levels of MPB biomass (Fig. 10). As the ingestion rate of *P. ulvae* was related to the MPB biomass and to the MST, the peaks of grazing simulated in spring resulted from both the high MPB biomass accumulated during the bloom and the MST close to the temperature optimum for grazing by *P. ulvae* (T$_{optZ}$ )."

And we modified the Discussion section (p 15, l 24-26): "In the model, *P. ulvae* grazing exceeds the MPB PP mainly in spring (11 days of MPB biomass removal). *P. ulvae* depletes a substantial part of the MPB biomass accumulated during the spring bloom. After the bloom, a moderate but sustained grazing by *P. ulvae* adds to the effect of thermo-inhibition on the MPB dynamics."

**22. "Figure 8. Why does the vertical axis start at 5? The plot seems to suggests that this truncates the data in mid-summer? "**

The vertical axis of figure 9 was extended to range from 0 to 23h.

**23. "emersion (is it emergence?) and immersion are easily confused, please use exposure and submergence."**

We understand that the two terms can be easily confused. However, they are commonly accepted and widely used and accepted amongst the community (e.g. Admiraal and Peletier, 1980; Underwood and Kromkamp, 1999).

**24. Figure 1. the font size used for latitude and longitude may be too small.**

The font size was increased.

**New cited reference:**

Lavergne, C., Agogué, H., Leynaert, A., Raimonet, M., De Wit, R., Pineau, P., Bréret, M., Lachaussée, N., and Dupuy, C.: Factors influencing prokaryotes in an intertidal mudflat and the resulting depth gradients, Estuarine, Coastal and Shelf Science, 189, 74–83, 2017.

[Figure]

Fig. R1: Difference between the observed air temperature and the simulated mud surface temperature (°C) in 2008.

**Answer to referee #3**

We gratefully thank referee #3 for her/his constructive comments with respect to our manuscript. In order to improve the manuscript with respect to these comments, we amended the manuscript as suggested by the referee wherever it was possible.

The modifications made in the manuscript are based on a new model run, which includes the model improvements suggested by the three reviewers. They include the mud temperature model, the *P. ulvae* grazing mathematical formulation and the setting of the mean time spent by a MPB cell at the sediment surface. As a result, the simulated data presented here are modified compared to the initial version of the manuscript.

**1. "The only outcome that I found did not fit my preconceived understanding on MPB dynamics was that Peringia grazing actually only had a significant effect on very few days over the summer. That is a surprise."**
**"P8 L30 and P15,L1 This is the one area I found surprising, given the number of published accounts of strong inverse correlations between Peringia (Hydrobia) abundance and biomass on NW European mudflats. Particularly when the authors have said in an earlier paragraph that during phase 2 light was limiting, which would make the biomass response even more susceptible to being grazed down? How convinced are the authors that this is a true situation, or is the model not capturing the real impact of grazers during this phase?"**

The improvements made on the grazing formulation in the model led to new model outputs with respect to the original version of the manuscript. In order to better highlight the impact of grazing, we replaced the panel b of figure 10 by the simulated time series of *P. ulvae* daily ingestion rate with the MPB daily production rate overlaid (Fig. 10b). The impact of grazing with respect to MPB primary production is now shown more clearly. The entire paragraph was modified as follows (p 9, l 14-23):

"With respect to grazing, the simulated biomass grazed by *P. ulvae* was compared to the simulated MPB biomass produced over the daytime emersion period (Fig. 10b). During phase 1, the ingested MPB biomass exceeded the MPB PP during 11 days (Fig. 10b). The simulated peaks of ingestion rate during these days varied between ~ 20 and 90 ng chl *a* ind$^{-1}$ h$^{-1}$ (Fig. 8c), which was consistent with the reported values from laboratory measurements (0.75-385 ng chl *a* ind$^{-1}$ h$^{-1}$ ; Table 3). The daily-averaged *P. ulvae* ingestion:MPB production

ratio was lower but more variable in phase 1 (0.31 ± 0.45) than in phase 2 (0.47 ± 0.18) (Fig. 10b). Phase 1 was characterised by a marked and synoptic impact of grazing at high MPB biomass levels. By contrast, grazing was moderate but more sustained in phase 2. Grazing contributed with thermo-inhibition to maintain relatively low levels of MPB biomass (Fig. 10). As the ingestion rate of *P. ulvae* was related to the MPB biomass and to the MST, the peaks of grazing simulated in spring resulted from both the high MPB biomass accumulated during the bloom and the MST close to the temperature optimum for grazing by *P. ulvae* ($T_{optZ}$)."

And we modified the Discussion section (p 15, l 24-26): "In the model, *P. ulvae* grazing exceeds the MPB PP mainly in spring (11 days of MPB biomass removal). *P. ulvae* depletes a substantial part of the MPB biomass accumulated during the spring bloom. After the bloom, a moderate but sustained grazing by *P. ulvae* adds to the effect of thermo-inhibition on the MPB dynamics."

**2. "I think the authors need to validate their model using some other data sets, perhaps from some of the other mudflat systems that they have (and are) working on within the Atlantic / Channel seaboard, or resolved at finer temporal scales to demonstrate the robustness of the assumptions under pinning the model. After all, if the model works on one mudflat, it ought to be applicable to other similar systems, and this would really demonstrate its value to others workers in the field."**

The lack of validation data was pointed out by the three referees. We agree with this comment. Located 1.7 km from the shore our study site is remote. It is, however, the most studied site in the area but the sampling variables and protocols vary from year-to-year. We hence made the choice to use 2008 data from the French national project VASIREMI as it is unique in the area in terms of space and time coincident *in situ* measurements of both physical (sediment temperature) and biological (MPB and grazer biomass) variables during two contrasting seasons. In addition, high resolution atmospheric and oceanic forcings required to constrain the model are available for 2008.

To cope with the lack of data, we used two datasets of in situ MPB Chl *a* concentration available for the same station. The two datasets cover the spring, summer and winter seasons in 2012 and 2013. We added a new Figure (Fig. 5) to show the MPB seasonal cycle in terms of Chl *a* concentration based on the 2008, 2012 and 2013 data.

We added a new sentence in the Materials and Methods section (p 4, l 24-26) as follows: "In addition to the 2008 dataset, we used data of *in situ* MPB Chl *a* concentration collected within the 1$^{st}$ cm of sediment at the same station in April 19 - 22, 2012, July 05, 2012, November 14, 2012, February 11, 2013 and April 10, 2013. The sampling protocol is fully detailed in Lavergne et al. (2017)."

The new figure (Fig. 5) aims at showing the observed seasonal cycle of MPB Chl *a* at our study site based on the data available, i.e. a 3-year dataset (2008, 2012, 2013). A new paragraph was added in the Results section (p 7, l 5-8) as follows:

"Based on *in situ* Chl *a* measurements sampled in the sediment 1$^{st}$ cm in 2008 and 2012-2013, the observed seasonal cycle of Chl *a* was characterised by concentrations increasing from February to April, when the values were the highest (234-306 mg chl *a* m$^{-2}$ ; Fig. 5). Then the Chl *a* concentration decreased to reach a seasonal minimum in July (48-191 mg chl *a* m$^{-2}$ ; Fig. 5)."

And p 7, l 10-12:

"The simulated seasonal maximum and minimum of MPB biomass during spring and summer were consistent with the observations of 2008 and 2012-2013 (Fig. 5)."

Applying the model to other intertidal systems requires year- and site-specific atmospheric and oceanic forcings along with multiparametric data to initiate and validate the model, which is not trivial to set up. However, we agree with the referee that the model portability to other mudflats should be envisaged as it would provide support to the model predicting capacity.

**3. "Overall, what are the error terms around the modelled responses? The figures show some significant error terms in the existing field data, but no errors around the model outcomes."**

*In situ* data include replicates at a same sampling time, which permits to compute the standard deviation around the mean. Such an approach is not possible with the model as a unique solution is estimated at each time step by the way of the numerical integration. A numerical model is by nature a mathematical approximation of a true state. As such, it will always depart from a true solution, which is difficult to quantify as it depends on the model complexity and the number of degrees of freedom. Some uncertainty is first introduced in the model through the quality of the atmospheric and oceanic

forcings. In addition, the model relies on the choice of mathematical functions and constants, which is based, however, on a theoretical background gathered from observations in the field and/or laboratory. The choice of the parameters values and functions also introduces some uncertainty in the numerical estimates. A way of quantifying this uncertainty and the relevance of a model structure is to perform a sensitivity analysis. We present in the manuscript such an analysis, reinforced by our response to the comment made by the referee #2. It results that the model is sensitive to the choice of the temperature- and light-related constants. More data, including remote sensing data, will be further required to quantitatively assess an error around the model predictions. Nevertheless, the model/data comparison we show in our study and that uses time-limited but time-coincident situ data covering physical (mud temperature) and biological (MPB and P. ulvae) variables brings some confidence to a reasonable predictive capability of the coupled model.

**4. "P3, L15. "in the light of current knowledge. . ..role still unclear". I think there is a very extensive set of literature on the roles of abiotic and biotic factors for MPB dynamics, so this statement portrays a false sense of uncertainty. "**

We modified the sentence (p 3, l 15-20) as follows: "The role of each individual abiotic or biotic factor involved in the MPB short term dynamics is well documented (e.g. Admiraal, 1977; Admiraal et al., 1983; Blanchard and Cariou-Le Gall, 1994; Montagna et al., 1995; Blanchard et al., 1997; Feuillet-Girard et al., 1997; Barranguet et al., 1998; Light and Beardall, 2001; Blanchard et al., 2002; Pinckney et al., 2003; Coelho et al., 2009; Weerman et al., 2011; Dupuy et al., 2014; Pniewski et al., 2015; Barnett et al., 2015; Cartaxana et al., 2015; Vieira et al., 2016). However, and in light of the current knowledge, the quantitative contribution of combined factors in the seasonal MPB dynamics remains uncertain."

**5. "Figure 5 is an important figure. It needs to be made clear in the legend that this refers to S*. Why when the NVDI signal varies by over 100% in the course of the year, does the S* value only vary by at most 6-7%. Though the "pattern" looks the same (what is the correlation or correspondence between the two annual cycles?), the order of magnitude of change does not. How can this be, when they are assumed to be measuring the same thing?"**

With respect to the satellite data, we agree with the referee #3 that the remotely-sensed NDVI and simulated Chl $a$ concentration data cannot be quantitatively compared as they are not the same variable.

However, the Spearman's correlation coefficient between the NDVI and the simulated Chl $a$ concentration in the biofilm is 0.58 (p < 0.05). The NDVI/simulated Chl $a$ relationship is therefore qualitatively reliable and can inform on the MPB seasonality. At a constant Chl $a$ concentration, the Chl $a$ pigments would absorb more light in summer than in winter because of the package effect. The remotely-sensed NDVI would hence be expected to be higher in summer than in winter for a same biomass. However, based on field measurements, the NDVI is observed to be higher in winter (March) than in summer suggesting that the package effect of the Chl $a$ pigments has no influence on the NDVI seasonality.

Furthermore, the simulated biofilm saturates quickly in terms of biomass at the sediment surface. Such a pattern therefore tempers short terms variations of the MPB dynamics at the sediment surface retrieved by the NDVI.

**6. "P7, L9 onwards. The variable Ts is dependent on overall biomass, but then the outcomes of this seem counter-intuitive to what we know about biofilms and cell microcycling. Cells appear to spend the time they need at the surface to photosynthesise and accumulate enough carbon, while minimising their risk of photodamage. So each cell spending 54 minutes at the surface during January and August, while only 12 minutes in April, appears to be an outcome of an underlying assumption about biomass, rather than an understanding about diatom photophysiology and behaviour?"**

The mathematical formulation was chosen to introduce an effect of carrying capacity in the simulated MPB dynamics. However, we agree with the referee #3 that, in terms of photophysiology and light requirements for the photosynthesis and inorganic carbon fixation, the mathematical formulation used in the model is counter-intuitive. To that respect, we replaced the initial mathematical formulation of $\tau_s$ by the one from Guarini et al., (2008), which assumes a constant Ts value ($\tau_s$=1 h) over the year.

**7. "P7, L20, clarify if this is the assumed intrinsic growth rate?**

The simulated growth rate is not the intrinsic growth rate. It is obtained from the product between the simulated production rate in mg C mg chl $a^{-1}$ h$^{-1}$ and the simulated Chl $a$:C ratio to get the production rate in mg chl $a$ mg chl $a^{-1}$ h$^{-1}$ or h$^{-1}$. As such, the simulated growth rate does not include loss terms and is hence a gross growth rate.

The sentence was hence modified as follows (p 7, l 34 and p 8, l 1-2): "The yearly-averaged MPB gross growth rate (μ) simulated within the biofilm was 0.25 ± 0.07 $d^{-1}$ with values ranging between 0.05 $d^{-1}$ and 0.41 $d^{-1}$, which compared to the observed growth rate (0.035-0.86 $d^{-1}$; Table 3). "

**8. "P8, L14 onwards. This section appears to be saying that during the summer periods, the biofilms are light limited, because there are longer days? If this just a mathematical artefact? After all, an individual cell only needs some many quanta of light to meet its photosynthetic requirements, and with variable migration, lower biomass and longer days, why would individual cells be light limited? "**

We agree that individual cells are supposed to meet their photosynthetic requirements more easily in summer than in winter. In the model, the simulated light limitation takes into account the effect of low tides occurrence over the daytime periods (i.e. variable light levels) and the temperature conditions (i.e. optimal or not compared to the temperature optimum for MPB growth).

On the first hand, light is limiting in the model during daytime emersion periods in summer when the daytime emersion periods occur early/late in the daytime period during neap tides. The simulated MPB migrates towards the sediment surface but is exposed to low light levels during dawn and dusk compared to spring tides conditions when the emersion periods occur in the middle of the day at high light levels.

On the other hand, the simulated light limitation during daytime emersion periods in summer also relies on the simulated mud surface temperature. Despite favourable light levels during daytime emersion periods, the simulated mud surface temperature can be close to the temperature optimum for MPB growth and can hence promote microphytobenthic growth in relatively low light conditions.

The text was modified as follows (p 8, l 31-33 and p 9, l 1): "In phase 2, light was the most limiting factor (60 %, Table 4). The increasing daytime duration allowed MPB to grow on two daytime emersion periods at the beginning and at the end of the daytime period during neap tides (Fig. 9). However, the simulated MPB was exposed to relatively low light levels during dawn and dusk compared to spring tides conditions, when the emersion periods occurred in the middle of the day and at relatively high light levels (Fig. 9)."

**9. "P13, L6, see Steele et al. Biofouling 30, 987 – 998 for a detailed study of EPS and desiccation on diatom photosynthetic capacity**

We thank the referee #3 for the reference. The positive effects of EPS on diatoms is much more developed than in the previous cited reference. We hence replaced it by that of Steele et al. (2014) (p 14, l 4).

**10. "P18, L3. What happens if a resuspension element is included in the model (Dupuy et al gives 3%, Blanchard et al 2006, in In J. Kromkamp [ed.], Functioning of microphytobenthos in estuaries: Proceedings of the microphytobenthos symposium, Amsterdam, The Netherlands, August 2003. Royal Netherlands Academy of Arts and Sciences, and Hanlon et al. 2006 Limnol. Oceanogr. 51: 79-93, provide other values, and de Jonge and van Beusekom (op. cit) provide some critical wind speeds)?"**

We agree with referees #1 and #3 that the resuspension process is not explicitly detailed in the manuscript. As there are no data available of current velocity on the sea bed in 2008 at our study station, we did not infer on hydrodynamically-related resuspension processes of MPB. In the model, we assumed a constant rate of MPB cells resuspended during immersion periods. During immersion periods, the generic loss term ($\nu_F$, 0.003 $h^{-1}$) includes the chronic resuspension, the MPB senescence processes and the grazing by subsurface deposit feeders. During emersion periods, the loss term is lower ($m_F$, 0.001 $h^{-1}$) as it only represents the MPB senescence and the grazing by subsurface deposit feeders.

The Results section was modified as follows (p 8, l 6-9): "In the model, a linear loss term representing the resuspension process was applied to the MPB biomass simulated within the $1^{st}$ cm of sediment (F compartment). In average over a high tide, 1.7 ± 0.3 % of the simulated MPB biomass was resuspended. With respect to primary production, 25 % of the MPB primary production simulated during low tides was resuspended, which corresponded in the model to a total annual resuspension of 31.6 g C $m^{-2}$ ."

The Discussion section was modified as follows (p 12, l 19-34): "The short-term daily dynamics of MPB is also regulated by resuspension events (Blanchard et al., 2002). The intensity of resuspension of MPB into the water column can be either chronic or catastrophic according to the flow velocity and the sediment stabilisation (Mariotti and Fagherazzi, 2012). Catastrophic

events can locally resuspend all the MPB biomass as the resuspended sediment layer is thicker than the vertical distribution of MPB biomass (Mariotti and Fagherazzi, 2012). The repeated occurrences of such events over several days could contribute to shape the seasonal cycle of MPB by lowering the biomass of photosynthetically competent MPB. In their model, Guarini et al. (2008) introduced a chronic resuspension of all the MPB biomass remaining in the biofilm when tidal floods occurred. In their parametrisation, the MPB biomass remains at the sediment surface according to a mean time spent at the surface (equivalent to $\tau s$ in our study). In our study, the chronic resuspension of MPB biomass is formulated by a linear loss term of the MPB biomass within the 1$^{st}$ cm (0.002 h$^{-1}$ ). In the absence of MPB biomass deposition, the total simulated MPB biomass that is resuspended into the water column represents 25 % of the simulated benthic MPB annual production. Such a value brings support to a significant contribution of the benthic MPB production to the pelagic food web (Perissinotto et al., 2003; Krumme et al., 2008). In light of the work of Mariotti and Fagherazzi (2012), resuspension and deposition are key mechanisms that need to be related to fauna bioturbation, sediment characteristics (e.g. nature and stabilisation) and hydrodynamics (Mariotti and Fagherazzi, 2012). Such an approach requires the availability of waves and current data to estimate the bed shear stress and modulate the intensity of resuspension (from chronic to catastrophic events), which are not available at our study site for 2008."

Bed shear stress induced by physical factors (i.e. current and wave orbital velocities, bed roughness) and sediment stabilisation control the resuspension of sediment and associated MPB (Tolhurst et al., 2003). Dupuy et al. (2014) showed that benthic diatoms are resuspended at a friction velocity of 3 cm s$^{-1}$. This critical friction velocity for diatoms resuspension can be lower than the tidal current velocity without the action of wind during spring tides on sheltered mudflats according to the simulations of Le Hir et al., (2000). In addition, the impact of grazing activity by benthic deposit feeders has to be considered. Bioturbation generates a fluff layer of sediment-organic matrix, which is resuspended at a lower critical friction velocity (1 cm s$^{-1}$ for *P. ulvae* bioturbated fluff layer; Orvain et al., 2004). Chronic resuspension of MPB cells can therefore occur with no wind, as shown by Guarini et al. (2008). Furthermore, waves and winds interact with tidal currents. When considering an angle between the waves and the current direction for the bed shear stress calculation (Soulsby, 1997), the wave forcing can be antagonistic, synergetic or neutral on the current bed shear stress according to the tidal and the

wave conditions. Resuspension can hence occur without any action of winds.

[revised manuscript text omitted]
 | 0.21 | 0.13 | 0.11 | 0.11 | 0.15 | 0.22 | 0.26 | 0.23 | 0.10 | 0.23 | 0.19 | 0.15 |

---

## Author Response (AR2)

Raphaël Savelli
LIttoral ENvironnement et Sociétés (LIENSs) - UMR 7266
Université de la Rochelle, Bâtiment ILE
2, rue Olympe de Gouges
17000 La Rochelle
France
Email: raphael.savelli1@univ-lr.fr

La Rochelle, November 23, 2018

Object: Revision of the manuscript bg-2018-325

Dear Editor,

Please find attached a revised version of the manuscript entitled "On biotic and abiotic drivers of the microphytobenthos seasonal cycle in a temperate intertidal mudflat: a modelling study" by R. Savelli, C. Dupuy, L. Barillé, A. Lerouxel, K. Guizien, A. Philippe, P. Bocher, P. Polsenaere, and V. Le Fouest. Based on your recommendations about the manuscript # bg-2018-325, we thank you to allow us providing a revised version of the manuscript which takes into account all the reviewers' comments. Following your request, we provide below a point-by-point response to the reviewer #2 and a list of all relevant changes made in the manuscript. The changes corresponding to the minor comments of reviewers are coloured in red in the revised version.

Yours sincerely,

Raphaël Savelli

**Answer to referee #1, Dr. Perran Cook**

We gratefully thank Dr. Perran Cook for his constructive review of our manuscript. All the minor comments were taken into account.

**Answer to referee #2**

We gratefully thank referee #2 for her/his constructive comments with respect to our manuscript. In order to improve the manuscript with respect to these comments, we amended the manuscript as suggested by the referee wherever it was possible.

**"The most substantial remaining issue is the full sensitivity analysis as presented in Figure 11. It is unclear why the vast majority of the simulations resulted in vanishing primary production (one would not expect this from looking at Fig 12), and for which combinations/regions of the parameter settings. It would also be interesting to know for which combinations/regions of parameter settings the high and low ends of the sustained primary production cases were achieved."**

Figures 11 and 12 illustrate two different approaches used for the sensitivity analysis. MPB annual primary production (PP) is highly sensitive to MPB temperature parameters in both methods. In the Monte-Carlo sensitivity analysis, the large number of vanishing runs results from a specific combination of parameters (mostly MPB temperature parameters and, in to lesser extent, the light saturation parameter). In order to highlight in the Monte-Carlo sensitivity analysis the parameters combinations responsible for vanishing runs, we added a parallel coordinates graph (Figure 12 in the new revised version). Figure 12 shows the 10,000 parameters combinations and the resulting simulated annual MPB PP. The vanishing runs are represented by the dark blueish lines (PP < 40 g C m$^{-2}$ yr$^{-1}$) while the light blueish to redish color gradient represented the sustainable runs (PP > 40 g C m$^{-2}$ yr$^{-1}$). The simulated annual PP was sensitive to MPB temperature parameters ($T_{opt}$, $T_{max}$ and $T_{amp}$) as the sustainable runs were characterised by specific parameters ranges within the full tested ranges (from 15 to 27 °C for $T_{opt}$, from 20 to 40 °C for $T_{max}$ and from 5 to 25 °C for $T_{amp}$). Runs with combinations which included a $T_{opt}$ value above 27 °C or a $T_{max}$ value below 20 °C resulted in the vanishing of PP. The simulated annual PP was also sensitive to the light

saturation parameter ($E_k$). Runs in which the simulated annual PP was high were characterised by $E_k$ values in the lower part (from 2.5 to 100 W m$^{-2}$) of the full tested range. Annual PP was sensitive to the half-saturation constant for light use ($K_E$) but to a lesser extent as a high annual PP was simulated using a $K_E$ value spanning within the full tested range. Nevertheless, parameters combinations including high $K_E$ values (15 – 20 Ein m$^{-2}$ d$^{-1}$) resulted into highest annual PP. Annual PP was not sensitive to the *P. ulvae* grazing parameters $T_{optZ}$ and $\alpha_Z$ as sustainable runs were obtained within the full tested ranges.

The new figure illustrates all the parameters combinations summarised in Table 5. We included and described Figure 12 in the Results section 3.5 (p. 9, l. 33 to p. 11, l. 13).

**"p. 3, l. 30. Pertuis Charentais sea: is this the proper name for this area ('sea' seems a bit of an exaggeration)?"**

The term "sea" is commonly accepted to describe the study area (e.g. de Montaudoin & Sauriau, 2000; Poirier et al., 2010, Gourbesville et al., 2015). The Pertuis Charentais is characterised by a body of water of few hundred of km$^2$ (see Sauriau & Pigeot, 2010) partly enclosed by land. Such a description corresponds to the geographical definition of a coastal sea.

**"p. 5, l. 24: MST: first occurrence (?), write out."**

The acronym definition first appears on page 2 (l. 18).

**"p. 6, l. 28: Table A3: Table 3? Please provide the default values in addition to the ranges."**

We removed the reference to the Appendix Table A3. We added the default parameters values in the parenthesis in the revised manuscript (p. 6, l. 27-29):

"Each single parameter varied while the others were fixed at the value set by default in the model ($T_{opt}$ = 18 °C; $T_{max}$ = 38 °C; $E_k$ = 100 W m$^{-2}$; $K_E$ = 20 Ein m$^{-2}$ d$^{-1}$; $T_{optZ}$ = 20 °C; $\alpha_Z$ = 15)."

**"p. 6, l. 32-33: winter-spring, summer-fall: it's not clear why these periods were defined."**

Winter-spring corresponds to November to April. Summer-fall corresponds to May to October. This has been changed in the revised manuscript (p.6, l. 33 to p.7, l. 1-3):

"From November to April, the simulated mud temperature was 9.7 ± 2.6 °C in average. The simulated average temperature was twice from May to October reaching 18.3 ± 3 °C."

**"p. 6, l. 33: 18.3 degrees C: why is this important?"**

This value is important because it shows that from May to October MPB thermo-inhibition ($MST > T_{opt}$) can occur in the model. The link between MST and MPB thermo-inhibition is discussed in the Discussion section 4.2.

**"p. 8, l. 6: linear loss term: linear with respect to what? More detail and/or a reference is needed to be able to understand/reproduce this."**

In the model, we assumed a constant rate at which MPB cells were resuspended from the sediment $1^{st}$ cm (the F compartment) during immersion periods. During immersion periods, the generic loss term ($v_F$, 0.003 $h^{-1}$) included the chronic resuspension, the MPB senescence processes and the grazing by subsurface deposit feeders. During emersion periods, the loss term is lower ($m_F$, 0.001 $h^{-1}$) as it only represented the MPB senescence and grazing by subsurface deposit feeders. The linear loss term for resuspension corresponded therefore to a loss rate of 0.002 $h^{-1}$ applied to the MPB biomass in the sediment $1^{st}$ cm. This mathematical formulation was first used by Guarini et al. (2000). On page 8 (l. 8), we added a reference to the Appendix section B1, where more details, along with the reference Guarini et al. (2000), are also given about the mathematical formulation of resuspension in our study.

**"p. 10, l. 23: KE correlated with primary production. Please consider cause and effect. It's the other way around. Please carefully check through the manuscript: there are many similar instances."**

As suggested by the referee, we checked the manuscript and corrected all the similar instances.

**"p. 12, l. 29: Such a value...etc: grammatically incorrect."**

The sentence was modified as follows (p. 13, l. 2-3): "Such a value suggests that the benthic MPB production contributes significantly to the pelagic food web functionning (Perissinotto et al., 2003; Krumme et al., 2008)."

**"p. 13, l. 7: heat inertia: really? Is this not caused by the water temperature?"**

The referee's comment is very relevant. We hence modified the text to focus more on the results described in page 13 (l. 13-17):

"In fall, the average solar irradiance during daytime exposure periods decreased faster than the simulated MST. The simulated MST departs slower from the temperature optimum for photosynthesis than does the downward irradiance from the light saturation parameter. Despite decreasing solar irradiance in fall, the simulated MPB PP increases until November, when the simulated MPB growth rate is limited by low light levels and MST values with respect to the MPB light saturation parameter (100 W m$^{-2}$) and temperature optimum for photosynthesis (18 °C), respectively."

**"p. 14, l. 6-10. Salinity. This is really ugly. Two wrongs do not make one right."**

We agree with the referee's comment that this part is confusing. We modified the text as follows (p. 14, l. 13-17):

"The detrimental effects of high salinity levels are not explicitly accounted for in the model. The underlying processes could be accounted for in the model in an implicit way by relating the MPB temperature-related growth parameters to PP, in simulated conditions when high evaporation is associated to high MST. The detrimental effects of desiccation on MPB cells motility could also be implicitly represented in the model through more photo-inhibition."

**"p. 14, l. 22. neap tides. Please check; should this not be spring tides?"**

We meant neap tides because light levels are lower during the emersion periods occuring early and late in the day during the neap tides conditions in summer. In the model, only low light levels limit the MPB growth through shading (lower part of the Production-Irradiance model of Platt & Jassby (1976)). In spring tides, even if the light levels are high during the emersion period occuring in the middle of the day, the MPB growth is not affected as the P-E model reaches a plateau at saturating light levels.

**"p. 16, l. 15. What about P. ulvae mobility? Can't they move horizontally to where the food is?"**

We agree with the referee that *P. ulvae* mobility is not discussed in the text. The sentence was modified as follows (p. 16, l. 18-19):

"Second, the *P. ulvae* density on the mudflat can change horizontally as a result of the foraging activity of the individuals and transport mediated by the wave- and tidal-induced shear stress on the bottom sediment."

**Cited references in the answer to referees:**

De Montaudouin, X., & Sauriau, P. G. (2000). Contribution to a synopsis of marine species richness in the Pertuis Charentais Sea with new insights in soft-bottom macrofauna of the Marennes-Oléron Bay. Cahiers de Biologie marine, (2).

Gourbesville, P., Cunge, J. A., & Caignaert, G. (Eds.). (2015). *Advances in Hydroinformatics: SIMHYDRO 2014*. Springer.

Guarini, J. M., Blanchard, G. F., Gros, P., Gouleau, D., & Bacher, C. (2000). Dynamic model of the short-term variability of microphytobenthic biomass on temperate intertidal mudflats. Marine Ecology Progress Series, 195, 291-303.

Poirier, C., Sauriau, P. G., Chaumillon, E., & Bertin, X. (2010). Influence of hydro-sedimentary factors on mollusc death assemblages in a temperate mixed tide-and-wave dominated coastal environment: implications for the fossil record. *Continental Shelf Research*, *30*(17), 1876-1890.

Sauriau, P. G., & Pigeot, J. (2010). Contribution à l'inventaire de la macrofaune marine en baie de Marennes-Oléron. *Ann. Société Sci. Nat. Charente-Marit*, *10*(1), 23-44.

[revised manuscript text omitted]
\left(z_{top},t\right)}{\partial t} = f\left(T_W\left(z_{top},t\right)\right), \tag{A5}$$

with $f\left(T_W\left(z_{top},t\right)\right) = R_S + R_{Atm} - R_W - S_{Air \rightarrow Water}$, $\tag{A6}$

where $\rho_W$ is the volumetric mass of water ($\mathrm{kg\,m^{-3}}$). $C_{P_W}$ is the specific heat capacity of seawater at constant pressure
10     ($\mathrm{J\,kg^{-1}\,K^{-1}}$). $T_W\left(z_{top},t\right)$ is the water temperature (K) in the surface mixed layer. The term $S_{Air \rightarrow Water}$ is the sensible heat flux ($\mathrm{W\,m^{-2}}$) mediated by thermal conduction due to water-air temperature differences. $R_W$ ($\mathrm{W\,m^{-2}}$) is the seawater upward radiation.

    The upper fraction of the water column influenced by atmospheric forcings is defined by the coefficient $\alpha_{top}$:

$$\alpha_{top} = 0.15\left(1 + \frac{U}{3}\right), \tag{A7}$$

15     where $U$ is the wind speed ($\mathrm{m\,s^{-1}}$). Consequently, the simulated seawater temperature of the whole water column ($T_W$) results from the mixing between the fraction $\alpha_{top}$ and the remaining fraction of the water column ($1 - \alpha_{top}$):

$$T_W(t) = \alpha_{top}T_W\left(z_{top},t\right) + \left(1 - \alpha_{top}\right)T_W\left(z_{bot},t\right) \text{ with } T_W\left(z_{bot},t\right) = 
[revised manuscript text omitted]
 | 0.21 | 0.13 | 0.11 | 0.11 | 0.15 | 0.22 | 0.26 | 0.23 | 0.10 | 0.23 | 0.19 | 0.15 |